# On the Need for a Language Describing Distribution Shifts: Illustrations on Tabular Datasets

**Jiashuo Liu**[1,*] **Tianyu Wang**[2,*]**, Peng Cui**[1]**, Hongseok Namkoong**[3]
[1]Department of Computer Science and Technology, Tsinghua University
[2]Department of Industrial Engineering and Operations Research, Columbia University
[3]Decision, Risk, and Operations Division, Columbia Business School
liujiashuo77@gmail.com, tw2837@columbia.edu
cuip@tsinghua.edu.cn, namkoong@gsb.columbia.edu

## Abstract

Different distribution shifts require different algorithmic and operational interventions. Methodological research must be grounded by the specific shifts they address. Although nascent benchmarks provide a promising empirical foundation, they *implicitly* focus on covariate shifts, and the validity of empirical findings depends on the type of shift, e.g., previous observations on algorithmic performance can fail to be valid when the $Y|X$ distribution changes. We conduct a thorough investigation of natural shifts in 5 tabular datasets over 86,000 model configurations, and find that $Y|X$-shifts are most prevalent. To encourage researchers to develop a refined language for distribution shifts, we build WHYSHIFT, an empirical testbed of curated real-world shifts where we characterize the type of shift we benchmark performance over. Since $Y|X$-shifts are prevalent in tabular settings, we *identify covariate regions* that suffer the biggest $Y|X$-shifts and discuss implications for algorithmic and data-based interventions. Our testbed highlights the importance of future research that builds an understanding of why distributions differ.[2]

## 1 Introduction

The performance of predictive models has been observed to degrade under distribution shifts in a wide range of applications, such as healthcare [9, 95, 76, 93], economics [36, 25], education [6], vision [74, 63, 86, 98], and language [62, 7]. Distribution shifts vary in type, typically defined as either a change in the marginal distribution of the covariates ($X$-shifts) or the conditional relationship between the outcome and covariate ($Y|X$-shifts). Real-world scenarios comprise of both types of shifts. In computer vision [62, 47, 81, 38, 101], $Y|X$-shifts are less likely to occur as $Y$ is constructed from human knowledge given an input $X$, unless the labeling noise is severe. For tabular datasets, $Y|X$-shifts can be more common because of missing variables and hidden confounders. For example, the prevalence of a disease may be affected by unrecorded covariates whose distribution changes across domains, such as lifestyle factors and socioeconomic status [39, 103, 93].

Different types of distribution shifts require different solutions. When facing $X$-shifts, the implicit goal of many researchers is to develop a single robust model that can be generalized effectively across multiple domains. Various algorithms have been developed to align the marginal distributions ($P_X$), including domain adaptation and importance sampling methods. However, under $Y|X$-shifts, there may be a fundamental trade-off between learning algorithms: to perform well on a target distribution, a model may have to *necessarily* perform worse on others. Algorithmically, typical methods for

---

*Equal contribution

[2]More information on the data, codes and python packages about WHYSHIFT are available at https://github.com/namkoong-lab/whyshift.

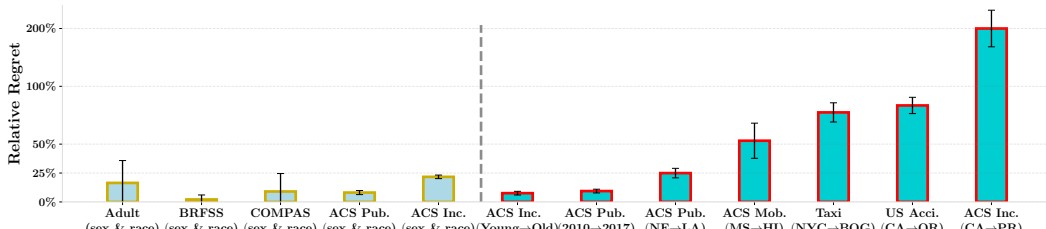

**Figure 1.** *Relative regret* (1.1) in typical benchmarks [26, 31] (*left 5 bars*) and seven settings designed in our benchmark (*right 7 bars*). We use XGBoost $\mathcal{F}$ here for illustration.

addressing $Y|X$-shifts include distributionally robust optimization (DRO) [13, 85, 28, 80, 27] and causal learning methods [72, 8, 83, 46]. Operationally, the modeler can identify and collect an unobserved confounder $C$ such that $Y|X, C$ remains invariant across domains, or resort to overhauling the entire model development pipeline to collect more samples from the target.

However, existing distribution shift benchmarks only focus on $X$-shifts [74, 47, 107, 81, 101]. To illustrate this concretely, consider popular tabular datasets used to benchmark model performance over demographic subgroups: `Adult`, `BRFSS`, `COMPAS`, `ACS Public Coverage`, and `ACS Income` [3, 102, 31]. We take the largest demographic group as training $P$ and the smallest as target $Q$ to simulate subgroup shifts, e.g., in `Adult`, $P$ =white men and $Q$ =non-white women. We measure the optimality gap of the model $f_P$ trained on $P$ as measured on the target $Q$ using the *relative regret*

$$\frac{\mathbb{E}_Q[\ell(Y, f_P(X))]}{\min_{f \in \mathcal{F}} \mathbb{E}_Q[\ell(Y, f(X))]} - 1, \quad \text{where} \quad f_P \in \underset{f \in \mathcal{F}}{\operatorname{argmin}} \mathbb{E}_P[\ell(Y, f(X))] \qquad (1.1)$$

and $\ell(\cdot, \cdot)$ is the 0-1 loss. For these widely-used benchmarks, the relative regret is small (left 5 bars in Figure 1), suggesting the $Y|X$ distribution is *largely transferable* across those demographic groups.

To study diverse distribution shift patterns, we consider 5 real-world tabular datasets constructed from the US Census (as proposed by Ding *et al.* [25]) and traffic measurements [64, 65, 2, 1]. We focus on spatiotemporal shifts to model most common natural shifts. Our full benchmark covers 22 settings (see Table 3 in Appendix D), where each setting includes one source (e.g., California) and a number of possible targets (e.g., other states). For illustration purposes, we focus on 7 settings covering 169 possible source-target pairs (see Table 2) and carefully select one target per setting to represent a wide range of $Y|X$-shifts (right bars in Figure 1).

We find $Y|X$-shifts constitute a substantial proportion of real-world distribution shifts, yet previous (unqualified) empirical findings in the literature only hold over mild $X$-shifts and fail to hold over $Y|X$-shifts (Section 2). Out of 169 source-target pairs with significant performance degradation ($> 8$ percentage points of accuracy drop), 80% of them are primarily attributed to $Y|X$-shifts. $Y|X$-shifts introduce considerable performance variations on the target distribution, leading to different relationships between in- and out-of-distribution performances across settings and datasets. This is in stark contrast to the recently observed accuracy-on-the-line phenomenon [63], where the in- and out-of-distribution performance have been posited to exhibit a strong linear relationship. In Figure 2, we showcase how the accuracy-on-the-line trend fails to hold when $Y|X$-shifts are strong. Our results imply that the standard practice of blindly evaluating performance over various shifts is only justified over $X$-shifts, where we expect there to be a single model that is robust across domains.

For severe $Y|X$-shifts, the training data may not even be informative for modeling the $Y|X$ relationship in the target. To inform algorithmic and data-based interventions, we must understand *why* the distribution changed. In Section 3, we illustrate the need for more methodological research that builds a deep understanding of distributional differences. As a concrete example, we show that a simple approach for identifying covariate regions with strong $Y|X$-shifts can suggest data-based interventions. Our case study shows it can be useful to collect target data over a particular covariate region, or features $C$ such that the $Y|X, C$ distribution is more stable across source and target.

In Section 4, we provide the details of our benchmark, WHYSHIFT. We use five datasets to construct 7 spatiotemporal distribution shifts, and evaluate 22 methods over 86,000+ model configurations. We compare a broad range of algorithms including tree ensembles, DRO, imbalance, and fairness methods and summarize our key findings below.

- Rankings of model performance change over different shift patterns. As the validity of empirical findings implicitly depends on the type of shift, any methodological development must be grounded by the specific shifts it addresses.
- Tree ensemble methods are competitive, but still suffer from significant performance degradation.

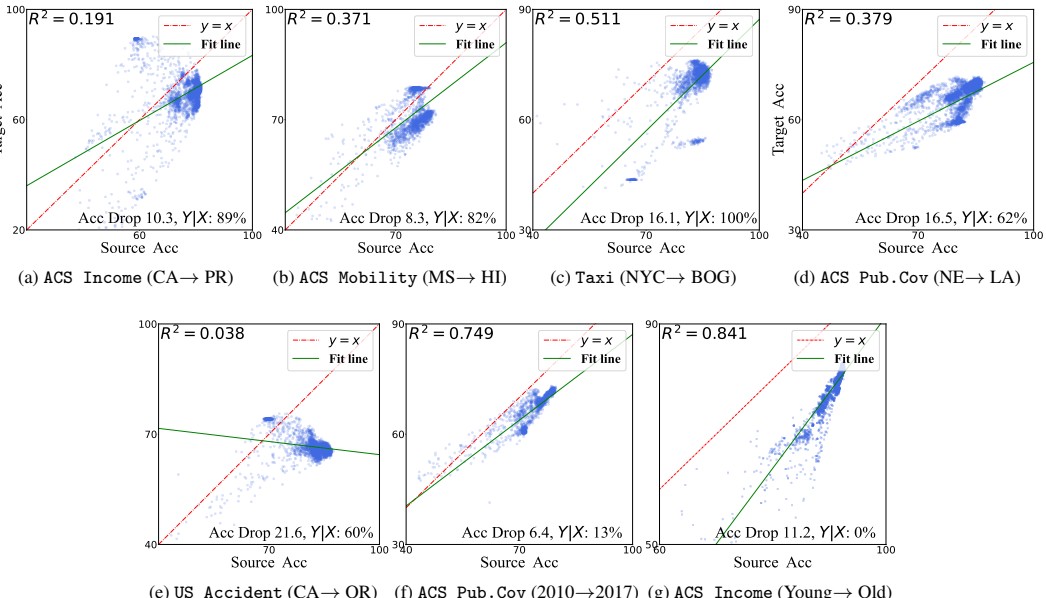

**Figure 2.** Target vs. source accuracies for 22 algorithms and datasets in our benchmark. A linear fit (green line) and its corresponding $R^2$ value is reported on the top left of each figure. Each blue point represents one hyperparameter configuration. **(a)-(f)**: six examples of `ACS Income`, `ACS Mobility`, `Taxi`, `ACS Pub.Cov`, `US Accident` datasets. **(g)**: simulated covariate shifts on on sub-sampled `ACS Income` dataset.

- DRO methods are sensitive to configurations and exhibit significant performance variations.
- Imbalance and fairness methods show similar performance with the base learner (XGBoost).
- A small validation data from the target distribution goes a long way, and more generally, non-algorithmic interventions warrant greater consideration.

## 2 Distribution Shifts in Tabular Settings

To illustrate how complex distribution shift patterns arise in tabular data. we compare 22 algorithms including tree ensemble methods, robust learning, imbalance, and fairness methods. On 5 real-world tabular datasets (`ACS Income`, `ACS Public Coverage (ACS Pub.Cov)`, `ACS Mobility` [25], `US Accident` [64, 65] and `Taxi` [2, 1]), we consider the natural spatial shifts between states/cities, e.g., California to Puerto Rico. For the `ACS Pub.Cov` dataset, we also consider temporal shifts, e.g., from 2010 to 2017. Since all natural distribution shifts we consider are largely induced by $Y|X$-shifts, we construct a synthetic subgroup shift from younger people to older people in order to simulate $X$-shifts. Deferring a detailed summary to Section 4.1, we focus on introducing representative phenomena in this section.

In Figure 2, we present the source (in-distribution) and target (out-of-distribution) performances of 22 algorithms, each with 200 hyperparameter configurations. To understand shift patterns, we utilize the recently proposed DIstribution Shift DEcomposition (DISDE) framework [16] which decomposes the performance degradation into components attributed to $Y|X$- and $X$-shifts. Using the best XGBoost configuration as the baseline model for each source-target pair, we present the total performance degradation and the proportion attributed to $Y|X$-shifts.

**Distribution shifts are predominantly $Y|X$-shifts in our empirical study** We find performance degradation under natural shifts is overwhelmingly attributed to $Y|X$-shifts, as illustrated in the curated list in Figure 2. More generally, out of the 169 source-target pairs whose performance degradation is larger than 8 percentage points, 87.2% of them have over 50% of the performance degradation attributed to $Y|X$-shifts (70.2% of them have over 60% of the gap attributed to $Y|X$-shifts). We conjecture that $Y|X$-shifts are prevalent in tabular data due to missing features. For example, in the context of income prediction, individual outcomes may change due to unobserved economic and political factors whose distribution changes over geographical locations [25]. In contrast, in vision and language tasks, the input (e.g., pixels and words) often encapsulates most of the necessary information for predicting the outcome, making strong $Y|X$-shifts less likely unless the labeling noise is severe. Consequently, compared to vision and language data in domain generalization

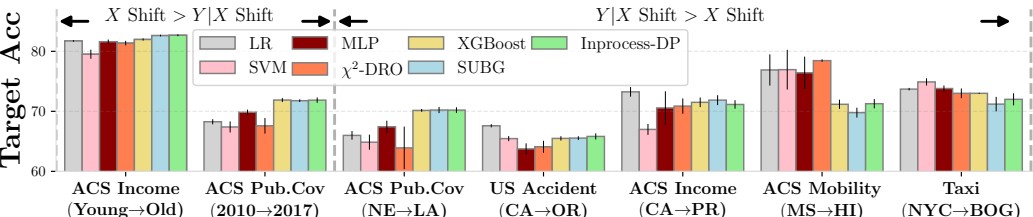

**Figure 3:** Performances of typical algorithms of 7 settings in our benchmark.

tasks, tabular data exhibits more pronounced real $Y|X$-shifts. Our findings highlight the importance of understanding the *cause* of the distribution shift.

**Accuracy-on-the-line fails to hold over $Y|X$-shifts**   We find significant variation in the relationship between source and target performance throughout all natural distribution shifts presented in Figure 2 (a)-(g). The correlation between source and target performance is relatively weak, and we tend to see poor linear fits (low $R^2$) when the bulk of the performance degradation is attributed to $Y|X$-shifts. That is, we see in Figures 2 (a)-(f) that the relationship between the two performances exhibits significant fluctuations across different source-target pairs. In Figure 3, we observe that performance rankings of algorithms substantially vary across different $Y|X$-shifts. Our finding highlights the inherent complexity associated with real distribution shifts in tabular datasets, which stands in sharp contrast to the "accuracy-on-the-line" phenomena [63]. The varied shift patterns in tabular data highlight how empirical observations must be qualified over the range of shifts they remain valid over. This is particularly important for $Y|X$-shifts which introduce larger variations in the relationship between source and target performance.

**Source and target performances are correlated when $X$-shifts dominate**   Across all natural shifts we study, we find $X$-shifts are only prominent in temporal shifts (`ACS Time` dataset; Figure 2f) To better investigate the role of $X$-shifts, we subsample the data to artificially induce strong covariate shifts over an individual's age. Specifically, we focus on individuals from California and form two groups according to whether their age is $\geq 25$. The source data oversamples low age groups where $80\%$ is drawn from the age $\leq 25$ group; proportions are reversed in the target data.

On this synthetic shift we construct, the DISDE [16] method attributes the bulk of the performance degradation to $X$-shifts in Figure 2g. Our finding confirms the intuition that unobserved economic factors remain relatively consistent for individuals from the same state (CA). In this synthetic example with $X$-shifts, we observe a relatively strong correlation between source and target performance. Moreover, the large performance degradation on these datasets suggests that existing robust learning methods are still severely affected by covariate shifts, indicating the need for future research that addresses covariate shifts in tabular data.

## 3   Case Study: Understanding Distribution Shifts Facilitates Interventions

Typical algorithmic approaches to handling practical distribution shifts aim to optimize performance over a postulated set of distribution shifts. Causal learning assumes the underlying causal structure can be learned to withstand distribution shifts [72, 83, 82, 77], while DRO methods explicitly optimize worst-case performance over a set of distributions [13, 53, 28, 27]. Despite progress in algorithm design, there are few efforts that examine the patterns of real-world distribution shifts. It remains unclear whether the data assumptions made by algorithms hold in practice, and this *mismatch* often leads to poor empirical performance [40, 95, 76, 19, 46].

Complementing the active literature on algorithmic development, we present an empirical study that underscores the practical significance of tools that provide a qualitative understanding of the shift at hand. In light of the prevalence of $Y|X$-shifts in tabular data, we introduce a simple yet effective approach for identifying covariate regions that suffer strong $Y|X$-shifts. We demonstrate our approach on the income prediction task (`ACS Income`), and show that it can guide operational interventions for addressing distribution shifts. Our case study is not meant to be a rigorous scientific analysis, but rather a (heuristic) vignette illustrating the need for future research on methodologies that can generate qualitative insights on distributional differences.

## 3.1  Identifying Regions with Strong $Y|X$-shifts

Here we propose a simple yet effective method for identifying covariate regions with strong $Y|X$-shifts. Despite its simplicity, we demonstrate in the following subsections that our method can inspire operational and modeling interventions. Letting $(X, Y)$ be random variable supported on the space $\mathcal{X} \times \mathcal{Y}$, consider a model $f : \mathcal{X} \to \mathcal{Y}$ that predicts outcome $Y \in \mathcal{Y}$ from covariates $X \in \mathcal{X}$ with the associated loss function $\ell(f(X), Y)$. Given samples $(X, Y)$ drawn from the source and target distributions $P$ and $Q$, our goal is to *identify a region $\mathcal{R} \subseteq \mathcal{X}$ where $P_{Y|X}$ differs a lot from $Q_{Y|X}$*.

Since $P_{Y|X}$ and $Q_{Y|X}$ are undefined outside of the support of $P_X$ and $Q_X$ respectively, the comparison can only be made on a subset of the *common support*. To aid comparisons on the common support, Cai *et al.* [16] introduced a shared distribution approach. The shared distribution has high density when both $p_X$ and $q_X$ are high, and low density whenever either is small. Following Cai *et al.* [16], we choose a specific *shared distribution* $S_X$ over $X$ from the likelihood ratio:

$$s_X(x) \propto p_X(x)q_X(x)/(p_X(x) + q_X(x)). \tag{3.1}$$

We provide more discussion on the choices of $S_X$ and other technical details justifying the correctness of $s_X$ in Appendix C.3. Since we do not have access to samples from the shared distribution $S_X$, we reweight samples from $P_X$ and $Q_X$ using the likelihood ratios $s_X(x)/p_X(x) \propto q_X(x)/(p_X(x) + q_X(x))$ and $s_X(x)/q_X(x) \propto p_X(x)/(p_X(x) + q_X(x))$. The ratio can be modeled as the probability that an input $x$ is from $P_X$ vs $Q_X$. Denote $\alpha^*$ as the proportion of the pooled data that comes from $Q_X$ and $\pi^*(x) := \mathbb{P}(\tilde{X} \text{ from } Q_X | \tilde{X} = x)$, we can express the likelihood ratios as:

$$\frac{s_X}{p_X}(x) \propto \frac{\pi^*(x)}{(1 - \alpha^*)\pi^*(x) + \alpha^*(1 - \pi^*(x))} =: w_P(\pi^*(x), \alpha^*), \tag{3.2}$$

$$\frac{s_X}{q_X}(x) \propto \frac{1 - \pi^*(x)}{(1 - \alpha^*)\pi^*(x) + \alpha^*(1 - \pi^*(x))} =: w_Q(\pi^*(x), \alpha^*). \tag{3.3}$$

With the likelihood ratios, we estimate the best prediction model under $P$ and $Q$ over the shared distribution $S_X$ (using XGBoost as the model class $\mathcal{F}$):

$$f_\mu := \arg\min_{f \in \mathcal{F}} \left\{ \mathbb{E}_{S_X} \left[ \mathbb{E}_\mu[\ell(f(X), Y)|X] \right] \left( = \mathbb{E}_\mu \left[ \ell(f(X), Y)w_\mu(\pi^*(x), \alpha^*) \right] \right) \right\}, \text{ for } \mu = P, Q. \tag{3.4}$$

Then, for any threshold $b \in [0, 1]$, $\{x \in \mathcal{X} : |f_P(x) - f_Q(x)| \geq b\}$ suggests a region that may suffer model performance degradation with at least $b$ due to $Y|X$-shifts. Without evaluating the performance on the shared distribution $S_X$, it is hard to distinguish the source of the model performance degradation, i.e. from $X$-shifts or $Y|X$-shifts.

Empirically, given samples $\{(x_i^P, y_i^P)\}_{i \in [n_P]}$ from $P$ and $\{(x_j^Q, y_j^Q)\}_{j \in [n_Q]}$ from $Q$, we estimate $\hat{\alpha} = \frac{n_Q}{n_P + n_Q}$ and then train a binary "domain" classifier $\hat{\pi}(x)$ to approximate the ratio $\pi^*(x)$. Note that the "domain" classifier can be any black-box method, and we use XGBoost throughout. Then we plug these empirical estimands in to obtain the estimated likelihood ratios $w_\mu(\hat{\pi}(x), \hat{\alpha})$ and learn prediction models $f_P$ and $f_Q$ in Equation (3.4). To investigate the model difference under $S_X$, we pool samples from $P$ and $Q$ together and set sample weights as:

$$\lambda_i^P = \frac{w_P(\hat{\pi}(x_i^P), \hat{\alpha})}{\sum_{k \in [n_P]} w_P(\hat{\pi}(x_k^P), \hat{\alpha})} \quad \forall i \in [n_P], \quad \lambda_j^Q = \frac{w_Q(\hat{\pi}(x_j^Q), \hat{\alpha})}{\sum_{k \in [n_Q]} w_Q(\hat{\pi}(x_k^Q), \hat{\alpha})} \quad \forall j \in [n_Q], \tag{3.5}$$

which are used to learn a prediction model $h(x)$ to approximate $|f_P(x) - f_Q(x)|$ on the *shared distribution $S_X$*. The pseudo-code is summarized in the Algorithm 1; To allow simple interpretation and efficient region identification, we use a shallow *decision tree* $h(x)$ and consider the region $\mathcal{R}$ corresponding to the feature range of a leaf node within the tree. More details could be found in Appendix C.5 and Appendix C.6. We show that the node splitting criterion in a standard decision trees training procedure is equivalent with our goal of finding regions with the largest discrepancy in Appendix C.4.

## 3.2  Data-based Interventions

Using Algorithm 1, we now demonstrate how a better understanding of distribution shifts can facilitate the design of interventions. We focus on the `ACS Income` dataset where the goal is to predict whether

---
**Algorithm 1:** Identify Regions with Strong $Y|X$-Shifts.
---
**Input:** Source samples $\{(x_i^P, y_i^P)\}_{i \in [n_P]} \overset{\text{i.i.d}}{\sim} P$ and target samples $\{(x_j, y_j)\}_{j \in [n_Q]} \overset{\text{i.i.d}}{\sim} Q$.
  Model discrepancy threshold $b$.

1  Estimate $\hat{\pi}(x) \approx \mathbb{P}(\tilde{X} \sim Q_x | \tilde{X} = x)$ by training a classifier on the source and target samples.
2  Calculate density ratios $w_\mu(\hat{\pi}(x), \hat{\alpha})$ according to Equation (3.2) and (3.3) for $\mu = P, Q$.
3  Fit prediction models $f_\mu$ according to Equation (3.4) replacing $w_\mu(\pi^*(x), \alpha^*)$ there with
   $w_\mu(\hat{\pi}(x), \hat{\alpha})$ for $\mu = P, Q$.
4  Fit a model $h(x)$ to predict $|f_P(x) - f_Q(x)|$ using samples $\{(x_i^P, y_i^P)\}_{i \in [n_P]}$ and
   $\{(x_j^Q, y_j^Q)\}_{j \in [n_Q]}$, each with the weight $\lambda_i^P$ (or $\lambda_j^Q$ respectively), according to Equation (3.5).

**Output:** Region $\mathcal{R} = \{x \in \mathcal{X} : h(x) \geq b\}$.

---

an individual's income exceeds 50k ($Y$) based on their tabular census data ($X$). We train an income classifier on 20,000 samples from California (CA, source), and deploy the classifier in Puerto Rico and South Dakota (PR & SD, target), where we get 4,000 samples from PR and SD after deployment. Given the considerable disparities in the economy, job markets, and cost of living between CA and PR/SD, we observe substantial performance degradation due to distribution shifts.

In Figure 4a, we first decompose the performance degradation from CA to PR to understand the shift and find $Y|X$-shifts are the predominant factor. The calculation of $X$-shifts and $Y|X$-shifts is deferred to Appendix C.1. We dive deeper into the significant $Y|X$-shifts and identify from CA to PR for the XGBoost and MLP classifier. From the region shown in Figure 4c and Figure 4d, we find college-educated individuals in business and educational roles (such as management, business, and educational work) exhibit large $Y|X$ differences.

To illustrate how our analysis can inspire subsequent operational interventions to enhance performance on the target distribution, we study two operational interventions.

**Collect specific data from the target**     To improve target performance, the most natural operational intervention is to collect additional data from the target distribution. While a rich body of work on domain adaptation [69, 23, 30, 90, 89] study how to effectively utilize data from the target distribution to improve performance, there is little work that discusses how to efficiently collect supervised data from the target distribution to maximize out-of-distribution generalization. To highlight the need for future research in this space, we use the interpretable region identified by Algorithm 1 as shown in Figure 4c to simulate a concerted data collection effort.

Since indiscriminately collecting data from the target distribution can be resource-intensive, we concentrate sampling efforts on the subpopulation that may suffer from $Y|X$-shifts and selectively gather data on them. For five base methods (logistic regression, MLP, random forest, lightGBM, and XGBoost), we randomly sample 250 points from the whole target distribution and the identified region suffering prominent $Y|X$-shifts, respectively. We report the test accuracies in Figure 4e, and observe that incorporating data from this region is more effective in enhancing OOD generalization. While preliminary, our results demonstrate the potential robustness benefits of efficiently allocating resources toward concerted data collection. Future methodological research in this direction may be fruitful; potential connections may exist with active learning algorithms [84, 96, 59].

**Add more relevant features**     We now illustrate the potential benefits of generating qualitative insights on the distribution shift at hand. Our analysis in Figure 4c suggests educated individuals in financial, educational, and legal professions tend to experience large $Y|X$-shifts from CA to PR. These roles typically need communication skills, and language barriers could potentially affect their incomes. In California (CA), English is the primary language, while in Puerto Rico (PR), despite both English and Spanish being recognized as official languages, Spanish is predominantly spoken. Consequently, for a model trained on CA data and tested on PR data, incorporating a new feature that denotes English language proficiency (hereafter denoted "ENG") might prove beneficial in improving generalization performances. However, this feature is not included in the ACS Income dataset.

To address this, we went back to the Census Bureau's American Community Survey database to include the ENG feature in the set of covariates. In Figure 4b, we observe that the inclusion of this feature substantially reduces the degradation due to $Y|X$-shifts, verifying that the originally missing

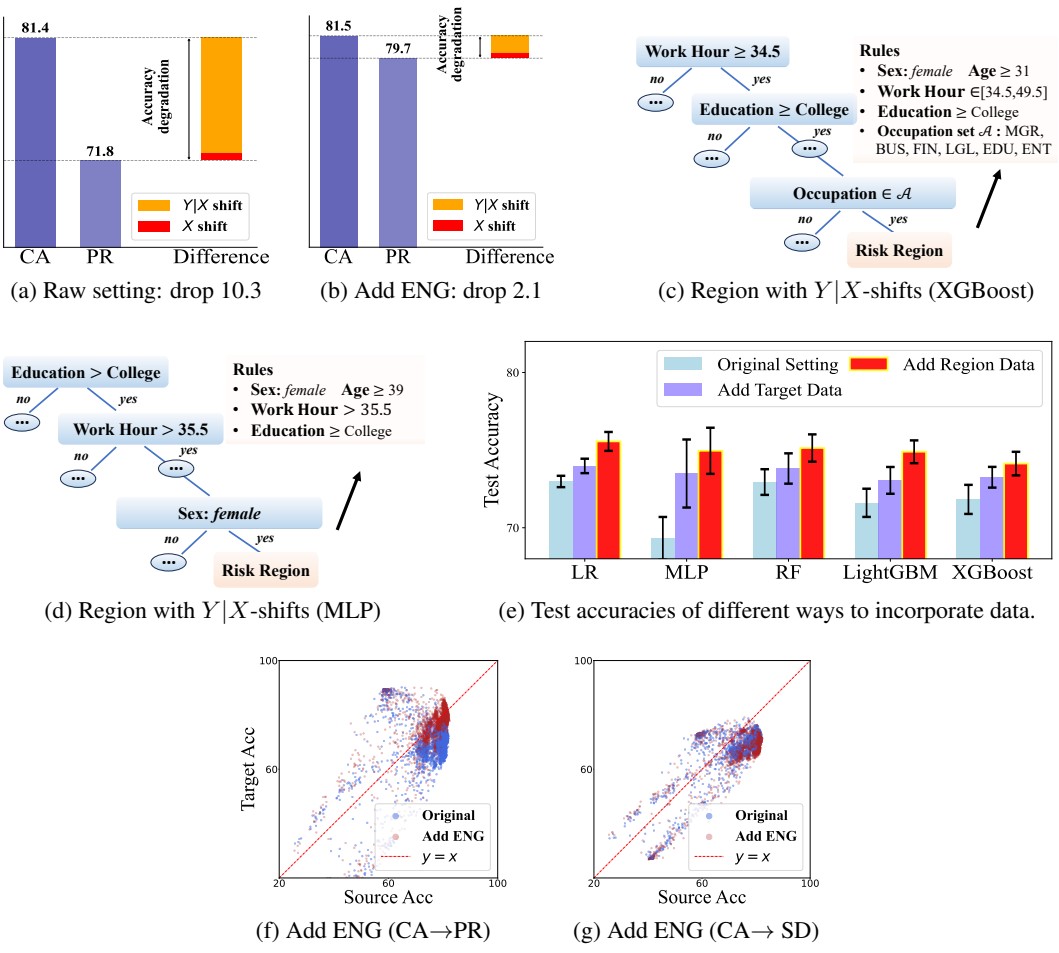

**Figure 4.** Case study illustrations. **(a)-(b)** Decomposition of performance degradation for the XGBoost classifier from CA to PR. Figure (a) is for the original setting and (b) corresponds to the results post-integration of the "ENG" feature. **(c)-(d)** Demonstration of Algorithm 1: an interpretable version of the region with strong $Y|X$-shifts for the XGBoost and MLP models, respectively. **(e)** Test accuracies of five typical base methods trained on the source, post addition of 250 randomly selected *target observations*, and 250 observations from the identified *risk region*. **(f)-(g)** Performances of all algorithms prior to and following the addition of the "ENG" feature. Figure (f) corresponds to the CA to PR, and Figure (g) is CA to SD.

**Table 1:** Overview of datasets and 7 selected settings.

| #ID | Dataset | Type | #Samples | #Features | Outcome | #Domains | Selected Settings | Shift Patterns |
|-----|---------|------|----------|-----------|---------|----------|-------------------|----------------|
| 1 | ACS Income | Natural | 1,599,229 | 9 | Income≥50k | 51 | California → Puerto Rico | $Y|X \gg X$ |
| 2 | ACS Mobility | Natural | 620,937 | 21 | Residential Address | 51 | Mississippi → Hawaii | $Y|X \gg X$ |
| 3 | Taxi | Natural | 1,506,769 | 7 | Duration time≥30 min | 4 | New York City→ Botogá | $Y|X \gg X$ |
| 4 | ACS Pub.Cov | Natural | 1,127,446 | 18 | Public Ins. Coverage | 51 | Nebraska → Louisiana | $Y|X > X$ |
| 5 | US Accident | Natural | 297,132 | 47 | Severity of Accident | 14 | California→ Oregon | $Y|X > X$ |
| 6 | ACS Pub.Cov | Natural | 859,632 | 18 | Public Ins. Coverage | 4 | 2010 (NY)→ 2017 (NY) | $Y|X < X$ |
| 7 | ACS Income | Synthetic | 195,665 | 9 | Income≥50k | 2 | Younger→ Older | $Y|X \ll X$ |

ENG feature may be one cause of $Y|X$-shifts. Figure 4f contrasts the performances of 22 algorithms (each with 200 hyperparameter configurations) with original features with those that additionally use the ENG feature. The new feature significantly improves target performances across all algorithms; roughly speaking, we posit that we have identified a variable $C$ such that $Y|X, C$ remains similar across CA and PR. However, when we extend this comparison to the source-target pair (CA → SD), we observe no significant improvement (Figure 4g). This highlights that the selection of new features should be undertaken judiciously depending on the target distributions of interest. A feature that proves effective in one target distribution might not yield similar results in another.

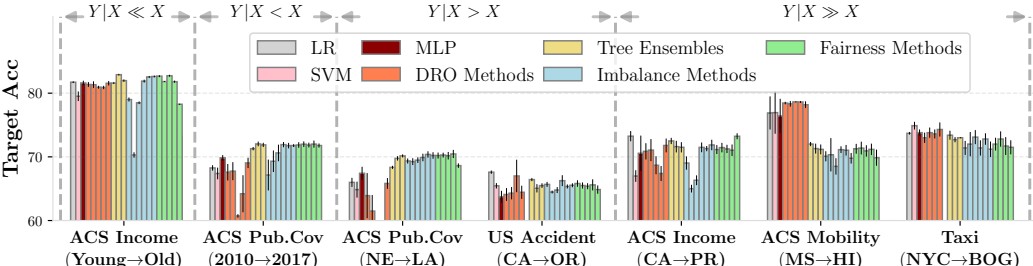

**Figure 5:** Overall performances of all algorithms on the target data in our selected 7 settings.

# 4 WHYSHIFT: Benchmarking Distribution Shifts on Tabular Data

In this section, we detail our benchmark and summarize the main observations. Our finding highlights the importance of future research that builds an understanding of *why* the distribution has shifted.

## 4.1 Setup

**Datasets**    We explore distribution shifts on 5 real-world tabular datasets from the economic and traffic sectors with *natural* spatiotemporal distribution shifts. For economic data, we use `ACS Income`, `ACS Mobility`, and `ACS Public Coverage` datasets from the US-wide ACS PUMS data [25], where the outcome is whether an individual's income exceeds 50k, whether an individual changed the residential address one year ago, and whether an individual is covered by public health insurance, respectively. We primarily focus on spatial shifts across different states in the US. To complement spatial shifts, we derive an `ACS Time` task based on the `ACS Public Coverage` dataset, where there are temporal shifts between different years (2010 to 2021). For traffic data, we use `US Accident` [64, 65] and `Taxi` [2, 1], where the outcome is whether an accident is severe and whether the total ride duration time exceeds 30 minutes, respectively. We focus on spatial shifts between different states/cities. We summarize the datasets in Table 1 and defer a full description to the Appendix D.1.

**Algorithms**    We evaluate **22** algorithms that span a wide range of learning strategies on tabular data, and compare their performances under different patterns of distribution shifts we construct. Concretely, these algorithms include: (1) *base learners*: Logistic Regression, SVM, fully-connected neural networks (MLP) with standard ERM optimization; (2) *tree ensemble models*: Random Forest, XGBoost, LightGBM; (3) *robust learning*: CVaR-DRO and $\chi^2$-DRO with fast implementation [53], CVaR-DRO and $\chi^2$-DRO of outlier-robust enhancement [105], Group DRO [79]; (4) *imbalanced learning*: JTT [57], SUBY, RWY, SUBG, RWG [42], DWR [51] and (5) *fairness-enhancing methods*: inprocessing method [4] with demographic parity, equal opportunity, error parity as constraints, postprocessing method [37] with exponential and threshold controls. For DRO methods (i.e. (3)), we use MLP as the backbone model. For other algorithms compatible with tree ensemble models (i.e. (4-5)), we use the XGBoost model due to its superior performance on tabular data [34]. For algorithms requiring group labels, we use 'hour' for `US Accident` and `Taxi`, and 'sex' for the others. Detailed descriptions for each algorithm can be found in Appendix D.5.

**Benchmarks**    We conduct experiments with more than 86,000 model configurations on various source-target distribution shift pairs, and carefully select *7 selected pairs with different distribution shift patterns*. In Table 1, we characterize the shift patterns of these 7 source-target pairs, which contain different proportions of $Y|X$-shifts and $X$-shifts corresponding with plots in Figure 2. The first six settings are natural shifts. In the last setting, we sub-sample the dataset according to age to introduce covariate shift, where we focus on individuals from CA and form two groups according to whether their age is $\geq 25$. The source data over-samples the low age group where 80% is drawn from the group where the individual's age $\geq 25$, and the proportions are reversed in the target data.

In Figure 5 and Figure 6, we plot the performance of algorithms using their best hyperparameter configuration on the validation dataset (*i.i.d.* with the source distribution). Additional results with various source distributions are in the Appendix. Our benchmark is designed to support empirical research, including new learning algorithms and diagnostics that provide qualitative insights on distribution shifts.

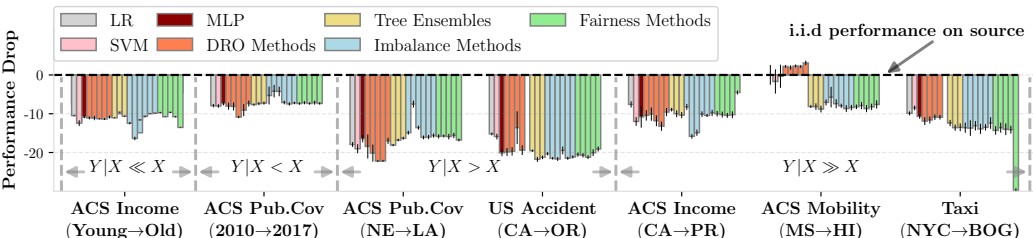

**Figure 6.** Performance drop between source and target data of all algorithms in our selected 7 settings.

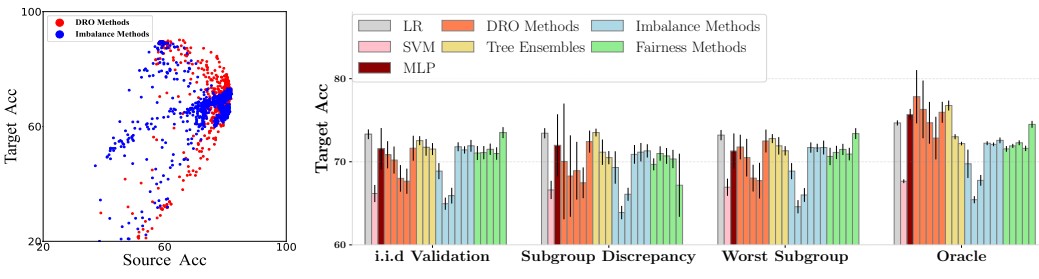

(a) DRO & Imb. Methods    (b) Performances under different validations on `ACS Income` dataset.

**Figure 7. (a)**: Sensitivity of DRO methods and Imbalance Methods w.r.t. configurations. **(b)** Target performances of 22 algorithms under different validation protocols on `ACS Income` (CA→PR) setting.

**Hyper-parameter Tuning**  For each model, we conduct a grid search over a large set of hyper-parameters. See Appendix D.3 for the complete search space for each method. When one method includes another as a "base" learner (e.g., DRO with MLP, RWY with XGBoost), we explore the full tuning space for the base model (e.g., the cross-product of all MLP hyper-parameters with all DRO hyper-parameters). To control for computational effort, each method is run with 200 configurations for each source-target pair and we select the best configuration according to the $i.i.d.$ validation performance. In Figure 7b, we further compare different choices of validation protocols.

**Evaluation Metrics**  In our benchmark, we include different metrics for a thorough evaluation. Specifically, we use Average Accuracy (micro-average), Worst-group Accuracy, and Macro-F1 score in our main results where we only have one target distribution. For the results with multiple target distributions (i.e. 3 in `Taxi`, 13 in `US Accident` and 50 in the others), we present *all* target accuracies and Macro-F1 scores, as well as the worst-distribution accuracy and Macro-F1 score among all target distributions in Appendix D.6, D.7, D.8, D.9.

## 4.2 Analysis

**Different algorithms do not exhibit consistent rankings over different shift patterns.**  In Figure 5, we observe the rankings across different shifts are quite different, especially for `ACS Income` (CA→PR) and `ACS Mobility` (MS→HI) where $Y|X$-shifts dominate. This observation reaffirms the phenomena in Figure 2 that as $Y|X$-shifts become stronger, the relationship between source vs target performances becomes less consistent. In Appendix D.3, we also show that even for a fixed source distribution in one fixed prediction task, algorithmic rankings of performances on different target distributions vary a lot.

**Tree ensemble methods show competitive performance, but do not significantly improve the generalization drop between source and target data.**  From Figure 5, tree-based ensembles (yellow bars) show robust and competitive performance on the target distribution in 6 out of 7 settings. However, in Figure 6 which plots the performance degradation between source and target, tree ensembles do not show improved robustness. This suggests that they do *not* actually achieve better robustness against real-world distribution shifts, and their better performances on target data may simply be due to better fitting the source distribution.

**DRO methods are sensitive to configurations, with rankings varying significantly across 7 different settings.**  From Figure 5, DRO methods exhibit competitive performances on `ACS Mobility` (MS→HI), `Taxi` (NYC→BOG), and `ACS Income` (Young→Old), yet underperform in others. This sensitivity to configurations, as shown in Figure 7a (red points), could be attributed to

the worst-case optimization that perturbs the training distribution within a pre-defined uncertainty set, without any information regarding the target distribution. However, when target information is incorporated for hyper-parameter tuning, as shown in Figure 7b, there is a notable improvement in the performance of DRO methods. Our observations suggest potential avenues for building more refined uncertainty sets in DRO methods.

**Imbalance methods and fairness methods show similar performance with the base learner (XGBoost).** In our experiments, we choose the XGBoost model as the base learner for imbalance and fairness methods due to its superior performance on tabular data [34]. However, from Figure 5 and Figure 6, imbalance methods and fairness methods do not show a clear improvement upon their base learner (XGBoost, last yellow bar). Further, as shown in Figure 7a, imbalance methods (green) are also quite sensitive to configurations, and their performances do not improve much when their hyperparameters are tuned over the target data (Oracle).

**Target information matters in validation.** Based on the ACS Income (CA→PR) dataset, we compare different validation protocols, including the best average accuracy, minimum subgroup discrepancy, and best worst-subgroup accuracy on validation data generated from the *source* distribution. We also use the Oracle validation that chooses the configuration with the best average accuracy on validation data generated from the *target* distribution. In Figure 7b, we find the first three protocols do not show a significant difference. However, oracle validation with target information substantially improves the effectiveness of both DRO and tree ensemble methods. We conclude using target information for model selection can provide robustness gains even with a small target dataset.

**Non-algorithmic interventions warrant greater consideration.** Reflecting on Section 3, it is clear that operational interventions yield significant enhancements for various methods, as demonstrated in Figure 4e and Figure 4f. In comparison to algorithmic interventions, such as designing different algorithms (e.g., DRO, Imbalance methods), a data-centric approach can be more effective in addressing distribution shifts. For instance, research on feature collection and feature engineering methods may prove impactful. Another avenue for future work is developing methods that can optimally incorporate expensive samples from the target distribution.

# 5   Discussion

We explore the complexity of distribution shifts in real-world tabular datasets in depth. Using natural shifts from 5 real-world tabular datasets across different domains, we specify each shift pattern and evaluate 22 methods via experiments with over 86k trained models. Our benchmark WHYSHIFT encompasses various distribution shift patterns to evaluate the robustness of the methods. We propose a simple but effective algorithm to identify regions with large $Y|X$-shifts, and through a comprehensive case study, we demonstrate how a better understanding of distribution shifts facilitates algorithmic and data-based interventions. Our findings highlight the importance of future research to understand how and why distributions differ in real-world applications.

Our study leaves many open directions for improvements in future work. Our benchmark only includes tabular datasets from the economic and transportation domain. Considering datasets from other domains such as the medicine or those involving feature embeddings may highlight different types of distribution shift. On the algorithmic side, our region-identification algorithm requires some target data to identify risky regions and cannot be used in cases where the target distribution is completely unknown. Furthermore, targeted data collection on regions of $Y|X$-shifts may be pose ethical and privacy concerns for marginalized groups. We provide more discussion in Appendix B.

## Acknowledgement

We thank Tiffany Cai for her help with implementing the DISDE method on our benchmarks. Peng Cui was supported in part by National Key R&D Program of China (No. 2018AAA0102004), National Natural Science Foundation of China (No. U1936219, 62141607). Hongseok Namkoong was partially supported by the Amazon Research Award.

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

# Appendices

## A  Relevant Work

### A.1  On Distribution Shifts

**Distribution Shift Benchmarks.**  Existing distribution shift benchmarks primarily concentrate on image and language datasets [74, 47, 107, 81, 101] to assess the robustness and efficiency of algorithms in real applications.

We briefly review benchmarks that address distribution shifts across various data types, including image and language data. For *image data*, several datasets capture natural distribution shifts, such as spatial and temporal variations. `PACS` [54] and `Office-Home` [91] categorize environments to be image styles. `VLCS` [29] and `iWildCam` [10] set their primary environments as data sources. `DomainNet` [108] is built on PACS and provides a more extensive selection with additional domains and categories. In recent developments, Koh et al. [47] collect several datasets to establish `WILDS`, setting a new structure for OOD generalization. Similarly, Yao et al. [100] introduced `Wild-Time`, highlighting temporal distribution shifts across diverse real-world scenarios. For *language* data, `CivilComments` [14] and `Amazon` [66] consist of individual comments collected from different users and distinctive groups (e.g. male and female). `GLUE-X` [97] provides a unified benchmark for evaluating OOD robustness in NLP models. For *other types* of data, `OGB-MolPCBA` [75] collects

molecular graphs in over 100,000 scaffolds and formulates a molecular property prediction task across different scaffolds. Towards auto-engineering, `Py150` [73] contains codes from 8,421 git repositories for code completion generalization.

However, these datasets/benchmarks do not specify or investigate in-depth the distribution shift patterns, and there is a noticeable lack of benchmark papers that specifically address real-world tabular datasets. These tabular datasets often present distinct patterns compared to image/language datasets, including prevalent methods [34] and shift patterns.

Therefore, it is important to understand detailed distribution patterns among these datasets to develop corresponding methodologies to address specific shifts, which is also observed in a recent benchmark [101]. Yang et al. [99] recently introduce one benchmark characterizing different patterns of subpopulation shift. This benchmark focuses on the relationship between a specific attribute and covariates, such as spurious correlation, which can be challenging to identify in tabular datasets. Besides, they focus on changes in the subpopulation, which is a subset of distribution shift patterns. For example, individuals in different states may suffer from little subpopulation shifts but still incur a large distribution shift. Individuals' demographic features are similar but the income level differs greatly between CA and PR in `ACS Income`, as demonstrated in Table 2. Besides, Kulinski and Inouye [52] use optimal transport to explain the shift between two distributions recently. But their method could not explicitly decompose the performance degradation as in our work. Motivated by the challenges above, our work hopes to fill in the gap by demonstrating one benchmark on tabular datasets with detailed analysis.

**Table 2:** Descriptive Characteristics under different states in `ACS income` dataset

| State | CA | MA | NY | NE | MT | SD | PR |
|---|---|---|---|---|---|---|---|
| Total Size | 195665 | 40114 | 103021 | 10785 | 5463 | 4899 | 9071 |
| White Man | 0.3311 | 0.4161 | 0.3687 | 0.4822 | 0.4959 | 0.4691 | 0.3526 |
| Nonwhite Man | 0.1969 | 0.0878 | 0.1379 | 0.0454 | 0.0428 | 0.0492 | 0.1818 |
| White Woman | 0.2873 | 0.4066 | 0.3464 | 0.4318 | 0.4221 | 0.4297 | 0.3124 |
| NonWhite Woman | 0.1847 | 0.0895 | 0.1471 | 0.0404 | 0.0392 | 0.0521 | 0.1532 |
| $P(Y = 0)$ | 0.589 | 0.532 | 0.585 | 0.688 | 0.707 | 0.730 | 0.894 |

**Choice of Domains in Distribution Shifts**   Although it is popular in the fairness literature to set each demographic subgroup in the adult and ACS dataset as one domain [104, 22, 25], the *relative regret* of models trained on these domains is quite small shown in Figure 1 empirically. This implicitly shows that $Y|X$-shifts are not strong across different demographic subgroups. In contrast, the model usually experiences relatively large $Y|X$-shifts under different spatial domains, which corresponds to the right-hand side of Figure 1. Besides, from the practical perspective, machine learning models often need to be deployed in spatialtemporal domains (by city, state, and country-wise; by year) only with few observed data while the model is trained with abundant data from the source domain. This is why we mainly choose spatiotemporal domains to benchmark distribution shifts in our case.

## A.2   Connection between Non-algorithmic Interventions and Algorithmic Interventions

Here we discuss the connections between our non-algorithmic interventions and several branches of methods addressing the distribution shifts, including active learning, imitation learning, causal learning, feature sensitivity analysis, and feature importance analysis.

**Active Learning.**   Active learning aims to improve model performance by acquiring a limited number of labels from the target distribution. See the survey [84] for a detailed reference. The challenge here is to quantify the value of unlabeled data so that we can select samples better. The selection criterion of existing approaches includes estimated variance [20], influence on the model performance [59], and Shapley value [32, 33, 44], or source-target distance metrics through the importance weighting [18, 61]. Some work also proposed running a regression problem for the query procedure based on learning strategies from the existing dataset [48]. Specifically, the target distribution where we query data and aim to evaluate the model subsequently differs from the source

distribution. Some work also develops active learning methods on domain adaptation to select additional samples from the target distribution [88, 87]. However, these works usually assume some restricted distribution structures between the source and target distribution such as only $X$-shifts occur, which may not hold in the tabular data. In contrast, we assume we have a few labeled target data but impose no restrictions on the two distributions. And we sample data in specific regions after identifying regions where we experience the largest $Y|X$-shifts.

**Imitation Learning**   Imitation learning aims to mimic the behavior of the expert with some off-policy data. This usually occurs in the reinforcement learning setup where a learner aims to learn the best action given each state [41]. Here, distribution shifts occur due to the mismatch of the state coverage between the observed data (source domain) and the environment to deploy (target domain). This raises the need for the importance sampling method to match the two distributions in imitation learning [92, 94]. In fact, the underlying conditional distribution does not change only with covariate (state) shifts in the imitation learning setup. Therefore, thiis does not fit into our case where the conditional distribution between the source and target domains differ.

**Causal Learning**   Causal learning methods receive much attention in the field of machine learning. The core idea of causal learning [72, 8] is to learn causally invariant relationships across multiple pre-defined training environments. Arjovsky et al. [8] propose Invariant Risk Minimization (IRM) to learn invariant representations across environments, and follow-up works [17, 49, 5, 50] propose variations with similar invariant regularizations. However, the effectiveness of these methods are challenged both theoretically and empirically. Theoretically, Rosenfeld et al. [78] illustrate that IRM could fail in a nonlinear context, and Liu et al. [58] demonstrate that the learned invariance property largely depends on the quality of pre-defined environments. Empirically, Gulrajani and Lopez-Paz [35] show that when carefully implemented and tuned, ERM still outperforms the state-of-the-art methods in terms of average performance.

Compared with the simple non-algorithmic interventions in this work, causal learning methods rely heavily on the invariance assumptions and have strict requirements on the quality of multi-environment data. This restricts their applicability in practice, since modern datasets are often collected without explicit environment labels, and in many scenarios, it is quite hard to pre-define meaningful environment labels. Our proposed non-algorithmic interventions (collecting features and data) do not rely on the invariance assumption, and it could serve as a "solution" when observing performance degradation, which helps to analyze the model failure and to direct further improvements. And we hope that these simple data-centric interventions could inspire future research in this direction to mitigate the effects of distribution shifts.

Since we are also considering other missing features to improve the performance under distribution shifts, there are some works providing insights on the feature / region importance to the final output.

**Feature Analysis: Sensitivity and Importance.**   There are two streams of literature measuring the relationship between input features and the response variable. These are feature sensitivity analysis and feature importance. Feature sensitivity analysis aims to quantify the sensitivity of the performance metrics to each input. This helps understand the impact of variations in input features on the output of a simulation system. Classical metrics include ANOVA, Sobol indices [68] and Morris methods [43]. Meanwhile, feature importance aims to understand the performance decomposition of different algorithms. Shapley value-based approaches gain the most popularity in understanding the attribution of predictions to each input feature [60, 21]. Some work leverages these ideas to understand the difference in distribution shifts to each existing input feature in the data [15, 106]. They decompose the shift on the joint distribution to particular $X_i$-shifts or the condition $X_j|X_i$-shifts ($X_i$ and $X_j$ denote different features) under a known causal graph. In contrast, we investigate the additional missing feature beyond existing datasets and aim to reduce the performance drop under distribution shifts in our paper. Specifically, we focus on the local regions where the distribution incurs the largest shifts and add the feature ENG in our main body due to our prior knowledge that "ENG" feature would yield the largest difference in that subpopulation between two distributions. Our region-based

approach, in fact, can be further rigorously extended to investigate the marginal distribution of what features would yield the largest difference in that region between two distributions. We hope this can inspire researchers to apply refined tools in this line such as Shapley value to help understand and mitigate distribution shifts.

**Region Analysis: Attribution of Distribution Shifts.**   Similar to our algorithmic goal in Section 3.1 of identifying regions where model learners are different, Oberst *et al.* [67] and Lim *et al.* [56] developed specific methods to identify specific covariate regions where model learners are different. The difference between their setup and ours lies in the sampling and assumption of observed data. In their approach, the observed data is sampled i.i.d. across various prediction models, without any distribution shift. Besides, they can observe only one selected prediction result per sample from all the model learners. However, in our case study, the models are built on datasets from two domains that experience distribution shifts. As a result, we can further isolate the model difference based on the shared input space $X$ and differentiate it from the total difference, specifically focusing on the $Y|X$-shifts.

# B   Discussion on Limitations

We discuss some limitations of our work.

**For Benchmark**   First, we only consider the source-target transfer pairs from datasets including the economic and transportation domain, and we leave the detailed pattern evaluation of these datasets in other domains such as the medical area (e.g. `MIMIC-III` [45]) as an interesting direction of the future work. Besides, we only consider the source and target from one fixed domain (i.e. one state or city). In practice, it is reasonable to extend this benchmark to consider the source and target distribution with multiple domains of varying proportions. Our results of characterizing the distribution shift patterns highlight the importance of utilizing other refined tools. These tools can help us understand the difference between real-world distribution shifts and enable further investigation and analysis.

**For Algorithm 1**   Our Algorithm 1 is proposed to identify risky regions with large $Y|X$-shifts when observing severe model degradation between the source target distributions. Therefore, one limitation is that it requires target data to identify the risky regions with large $Y|X$-shifts. This algorithm cannot evaluate the generalization performance when the target distribution is completely unknown.

Furthermore, when conducting non-algorithmic interventions based on the risky regions learned by Algorithm 1, researchers should be careful and *incorporate more background knowledge* to find the proper way. For example, in Section 3.2, we analyze the risky regions and find that the "ENG" feature may be important to mitigate the $Y|X$-shifts. This is from our prior knowledge that the official language between the two states is different. Also, when Algorithm 1 is misused (e.g., the learned risky regions are used without destination or not being checked carefully), it might harm some vulnerable groups.

**For Data-Collection**   In Section 3.2, we propose two simple non-algorithmic interventions to mitigate the $Y|X$-shifts, one of which is to efficiently collect target data from the risky region. It achieves much better results combined with several typical methods on our benchmark. However, in practice, we acknowledge that this technique could only be used when we can obtain data from the target distribution and the data collection procedure does not raise any privacy concerns and predatory inclusion.

**Applicability Across Different Data Modalities**   In this work, we focus on real-world distribution shifts in tabular data settings. We propose Algorithm 1 to find the risk regions and come up with some simple data-based interventions. Therefore, we do not investigate in-depth the applicability of the proposed methods across other data types. However, we argue that our proposed Algorithm 1 could generalize to complicated data types (e.g., image data) with corresponding deep models.

Specifically, both the domain classifier (to estimate $\mathbb{P}(X \text{ from } Q_X | X)$) and region learner $h(X)$ should be replaced by deep neural networks. And our data-based interventions also have the potential to incorporate with more complicated data types. We leave this investigation to future work.

## C   Case Study Details

In this section, we provide more details about our case study in the main body.

### C.1   DISDE to $X$-shifts and $Y|X$-shifts

When facing performance degradation under distribution shifts, one direct idea is to figure out the reasons why the performance drop. To this end, Cai et al. [16] propose DIstribution Shift DEcomposition (DISDE) to attribute the total performance degradation to $Y|X$-shifts and $X$-shifts. Specifically, given samples $(X, Y)$ from distributions $P$ and $Q$, to quantify the discrepancy between $P_{Y|X}$ and $Q_{Y|X}$, they first control the marginal distribution on $\mathcal{X}$ by introducing the shared distribution $S_X$. From that, we can estimate the performance degradation caused by $Y|X$-shifts and that caused by $X$-shifts could also be estimated by comparing $S_X$ with $P_X$ and $Q_X$, respectively. Note that DISDE could be used in image datasets (see Section 4.2 in [16]). The official code for DISDE could be found at `https://github.com/namkoong-lab/disde`. We specify the formula of DISDE as follows:

$$
\begin{aligned}
\mathbb{E}_Q[\ell(f_P(X), Y)] - \mathbb{E}_P[\ell(f_P(X), Y)] = \quad & \mathbb{E}_{S_X}[R_P(X)] - \mathbb{E}_P[R_P(X)] && \text{(I)} \\
& + \mathbb{E}_{S_X}[R_Q(X) - R_P(X)] && \text{(II)} \\
& + \mathbb{E}_Q[R_Q(X)] - \mathbb{E}_{S_X}[R_Q(X)], && \text{(III)}
\end{aligned}
$$

where $R_\mu(x) := \mathbb{E}_\mu[\ell(f_P(X), Y)|X = x]$ for $\mu = P, Q$ is denoted as the conditional risks on $P$ and $Q$. $S_X$ is the share distribution with support contained in both $P_X$ and $Q_X$. *Then we see the sum of the two terms (I) and (III) as the performance drop attributed to $X$-shifts and the term (II) as the performance drop attributed to $Y|X$-shifts.* Besides, we also implement DISDE in our released package named WHYSHIFT, which could be found at `https://github.com/namkoong-lab/whyshift`.

### C.2   Prevalence of $Y|X$-Shifts

For each of the 7 settings in Table 1, we only select one target distribution in the main body, while our benchmark supports multiple target distributions. Specifically, for `ACS Income`, `ACS Mobility`, `ACS Pub.Cov`, we have 50 target distributions (the other 50 American states) in total for each setting; for `Taxi`, we have 3 target distributions (other cities Mexico City, Bogotá, Quitio); for the temporal shift version of `ACS Pub.Cov` (i.e. setting 6), we have 3 target distributions (i.e. the year 2014, 2017, and 2021); and for `US Accident`, we have 13 target distributions (13 American states). Therefore, we have 169 source-target transfer pairs in total for these 7 settings.

We use XGBoost classifier and calculate the decomposition of performance degradation via DISDE [16]. We calculate the $Y|X$-**shift ratio** from the source distribution $P$ to the target distribution $Q$:

$$
Y|X\text{-shift ratio} = \frac{\mathbb{E}_{S_X}[R_Q(X) - R_P(X)]}{\mathbb{E}_Q[\ell(f_P(X), Y)] - \mathbb{E}_P[\ell(f_P(X), Y)]},
$$

where $R_\mu(X)$ is defined in Appendix C.1. We first focus on pairs with relatively strong distribution shifts (i.e. performance degradation larger than 8 percentage points). Across these pairs, we find that **70.2%** pairs have over **60%** $Y|X$-shifts, and **87.2%** pairs have over **50%** $Y|X$-shifts, indicating the prevalence of $Y|X$ shifts. To visualize the result better, in all 22 settings in our benchmark (as shown in Table 3), we focus on transfer pairs with performance degradation larger than 5 percentage points and plot the histogram of the ratio of $Y|X$-shifts in Figure 8. From Figure 8, we could see that $Y|X$-shifts are prevalent in real-world distribution shifts.

### C.3   Details of Algorithm 1

In this section, we provide a more detailed introduction of our proposed Algorithm 1. We propose a simple yet efficient method to identify data regions with strong $Y|X$ shifts, and it could inspire operational and modeling interventions as shown in Section 3.2.

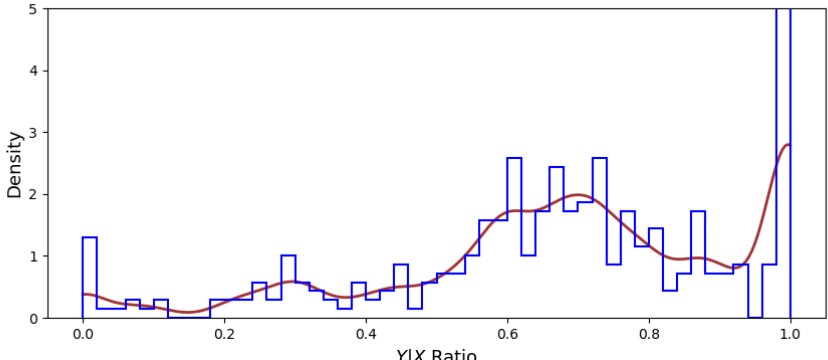

**Figure 8:** Histogram of $Y|X$-shift ratio in our benchmark.

Consider a model $f : \mathcal{X} \to \mathcal{Y}$ that predicts outcome $Y$ from covariates $X$. Let $\ell(f(x), y)$ be a loss function denoting a notion of predictive error (e.g., cross-entropy loss, mean-square error). Our goal is to identify a region $\mathcal{S} \subseteq \mathcal{X}$ where $P(Y|X)$ differs a lot from $Q(Y|X)$. In order to directly compare $P(Y|X)$ and $Q(Y|X)$, we must do an apples-to-apples comparison: we cannot know $Q(Y|X)$ for $X$'s that are primarily observed in $P$ (and vice versa).

To address this, we first construct a shared distribution $S_X$ over $X$ whose support is contained in that of both $P_X$ and $Q_X$. We choose a specific *shared distribution* $S_X$ over $X$ whose support is contained in that of $P_X$ and $Q_X$ (following [16]). Ideally, the chosen shared distribution would exhibit a higher density when both $P_X$ and $Q_X$ densities are high, and a lower density when either of the two possesses a low density. This strategy effectively allows regions of shared density to be more pronounced. Recall that $p_X, q_X, s_X$ are the densities of $X$ under $P, Q$ and $S$, we formulate $s_X$ as

$$s_X(x) \propto p_X(x)q_X(x)/(p_X(x) + q_X(x)). \tag{C.1}$$

**Choices of $S_X$** Here we would like to discuss different choices of $S_X$ to provide more intuitions. As demonstrated in [16], we can use many forms of the shared distribution. For example, we could choose the following form:

$$s_X(x) \propto \min\{p_X(x), q_X(x)\}, \tag{C.2}$$

which guarantees that the support of $S_X$ is contained in that of both $P_X$ and $Q_X$. Another choice is:

$$s_X(x) \propto \begin{cases} p_X(x) + q_X(x) & \text{if } \min\{\frac{p_X(x)}{q_X(x)}, \frac{q_X(x)}{p_X(x)}\} \geq \epsilon, \\ 0 & \text{otherwise,} \end{cases} \tag{C.3}$$

for some $\epsilon \geq 0$. This form of $S_X$ defines shared samples as those with high likelihood ratios. Notably, for all the three forms of $S_X$, if $P_X = Q_X$, then $S_X = P_X = Q_X$. And when $p_X(x) \gg q_X(x)$ or $p_X(x) \ll q_X(x)$, Equation (C.1) and Equation (C.2) become similar. In our Algorithm 1, we use the form of Equation (C.1), and Cai et al. [16] observe that in practice the qualitative conclusions are not very sensitive to the specific choice of shared distribution.

**Intuitions behind Algorithm 1** Since we do not have access to samples from the shared distribution $S_X$, we reweight samples from $P_X$ and $Q_X$ using the likelihood ratios:

$$\frac{s_X}{p_X}(x) \propto \frac{q_X(x)}{p_X(x) + q_X(x)} \quad \text{and} \quad \frac{s_X}{q_X}(x) \propto \frac{p_X(x)}{p_X(x) + q_X(x)}. \tag{C.4}$$

Then we define $\hat{\alpha}$ as the proportion of the pooled data that comes from distribution $Q$:

$$\hat{\alpha} = \frac{n_Q}{n_P + n_Q} \quad \text{and} \quad \hat{\pi}(x) = \mathbb{P}(\tilde{X} \text{ from } Q_X | \tilde{X} = x), \tag{C.5}$$

where $\hat{\pi}(x)$ denotes the probability of a sample to come from $Q_X$. Using Bayes' rule, we have:

$$\hat{\pi}(x) = \frac{\mathbb{P}(\tilde{X} = x | \tilde{X} \text{ from } Q_X)\mathbb{P}(\tilde{X} \text{ from } Q_X)}{\mathbb{P}(x)} = \frac{\hat{\alpha}q(x)}{\hat{\alpha}q(x) + (1-\hat{\alpha})p(x)},$$

$$= \frac{\hat{\alpha}}{\hat{\alpha} + (1-\hat{\alpha})\frac{p(x)}{q(x)}}. \tag{C.6}$$

Noting that the ratio $\hat{\pi}(x)$ can be modeled as the probability that an input $x$ came from $P_X$ vs $Q_X$, we train a binary "domain" classifier to estimate the ratios. (The "domain" classifier can be any black-box method, and we use XGBoost throughout.)

Then the likelihood ratios that we care about could be reformulated as:

$$\frac{s_X}{p_X}(x) \propto \frac{1}{\frac{p_X(x)}{q_X(x)} + 1} \quad \text{and} \quad \frac{s_X}{q_X}(x) \propto \frac{\frac{p_X(x)}{q_X(x)}}{\frac{p_X(x)}{q_X(x)} + 1}, \tag{C.7}$$

which gives that:

$$\frac{s_X}{p_X}(x) \propto \frac{\hat{\pi}(x)}{(1-\hat{\alpha})\hat{\pi}(x) + \hat{\alpha}(1-\hat{\pi}(x))} \quad \text{and} \quad \frac{s_X}{q_X}(x) \propto \frac{1 - \hat{\pi}(x)}{(1-\hat{\alpha})\hat{\pi}(x) + \hat{\alpha}(1-\hat{\pi}(x))}. \tag{C.8}$$

After obtaining the likelihood ratios $\frac{s_X}{p_X}(x)$ and $\frac{s_X}{q_X}(x)$, we could do an apples-to-apples comparison: we estimate $P_{Y|X}$ and $Q_{Y|X}$ over the shared distribution $S_X$ (using XGBoost $\mathcal{F}$)

$$f_P := \arg\min_{f \in \mathcal{F}} \left\{ \mathbb{E}_{S_X}\left[ \mathbb{E}_P[\ell(f(X), Y)|X] \right] = \mathbb{E}_P\left[ \ell(f(X), Y)\frac{dS_X}{dP_X}(X) \right] \right\}, \tag{C.9}$$

$$f_Q := \arg\min_{f \in \mathcal{F}} \left\{ \mathbb{E}_{S_X}\left[ \mathbb{E}_Q[\ell(f(X), Y)|X] \right] = \mathbb{E}_Q\left[ \ell(f(X), Y)\frac{dS_X}{dQ_X}(X) \right] \right\}. \tag{C.10}$$

Then, for any threshold $b \in [0, 1]$, $\{x \in \mathcal{X} : |f_P(x) - f_Q(x)| \geq b\}$ suggests a region that may suffer model performance degradation due to $Y|X$-shifts.

## C.4 Analysis of Decision Tree

In Algorithm 1, we use a shallow *decision tree* $h(x)$ to approximate $y = |f_P(x) - f_Q(x)|$ on the shared distribution $S_X$ to find the covariate region with highest discrepancy. In our decision tree, we use the *squared error* as the splitting criterion. And below we demonstrate that this criterion is equivalent to maximizing the discrepancy between two children nodes.

Suppose there are $N$ samples with outcomes $\{y_i\}_{i \in [N]}$ belonging to tree node $fa$, and these samples are split into two children nodes $s_1, s_2$, where the node $s_1, s_2$ denote the set of sample indices in the two children nodes respectively. The squared error criterion to split $fa$ into $s_1$ and $s_2$ is:

$$\min_{s_1, s_2} \left\{ \mathcal{L}(s_1, s_2) := \frac{1}{N}\left( \sum_{i \in s_1}(y_i - \mu_{Y,1})^2 + \sum_{i \in s_2}(y_i - \mu_{Y,2})^2 \right) \right\}, \tag{C.11}$$

where

$$\mu_{Y,1} := \frac{\sum_{i=1}^N y_i \mathbf{1}_{\{i \in s_1\}}}{\sum_{i=1}^N \mathbf{1}_{\{i \in s_1\}}}, \quad \mu_{Y,2} := \frac{\sum_{i=1}^N y_i \mathbf{1}_{\{i \in s_2\}}}{\sum_{i=1}^N \mathbf{1}_{\{i \in s_2\}}} \tag{C.12}$$

denote the mean values of the outcome $Y$ with samples in children nodes $s_1$ and $s_2$. Denote the distribution of the outcome $Y$ follows the empirical distribution over the $N$ samples $\{y_i\}_{i \in [N]}$. Simplifying (C.11), we have:

$$\mathcal{L}(s_1, s_2) = P(Y \in s_1)\text{Var}_{s_1}(Y) + P(Y \in s_2)\text{Var}_{s_2}(Y) = \mathbb{E}_S[\text{Var}(Y|S)], \tag{C.13}$$

where $\text{Var}_s(Y)$ denotes the variance of the outcome variable $Y$ in node $s$, $S = \{s_1, s_2\}$ is the variable representing the children nodes. Therefore, given that $\text{Var}_{fa}(Y)(:= \text{Var}_S(\mathbb{E}[Y|S]) + \mathbb{E}_S[\text{Var}(Y|S)])$ is constant, the minimal $\mathbb{E}_S[\text{Var}(Y|S)]$ corresponds with the largest $\text{Var}_S(\mathbb{E}[Y|S])$, which maximizes the discrepancy of the outcome between two children nodes.

## C.5 Details of Non-algorithmic Interventions

In Section 3.2, we propose two potential non-algorithmic interventions to mitigate the performance degradation. In this section, we introduce in detail the intervention of collecting specific data from the target.

**Experiment Setup**   We focus on the income prediction task using the `ACS Income` dataset. Consider a practical scenario where the training set consists of 20,000 samples from California (CA) and the trained model was deployed in Puerto Rico (PR) in trial. After the trial deployment in PR, we got a *small amount* of samples from PR with labels and observed performance degradation. Under this setting, we investigate the effect of non-algorithmic interventions.

**Collect specific data from the target**   We first identify the regions with high discrepancy between source and target. Note that the sample size of the target state is small compared to the training samples. Then for typical algorithms like logistic regression (LR), MLP, random forest (RF), LightGBM and XGBoost, we compare the performances of:

- original setting (only 20,000 samples from the source);
- original setting with $N$ additional random samples drawn from the whole target state;
- original setting with $N$ additional random samples drawn from the *risk region* of the target state.

In this experiment, we first select the best configuration of each method according to the $i.i.d$ validation set in the original setting (only samples from CA), and fix it for the other two interventions. We vary $N$ as 100, 200, 300 and the results are shown in Figure 9. From Figure 9, incorporating data from the risk region leads to a *stable improvement* on typical algorithms even for small target sample sizes. However, we observe that LightGBM and XGBoost would easily overfit $f_Q$ on the target data, and we use random forest under this setup as an alternative. It is worth investigating approaches to find risk regions effectively under small/imbalanced sample sizes in the future. The approach mentioned here is a simple way of non-algorithmic and explainable interventions and we hope it could inspire further research in this direction.

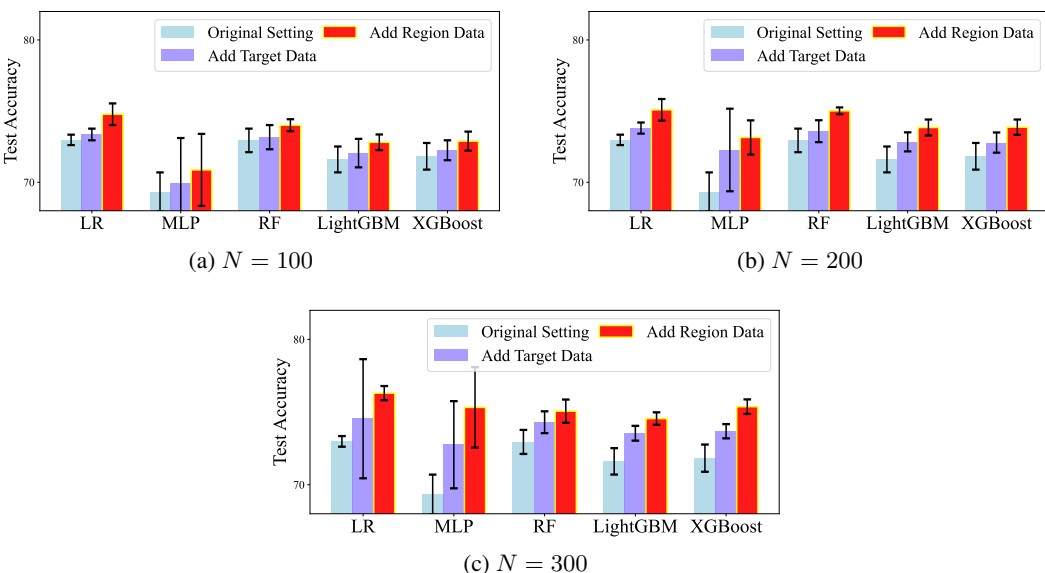

**Figure 9:** Test accuracies of different ways to incorporate data.

## C.6 Potential Alternative Approach of Identifying Risk Region

In Algorithm 1, we propose a simple way to identify the risk region to explain the cause of $Y|X$-shifts. And the identified region could be used to guide the collecting process of target data, which could help to further reduce the effects of performance degradation. However, in practice, when the amount of target samples is quite small, it may be hard to train the $f_Q$ only with target samples accurately.

Therefore, it may be better to propose a method that does not need $f_Q$ to fit $Q(Y|X)$ on the target distribution.

Following this idea, we propose an alternative way of Algorithm 1. Since the training data is enough, it is feasible to fit $f_P$ on the shared distribution $S_X$ as:

$$f_P := \underset{f \in \mathcal{F}}{\arg\min} \, \mathbb{E}_P \left[ \frac{dS_X}{dP_X} \ell(f(X), Y) \right]. \tag{C.14}$$

Then for $n_Q$ samples from target distribution $Q$, we could use a prediction model $h(x)$ to approximate $|Y - f_P(X)|$ on the shared distribution (by reweighting density ratio $\frac{dS_X}{dQ_X}$). Note that this method does not need to train $f_Q$ on target samples, but the quality of the density ratio $dS_X/dP_X$ and the prediction model $h(x)$ still depend on the target samples. We hope the estimation of the ratio and the residuals $|Y - f_P(X)|$ can be less affected by the low sample size $n_Q$. This non-algorithmic intervention is not the main focus of this work, but we hope this idea could help to promote this line of research.

# D    Benchmark Details

In this section, we provide the details of our benchmark. In our benchmark, we explore distribution shifts on 5 real-world tabular datasets from the economic and traffic sectors with natural spatiotemporal distribution shifts. We carefully design 22 settings utilizing these 5 datasets, and the overview of 22 settings is shown in Table 3. Note that in our main body, due to space limitations, we only pick 7 typical settings and select only one representative target domain for each setting, as shown in Table 1. Detailed introductions of all datasets, algorithms, and experiment settings will be given in the following sections. A Python package for our benchmark can be found at https://github.com/namkoong-lab/whyshift.

**Table 3.** Overview of datasets and the whole 22 selected settings. In our benchmark, each setting has multiple target domains (except the last setting). In our main body, we select only one target domain for each setting. ◇: we report the "Dom. Ratio" to represent the dominant ratio of $Y|X$-shifts or $X$-shifts in source-target pairs with performance degradation larger than **5** percentage points in each setting. For example, "$Y|X$: 13/14" means that there are 14 source-target pairs in Setting 1 with degradation larger than 5 percentage points and 13 out of them with over 50% degradation attributed to $Y|X$ shifts. We use XGBoost to measure this.

| #ID | Dataset | Type | #Features | Outcome | Source | #Train Samples | #Test Domains | Dom. Ratio◇ |
|-----|---------|------|-----------|---------|--------|---------------|---------------|-------------|
| 1 | ACS Income | Spatial | 9 | Income≥50k | California | 195,665 | 50 | $Y|X$: 13/14 |
| 2 | ACS Income | Spatial | 9 | Income≥50k | Connecticut | 19,785 | 50 | $Y|X$: 24/24 |
| 3 | ACS Income | Spatial | 9 | Income≥50k | Massachusetts | 40,114 | 50 | $Y|X$: 21/22 |
| 4 | ACS Income | Spatial | 9 | Income≥50k | South Dakota | 4,899 | 50 | $Y|X$: 9/9 |
| 5 | ACS Mobility | Spatial | 21 | Residential Address | Mississippi | 5,318 | 50 | $Y|X$: 28/34 |
| 6 | ACS Mobility | Spatial | 21 | Residential Address | New York | 40,463 | 50 | $Y|X$: 30/31 |
| 7 | ACS Mobility | Spatial | 21 | Residential Address | California | 80,329 | 50 | $Y|X$: 9/17 |
| 8 | ACS Mobility | Spatial | 21 | Residential Address | Pennsylvania | 23,918 | 50 | $Y|X$: 17/17 |
| 9 | Taxi | Spatial | 7 | Duration time≥30 min | Bogotá | 3,063 | 3 | $Y|X$: 1/2 |
| 10 | Taxi | Spatial | 7 | Duration time≥30 min | New York City | 1,458,646 | 3 | $Y|X$: 3/3 |
| 11 | ACS Pub.Cov | Spatial | 18 | Public Ins. Coverage | Nebraska | 6,332 | 50 | $Y|X$: 32/39 |
| 12 | ACS Pub.Cov | Spatial | 18 | Public Ins. Coverage | Florida | 71,297 | 50 | $Y|X$: 28/29 |
| 13 | ACS Pub.Cov | Spatial | 18 | Public Ins. Coverage | Texas | 98,928 | 50 | $Y|X$: 33/34 |
| 14 | ACS Pub.Cov | Spatial | 18 | Public Ins. Coverage | Indiana | 24,330 | 50 | $Y|X$: 11/13 |
| 15 | US Accident | Spatial | 47 | Severity of Accident | Texas | 26,664 | 13 | $Y|X$: 7/7 |
| 16 | US Accident | Spatial | 47 | Severity of Accident | California | 64,909 | 13 | $X$: 22/31 |
| 17 | US Accident | Spatial | 47 | Severity of Accident | Florida | 32,278 | 13 | $X$: 5/7 |
| 18 | US Accident | Spatial | 47 | Severity of Accident | Minnesota | 8,927 | 13 | $X$: 8/11 |
| 19 | ACS Pub.Cov | Temporal | 18 | Public Ins. Coverage | Year 2010 (NY) | 73,208 | 3 | $X$: 2/2 |
| 20 | ACS Pub.Cov | Temporal | 18 | Public Ins. Coverage | Year 2010 (CA) | 149,441 | 3 | $X$: 2/2 |
| 21 | ACS Income | Synthetic | 9 | Income≥50k | Younger People (80%) | 20,000 | 1 | $X$: 1/1 |
| 22 | ACS Income | Synthetic | 9 | Income≥50k | Younger People (90%) | 20,000 | 1 | $X$: 1/1 |

## D.1    Datasets & Settings

Here we introduce 5 real-world tabular datasets in detail. ACS Income, ACS Mobility, ACS Public Coverage are based on the American Community Survey (ACS) Public Use Microdata Sample (PUMS) [25]. And here we use the same data filtering as [25].

**ACS Income**    The task is to predict whether an individual's income is above $50,000. We filter the dataset to only include individuals above the age of 16, usual working hours of at least 1 hour per week in the past year, and an income of at least $100.

- In our Setting 1∼4, we use data from 2018 and the source/target domain refers to American states, which contain natural spatial distribution shifts. We train methods on data from California, Connecticut, Massachusetts, and South Dakota respectively, and test the generalization performance on the other 50 American states.

- In our Setting 21∼22, we sub-sample the dataset according to age to introduce covariate shift, where we focus on individuals from California and form two groups according to whether their age is $\geq 25$. In Setting 21, the source data over-samples the low age group where 80% is drawn from the age $\geq 25$ group, and the proportions are reversed in the target data (20% from age $\geq 25$ group). In Setting 22, the source data over-samples the low age group where 90% is drawn from the age $\geq 25$ group, and the proportions are reversed in the target data (10% from age $\geq 25$ group).

**ACS Mobility**    The task is to predict whether an individual had the same residential address one year ago. We filter the dataset to only include individuals between the ages of 18 and 35, which increases the difficulty of the prediction task.

- In our Setting 5∼8, we use data from 2018 and the source/target domain refers to American states, which contain natural spatial distribution shifts. We train methods on data from Mississippi, New York, California, and Pennsylvania respectively, and test the generalization performance on the other 50 American states.

**ACS Public Coverage**    The task is to predict whether an individual has public health insurance. We focus on low-income individuals who are not eligible for Medicare by filtering the dataset to only include individuals under the age of 65 and with an income of less than $30,000.

- In our Setting 11∼14, we use data from 2018 and the source/target domain refers to American states, which contain natural spatial distribution shifts. We train methods on data from Nebraska, Florida, Texas, and Indiana respectively, and test the generalization performance on the other 50 American states.

- In our Setting 19∼20, we consider the temporal shifts. We use data from 2010 in training and data from 2014, 2017, and 2021 in testing (3 test domains). In Setting 19, the training data come from New York, and in Setting 20, the training data come from California.

**US Accident**    The task is to predict whether an accident is severe (long delay) or not (short delay) based on weather features and Road conditions features.

- In our Setting 15∼18, the source/target domain refers to American states, which contain natural spatial distribution shifts. We train methods on data from California, Florida, Texas, and Minnesota respectively, and test the generalization performance on other 13 American states. Here we only involve 13 test domains because the sample sizes in the other states are quite small.

**Taxi**    The task is to predict whether the total ride duration time exceeds 30 minutes, based on location and temporal features. We filter the data in 2017 and remove some extremely large or small features (e.g. samples with too long distances which can be easily classified).

- In our Setting 9∼10, we use data from 2016 and the source/target domain refers to different cities. We train methods on data from Bogota and New York City and test the generalization performance in the other cities.

### D.2    Algorithms & Implementations

In our benchmark, we evaluate 22 algorithms that span a wide range of learning strategies on tabular data and compare their performances under different patterns of distribution shifts we construct. Concretely, these algorithms include:

**Base Methods**   We include some typical supervised learning methods for tabular data: Logistic Regression (LR), SVM, and fully-connected neural networks (MLP) with standard ERM optimization. For LR and SVM, we use the standard implementation in `scikit-learn` [71] and train them on CPUs. For MLP, we implement it via `PyTorch` [70] and train it on GPUs.

**Tree Ensemble Models**   As shown by Gardner *et al.* [31], several tree-based methods achieve good performances on tabular datasets. And gradient-boosted trees (e.g., XGBoost, LightGBM) are widely considered as the state-of-the-art methods on tabular data. Therefore, we evaluate XGBoost and LightGBM in our benchmark. Also, we evaluate Random Forest (RF) to incorporate the performance of tree bagging methods.

**DRO Methods**   Distributionally Robust Optimization (DRO) methods are proposed to address the distribution shifts, which is the form of:

$$\min_{\theta \in \Theta} \sup_{Q \in \mathcal{P}(P_{tr})} \mathbb{E}_Q[\ell(f_\theta(X), Y)], \tag{D.1}$$

where $\mathcal{P}(P_{tr})$ denotes the uncertainty set around the training distribution $P_{tr}$. Following Gardner *et al.* [31], we implement two typical variants of DRO, namely CVaR-DRO and $\chi^2$-DRO. CVaR-DRO is equivalent to Conditional Value at Risk (CVaR), and $\chi^2$-DRO uses $\chi^2$-divergence to regulate the uncertainty set. We use the fast implementation [53] in `PyTorch` [70]. We also consider the DORO [105], which discards a proportion $\epsilon$ of the largest error points in each iteration to mitigate the outliers in DRO with two variants, CVaR-DORO and $\chi^2$-DORO. We also evaluate the Group DRO [79], which is proposed to minimize the worst-group loss and shows good generalization performances on many vision tasks. This method needs the group label and we define groups according to the "SEX" feature on ACS datasets. For `US Accident` and `Taxi`, we do not run Group DRO, since the current Group DRO model in the codebase only accepts the input with few groups while it is hard to define such group here. For all DRO methods, we use the MLP as the backbone model. We do not choose tree ensemble methods since tree ensemble methods are difficult to adapt to the distributionally robust case. Therefore, we leave their method developments and implementations as our future work.

**Imbalanced Learning Methods**   Recently, some simple data balancing methods [42] have shown good worst-group performances under distribution shifts. In our benchmark, we implement 4 typical balancing methods, namely Sub-Sampling $Y$ (SUBY), Reweighting $Y$ (RWY), Sub-Sampling Group (SUBG), and Reweighting Group (RWG). Besides, "Just Train Twice" (JTT [57]) exhibits good performances in many vision tasks, and therefore we also evaluate it in our tabular settings. Furthermore, stable learning methods [51, 24] propose to de-correlate sample covariates for an accurate estimation of causal relationships via global balancing, which could mitigate distribution shifts. In our benchmark, we implement one typical method named DWR [51]. For these imbalanced learning methods, we use XGBoost as the backbone model due to its superiority on tabular data and adjust sample weight or training procedure accordingly for each of the methods.

**Fairness-enhancing Methods**   Following Ding *et al.* [25] and Gardner *et al.* [31], fairness-enhancing methods have the potential to mitigate the performance degradation under distribution shifts. In our benchmark, we evaluate the in-processing and post-processing intervention methods. The in-processing method [4] minimizes the prediction error subject to some fairness constraints, and in our benchmark, we choose three typical fairness constraints, including demographic parity (DP), equal opportunity (EO), error parity (EP). And the post-processing method [37] randomizes the predictions of a fixed classifier to satisfy equalized odds criterion, and we use exponential and threshold controls in our benchmark. We use the implementations of `aif360` [11] and `fairlearn` [12].

### D.3   Parameter Search Space

We provide the hyperparameter grids in Table 4. We mainly use the hyperparameter grids proposed in [31], and we restrict the grid size of each method in each setting to 200 in consideration of computational costs. For each setting, we randomly pick 200 configurations for each algorithm for a fair comparison. For methods incorporating backbone models (e.g., MLP/XGBoost), we choose the top 10 best configurations for that backbone model to reduce the search space, making the searched best configuration represent its best performance more accurately. Moreover, to accelerate the grid search process, we utilize `Ray` [55] to run experiments in parallel.

**Table 4.** Hyperparameter grids used in all experiments. ⋄: for methods with the total grid size above 200, we randomly sample *200* configurations for fair comparisons. For methods incorporating backbone models (e.g., MLP/XGBoost), we choose top-10 best configurations for that backbone model to reduce the search space, making the searched best configuration represent its best performance more accurately.

| Model | Total Grid Size | Hyperparameter | Value Range |
|---|---|---|---|
| **Base Methods** | | | |
| MLP | $270^{\diamond}$ | Learning Rate | $\{0.001, 0.003, 0.0001, 0.005, 0.01\}$ |
| | | Batch Size | $\{64, 128, 256\}$ |
| | | Hidden Units | $\{16, 32, 64\}$ |
| | | Dropout Ratio | $\{0, 0.1\}$ |
| | | Train Epoch | $\{50, 100, 200\}$ |
| SVM | 96 | C | $\{0.01, 0.1, 1, 10, 100, 1000\}$ |
| | | Kernel | $\{\text{linear}, \text{RBF}\}$ |
| | | Loss | Squared Hinge |
| | | $\gamma$ | $\{0.1, 0.3, 0.5, 1.0, 1.5, 2.0, \text{scale}, \text{auto}\}$ |
| Logistic Regression | 23 | $L_2$ penalty | $\{0.001, 0.03, 0.005, 0.007, 0.01, 0.03, 0.05, \ldots$ $1.3, 1.7, 5, 10, 50, 100, 5e2, 1e3, 5e3, 1e4\}$ |
| **Tree Ensemble Methods** | | | |
| Random Forest | $640^{\diamond}$ | Num. Estimators | $\{32, 64, 128, 256, 512\}$ |
| | | Max Features | $\{\text{sqrt}, \text{log2}\}$ |
| | | Min. Samples Split | $\{2, 4, 8, 16\}$ |
| | | Min. Samples Leaf | $\{1, 2, 4, 8\}$ |
| | | Cost-Complexity $\alpha$ | $\{0., 0.001, 0.01, 0.1\}$ |
| XGBoost | $1944^{\diamond}$ | Learning Rate | $\{0.1, 0.3, 1.0, 2.0\}$ |
| | | Min. Split Loss | $\{0, 0.1, 0.5\}$ |
| | | Max. Depth | $\{4, 6, 8\}$ |
| | | Column Subsample Ratio (tree) | $\{0.7, 0.9, 1\}$ |
| | | Column Subsample Ratio (level) | $\{0.7, 0.9, 1\}$ |
| | | Max. Bins | $\{128, 256, 512\}$ |
| | | Growth Policy | $\{\text{Depthwise}, \text{Loss Guide}\}$ |
| LightGBM | $1680^{\diamond}$ | Learning Rate | $\{0.01, 0.1, 0.5, 1.\}$ |
| | | Num. Estimators | $\{64, 128, 256, 512\}$ |
| | | $L_2$-reg. | $\{0., 0.001, 0.01, 0.1, 1.\}$ |
| | | Min. Child Samples | $\{1, 2, 4, 8, 16, 32, 64\}$ |
| | | Column Subsample Ratio (tree) | $\{0.5, 0.8, 1.\}$ |
| **DRO Methods** | | | |
| DRO $\chi^2$ | $1890^{\diamond}$ | Uncertainty set size $\alpha$ | $\{0.01, 0.1, 0.2, 0.3, 0.4, 0.5, 0.6\}$ |
| | | Backbone Model | MLP |
| DRO CVaR | $1890^{\diamond}$ | Uncertainty set size $\alpha$ | $\{0.001, 0.01, 0.1, 0.2, 0.3, 0.4, 0.5, 0.6\}$ |
| | | Backbone Model | MLP |
| DORO $\chi^2$ | $8100^{\diamond}$ | Uncertainty set size $\alpha$ | $\{0.1, 0.2, 0.3, 0.4, 0.5, 0.6\}$ |
| | | Outlier proportion $\epsilon$ | $\{0.001, 0.01, 0.1, 0.2, 0.3\}$ |
| | | Backbone Model | MLP |
| DORO CVaR | $8100^{\diamond}$ | Uncertainty set size $\alpha$ | $\{0.1, 0.2, 0.3, 0.4, 0.5, 0.6\}$ |
| | | Outlier proportion $\epsilon$ | $\{0.001, 0.01, 0.1, 0.2, 0.3\}$ |
| Group DRO | $1080^{\diamond}$ | Group weights step size | $\{0.001, 0.01, 0.1, 0.2\}$ |
| | | Backbone Model | MLP |
| **Imbalanced Learning Methods** | | | |
| SUBY, RWY, SUBG, RWG | $1944^{\diamond}$ | Backbone Model | XGBoost |
| JTT | $11664^{\diamond}$ | Up-weight $\lambda$ | $\{4, 5, 6, 20, 50, 100\}$ |
| | | Backbone Model | XGBoost |
| DWR | $48600^{\diamond}$ | $L_1$ penalty $\lambda$ | $\{1e-3, 1e-2, 1e-1, 0.2, 0.3\}$ |
| | | $L_2$ penalty $\lambda$ | $\{1e-3, 1e-2, 1e-1, 0.2, 0.3\}$ |
| | | Backbone Model | XGBoost |
| **Fairness Methods** | | | |
| In-processing | $1944^{\diamond}$ | Constraint Type | $\{\text{DP}, \text{EO}, \text{Error Parity}\}$ |
| | | Backbone Model | XGBoost |
| Post-processing | $1944^{\diamond}$ | Constraint Type | $\{\text{Exp}, \text{Threshold}\}$ |
| | | Backbone Model | XGBoost |

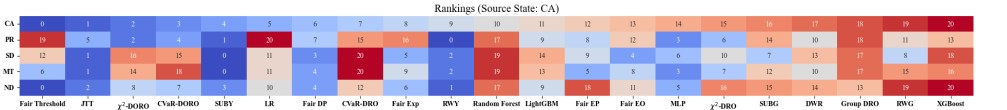

**Figure 10:** Algorithmic rankings vary across different target states for the fixed source state and task.

## D.4 Training Details

In each setting, we randomly sample 20,000 samples from the source domain for training, 20,000 samples from the source domain for validation, and 20,000 samples from the target domain for testing. For US Accident and Taxi datasets, we only randomly sample 8,000 samples for the source/target domain due to fewer samples involved in each setting. For setting where the source domain has fewer samples, we use 80% samples for training and 20% for validation. For results not specified by the validation protocol, we selected the best configuration according to the performance on such validation set ($i.i.d$ with training).

For settings with ACS Income, ACS Public Coverage, ACS Mobility and US Accident datasets, all experiments were run on a server using 48 cores from two AMD EPYC 7402 24-Core Processors. For settings with Taxi dataset, all experiments were run on a cluster using 24 cores from an Intel Xeon Gold 6126 Processor. Neural-network-based models (MLP, DRO methods) were trained on GPU, NVIDIA GeForce RTX 3090, the other methods were trained only on CPUs. Note that all experiments are small-scale and could be run efficiently. Besides, during training, we found that the bi-search in $\chi^2$-DRO sometimes failed to converge, and therefore we set the maximal iteration num to 2500.

## D.5 Detailed Results of 7 Selected Settings in Main Body

In the main body, due to space limitations, we only visualize the target performances of different methods. Here we provide the detailed results of all algorithms on the 7 selected settings in Table 1. We select the top-10 configurations according to the validation set $i.i.d$ with training data and report the mean accuracy as well as the standard deviation in Table 5. Also, in Figure 11, we visualize the results of 22 algorithms of all configurations in our 7 selected settings.

**Table 5.** Results of 7 selected settings in the main body, where we run each method with its best configuration 10 times and report their mean accuracies and standard deviations.

| Dataset / Shift Pattern / Source → Target Pair | | ACS Income $Y\|X$ dominates CA→PR | | ACS Mobility $Y\|X$ dominates MS→HI | | US Taxi $Y\|X$ dominates NYC→BOG | | ACS Pub.Cov $Y\|X$ more NE→LA | | US Accident $Y\|X$ more CA→OR | | ACS Time $X$ more 2010→2017 | | Sub-Sampling $X$ dominates Young→Old | |
|---|---|---|---|---|---|---|---|---|---|---|---|---|---|---|---|
| | | i.d. | o.o.d | i.d. | o.o.d | i.d. | o.o.d | i.d. | o.o.d | i.d. | o.o.d | i.d. | o.o.d | i.d. | o.o.d |
| Base Methods | LR | 80.5±0.2 | **73.2**±0.2 | 76.9±0.2 | **76.9**±2.6 | 83.6±0.3 | 73.7±0.2 | 83.2±0.5 | 66.3±1.4 | 82.3±0.3 | **67.7**±0.3 | 75.3±1.8 | 66.7±2.4 | 92.4±0.0 | 81.5±0.0 |
| | SVM | 79.0±0.3 | 67.0±0.9 | 78.6±0.6 | **76.9**±0.3 | 83.4±0.2 | **74.9**±0.6 | 83.9±0.4 | 64.9±1.2 | 81.3±0.8 | 65.5±0.4 | 75.4±0.9 | 67.4±0.9 | 91.9±0.1 | 79.5±0.7 |
| | MLP | 80.9±0.4 | 68.9±2.5 | 76.8±0.5 | 76.4±2.7 | 84.5±0.5 | 73.8±0.5 | 82.5±1.1 | **66.4**±2.6 | 83.6±0.5 | 63.5±0.8 | 77.0±1.0 | **70.5**±0.5 | 92.3±0.1 | **81.6**±0.3 |
| Tree Ensemble Methods | Random forest | 81.1±0.2 | **72.9**±0.4 | 80.2±0.3 | **72.0**±0.3 | 85.8±0.2 | **73.4**±0.4 | 85.5±0.3 | 68.3±0.4 | 86.2±0.5 | **66.5**±0.4 | 78.5±0.2 | 70.6±0.3 | 92.2±0.1 | 81.2±0.1 |
| | LightGBM | 81.2±0.2 | 71.4±0.5 | 79.5±0.3 | 71.3±0.8 | 86.4±0.4 | 72.7±0.8 | 85.7±0.4 | 70.0±0.4 | 86.6±0.4 | 65.4±0.4 | 79.1±0.3 | **71.8**±0.3 | 93.0±0.0 | **82.4**±0.0 |
| | XGBoost | 81.4±0.2 | 71.7±0.6 | 80.0±0.4 | 71.2±0.8 | 86.5±0.4 | 73.0±0.7 | 85.8±0.5 | 70.1±0.3 | 86.6±0.4 | 65.4±0.3 | 79.0±0.3 | 71.6±0.2 | 92.6±0.0 | 81.6±0.0 |
| DRO Methods (base: MLP) | $\chi^2$-DRO | 80.7±0.2 | **73.2**±2.2 | 76.3±0.3 | 78.4±0.2 | 84.8±0.4 | 73.0±1.1 | 80.6±0.6 | 60.9±1.7 | 83.3±0.4 | 64.0±0.8 | 70.5±12.6 | 64.2±7.6 | 92.1±0.2 | 81.2±0.5 |
| | CVaR-DRO | 80.9±0.2 | 71.7±1.7 | 76.4±0.5 | 78.3±0.4 | 85.4±0.4 | 73.8±0.8 | 79.5±4.0 | 61.7±1.7 | 84.1±0.3 | **64.8**±0.7 | 75.6±1.6 | 67.5±2.7 | 92.3±0.1 | 81.4±0.3 |
| | $\chi^2$-DORO | 77.2±6.2 | 71.0±10.2 | 76.3±0.4 | **78.6**±0.0 | 84.6±0.4 | 73.6±0.7 | 80.6±0.4 | 59.4±0.0 | 83.5±0.4 | 64.5±0.8 | 71.2±0.6 | 64.5±0.3 | 91.9±0.3 | 80.6±1.2 |
| | CVaR-DORO | 78.8±0.6 | 72.4±4.2 | 76.4±0.4 | **78.6**±0.1 | 85.2±0.3 | **74.3**±0.4 | 83.7±0.5 | **66.9**±0.9 | 83.4±2.4 | 63.9±1.0 | 77.6±0.3 | **70.7**±0.4 | 92.0±0.2 | 81.0±0.6 |
| | Group DRO | 80.7±0.3 | 71.9±3.4 | 75.5±0.4 | 78.5±0.2 | N/A | N/A | 80.3±7.3 | 65.8±3.6 | N/A | N/A | 74.4±7.1 | 68.3±4.0 | 92.3±0.1 | **81.6**±0.2 |
| Imbalanced Learning Methods (base: XGB) | JTT | 77.9±0.4 | 69.7±1.3 | 77.2±0.3 | 70.1±0.8 | 85.0±0.3 | 71.4±1.1 | 84.1±0.8 | 68.9±0.4 | 85.9±0.4 | **65.9**±0.7 | 72.9±0.4 | 67.7±1.0 | 91.5±0.0 | 79.0±0.0 |
| | SUBY | 80.5±0.2 | 64.9±0.7 | 76.0±0.2 | 70.3±2.6 | 85.8±0.4 | 72.0±1.9 | 75.2±1.0 | 68.6±0.7 | 85.3±0.3 | 64.4±0.4 | 73.5±0.4 | 70.3±1.5 | 85.7±0.4 | 68.5±0.7 |
| | RWY | 80.9±0.2 | 65.9±0.6 | 75.9±0.4 | 68.5±1.3 | 86.2±0.5 | **73.1**±1.1 | 82.1±0.7 | 69.7±0.5 | 86.1±0.4 | 65.3±0.6 | 74.7±0.5 | 71.3±0.6 | 90.6±0.0 | 78.5±0.0 |
| | SUBG | 81.2±0.3 | 70.9±0.6 | 79.1±0.2 | **71.1**±0.6 | 85.3±0.3 | 71.4±1.2 | 85.1±0.5 | **70.2**±0.4 | 86.4±0.4 | 65.2±0.4 | 78.8±0.3 | 71.4±0.3 | 92.4±0.1 | 81.2±0.3 |
| | RWG | 81.4±0.2 | **71.5**±0.5 | 79.7±0.2 | 71.0±0.8 | 86.3±0.4 | 72.8±0.9 | 85.8±0.5 | 70.1±0.3 | 86.7±0.4 | 65.2±0.4 | 79.1±0.4 | **72.1**±0.3 | 92.9±0.0 | **82.5**±0.0 |
| | DWR | 81.3±0.2 | 71.3±0.6 | 78.2±0.4 | 69.8±0.8 | 83.5±0.5 | 71.2±1.2 | 85.8±0.4 | 69.8±0.4 | 86.7±0.4 | 65.4±0.4 | 79.1±0.3 | 72.0±0.3 | 92.5±0.0 | 81.2±0.0 |
| Fairness Methods (base: XGB) | DP | 80.8±0.3 | 70.4±0.6 | 79.5±0.3 | 71.3±0.8 | 86.3±0.4 | 72.0±1.0 | 85.2±0.5 | **70.2**±0.5 | 86.3±0.4 | 65.6±0.5 | 79.0±0.3 | **71.8**±0.2 | 92.7±0.0 | **81.6**±0.0 |
| | EO | 81.3±0.2 | 71.2±0.6 | 79.3±0.3 | **71.4**±1.0 | 86.4±0.3 | **72.8**±1.2 | 85.8±0.4 | **70.2**±0.4 | 86.4±0.5 | 65.4±0.5 | 79.0±0.3 | **71.8**±0.2 | 92.7±0.0 | 81.5±0.0 |
| | EP | 81.3±0.2 | 71.4±0.5 | 79.6±0.3 | 70.9±0.9 | 85.7±0.4 | 71.7±1.4 | 86.0±0.2 | 70.1±0.5 | 86.3±0.4 | **66.1**±0.1 | 79.0±0.3 | **71.8**±0.2 | 92.6±0.0 | 81.5±0.0 |
| | Exp | 80.9±0.2 | 70.6±0.6 | 79.4±0.6 | 71.2±0.9 | 85.6±0.5 | 71.5±1.1 | 85.5±0.2 | **70.2**±0.5 | 85.8±0.4 | 65.5±0.6 | 79.0±0.3 | **71.8**±0.2 | 92.6±0.0 | 81.5±0.0 |
| | Threshold | 77.2±0.3 | **73.6**±0.4 | 77.5±0.5 | 69.9±1.3 | 84.1±0.6 | 54.3±0.4 | 84.5±0.3 | 68.2±0.7 | 83.8±0.5 | 64.5±0.4 | 78.8±0.2 | 71.7±0.3 | 91.6±0.0 | 78.3±0.0 |
| **Top 3 Classes** | | Fair>Base≈Rob. | | DRO>Base≫Tree. | | Base>DRO>Tree | | Tree≈Fair≈Imb. | | Base>Tree>Fair | | Imb>Fair≈Tree | | Tree≈Imb>Others | |

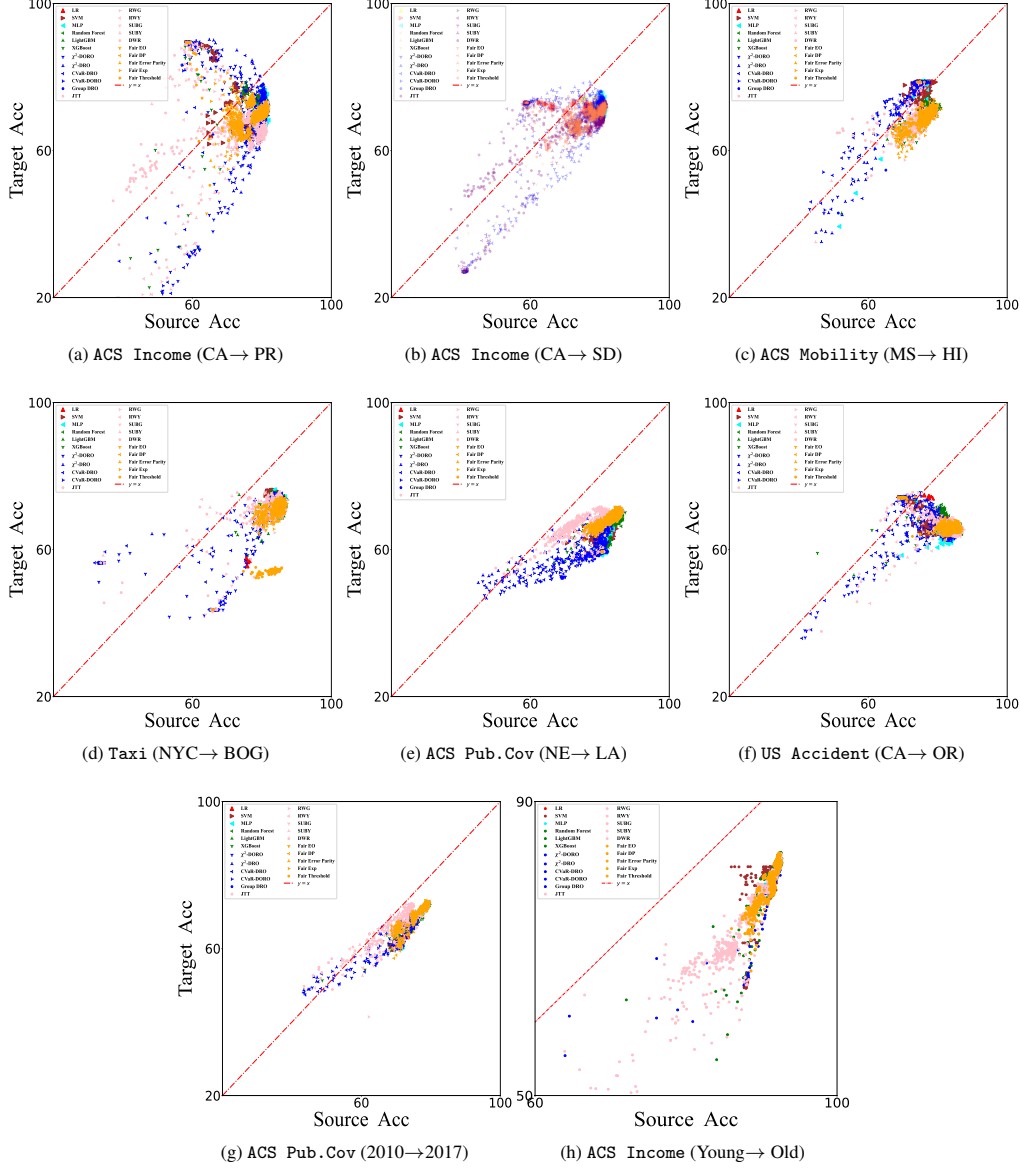

**Figure 11.** Target vs. source accuracies for 22 algorithms and datasets in our benchmark. Each point represents one hyperparameter configuration. **(a)-(b)**: two examples of `ACS Income` dataset with California (CA) as the source state, and Puerto Rico (PR) and South Dakota (SD) as targets. **(c)-(g)**: five examples of `ACS Mobility`, `Taxi`, `ACS Pub.Cov`, `US Accident` datasets. **(h)**: simulated covariate shifts on on sub-sampled `ACS Income` dataset.

**Analysis**   From the numbers of numerical results in that Table, we give a more detailed analysis corresponding with our main body.

- Different algorithms do not exhibit consistent rankings over different distribution shift patterns. In Table 5, we boldface the best target performance within each class of methods. And we show the top-3 classes at the bottom. The results show that the algorithmic rankings across different settings and tasks are quite different. And even the rankings of algorithms within the same class vary a lot. This further demonstrates the complexity of $Y|X$ shifts. Also, Figure 10 shows that even for the fixed source state, algorithmic rankings vary a lot across different target states.

- Tree ensemble methods show competitive performances but do not significantly eliminate the generalization error between source and target data, characterized by the difference

between *o.o.d.* and *i.i.d.* model performance. From the bottom, we could see that tree ensemble methods achieve top-3 performances in most of the selected settings (6 out of 7), exhibiting their superiority on tabular data. However, the performance degradation between source and target is still large.

- Imbalance methods and fairness methods show similar performance with the base learner (XGBoost). We find these methods do not have a significant improvement over the base learner (XGBoost). This may be due to that they are not designed to incorporate tree ensemble methods.

### D.6 Full Results of 22 Settings

In this section, we provide the full results of our 22 settings in Table 3.

#### D.6.1 Average Accuracy

In Table 6, Table 7 and Table 8, we report the source accuracy and the target accuracy of all algorithms. Since in these settings, we have multiple target domains, as for the target accuracy of each algorithm, we report the average accuracy as well as the standard deviation calculated on all target domains. Here the standard deviation reflects the stability of out-of-distribution generalization performances across different target domains different instead of random seeds.

From the average result over, Table 6, Table 7 and Table 8, we can still obtain similar findings compared to that from the 7 selected settings in the main body. Especially, while tree ensemble methods serve as a strong in-sample benchmark illustrated by their competitive *i.i.d.* performance, these methods do not yield better performance than other methods in the target distribution uniformly. And when we average across different target domains, DRO methods do not perform better than their empirical counterpart, indicating that the worst-case optimization from the uncertainty sets constructed by existing DRO methods can rarely occur in practical setups and does not help much compared with standard ERM cases. This tricky part of worst-case intervention also holds when we compare imbalanced learning and fairness methods with the XGB base learner. These methods can lead to better *o.o.d.* results than the base learner sometimes indeed, while this may not hold true when we average over different target domains. In general, existing methods cannot generalize well averaging over different target domains, and it would be interesting to develop methods to close that gap.

### D.7 Worst-Domain Accuracy

Another widely-used metric in evaluating generalization performance is worst-case accuracy. In settings built on `ACS Income`, `ACS Pub.Cov`, `ACS Mobility`, and `US Accident`, for the target accuracy of each algorithm, we report the worst target domain accuracy in Table 9 and Table 10. For other settings, we do not report the worst-case accuracy because there are not many target domains, and the average& standard deviation results could reflect the population generalization performance well. In terms of the worst target accuracy for each setting, we usually observe a large performance degradation (usually over a 10 percent performance drop) in all of the existing methods. And algorithms do not show a consistent and stable ranking performance in the *o.o.d.* performance. DRO / Imbalanced learning methods can perform quite well in one *o.o.d.* setup but are dominated by other methods in some other cases. This demonstrates a need to carefully understand the difference between different domains first and develop corresponding methods then.

### D.8 Average Macro-F1 Score

In consideration of the label imbalance in real-world tabular datasets, we also calculate the Macro-F1 Score for all algorithms under our settings. In this section, we report the average results in Table 11, Table 12, and Table 13. Here the standard deviation reflects the stability of out-of-distribution generalization performances (not the randomness).

### D.9 Worst-Domain Macro-F1 Score

In settings built on `ACS Income`, `ACS Pub.Cov`, `ACS Mobility`, and `US Accident`, for the target accuracy of each algorithm, we report the worst target domain Macro-F1 score in Table 14 and

**Table 6. Average target domain accuracy** of all algorithms on the `ACS Income` and `ACS Pub.Cov` datasets. In each setting, we test all algorithms on the 50 American states and report the mean and standard deviation across all 50 target states. The standard deviation here reflects the stability of performances across different target states.

| Dataset | | ACS INCOME | | | | | | | | ACS PUB.COV | | | | | | | |
|---|---|---|---|---|---|---|---|---|---|---|---|---|---|---|---|---|---|
| Source State | | CA | | CT | | MA | | SD | | FL | | TX | | NE | | IN | |
| | | i.d. | o.o.d | i.d. | o.o.d | i.d. | o.o.d | i.d. | o.o.d | i.d. | o.o.d | i.d. | o.o.d | i.d. | o.o.d | i.d. | o.o.d |
| Base Methods | LR | 81.1 | 77.3 ±1.9 | 80.7 | 76.2 ±2.3 | 80.9 | 76.8 ±2.2 | 80.1 | 75.8 ±3.3 | 81.3 | 73.7 ±7.3 | 84.1 | 73.9 ±7.2 | 84.3 | 75.0 ±6.7 | 78.6 | 75.8 ±5.9 |
| | SVM | 79.4 | 74.8 ±1.9 | 78.6 | 72.4 ±2.9 | 80.1 | 73.0 ±2.7 | 79.3 | 73.3 ±3.8 | 80.5 | 74.4 ±6.3 | 84.2 | 74.1 ±6.9 | 84.8 | 73.8 ±6.5 | 77.1 | 74.3 ±5.1 |
| | MLP | 81.8 | 76.4 ±2.6 | 81.4 | 76.8 ±2.3 | 81.7 | 77.4 ±2.2 | 80.5 | 76.8 ±2.9 | 83.3 | 76.5 ±6.4 | 85.4 | 76.6 ±6.5 | 84.5 | 75.3 ±6.3 | 78.8 | 76.2 ±5.9 |
| Tree Ensemble Methods | Random Forest | 81.7 | 77.9 ±1.8 | 81.2 | 77.1 ±2.1 | 81.5 | 77.4 ±2.0 | 80.2 | 75.5 ±3.6 | 84.6 | 77.3 ±6.7 | 86.8 | 77.0 ±6.8 | 86.7 | 76.2 ±6.7 | 81.2 | 77.7 ±5.4 |
| | LightGBM | 81.8 | 77.3 ±2.1 | 82.1 | 76.7 ±2.4 | 82.3 | 77.4 ±2.4 | 80.2 | 76.1 ±2.9 | 84.6 | 77.7 ±6.4 | 86.9 | 77.4 ±6.6 | 86.9 | 76.6 ±6.3 | 81.2 | 78.0 ±5.2 |
| | XGBoost | 82.0 | 72.5 ±0.4 | 81.9 | 76.6 ±2.4 | 82.2 | 77.2 ±2.3 | 80.5 | 76.9 ±3.0 | 84.6 | 77.4 ±6.7 | 86.7 | 77.2 ±6.7 | 87.3 | 76.3 ±5.8 | 81.2 | 77.9 ±5.1 |
| DRO Methods | $\chi^2$-DRO | 81.8 | 77.0 ±2.3 | 81.2 | 76.7 ±2.4 | 81.6 | 76.0 ±2.7 | 80.2 | 76.3 ±3.0 | 82.9 | 75.9 ±6.9 | 85.4 | 75.6 ±7.2 | 83.4 | 74.7 ±6.2 | 78.4 | 75.5 ±4.6 |
| | CVaR-DRO | 81.4 | 77.7 ±1.9 | 81.2 | 76.3 ±2.4 | 82.0 | 76.7 ±2.4 | 79.7 | 75.5 ±3.5 | 82.6 | 75.3 ±7.0 | 85.2 | 75.8 ±6.9 | 83.3 | 73.4 ±6.5 | 77.8 | 75.2 ±6.4 |
| | $\chi^2$-DORO | 81.1 | 77.8 ±1.7 | 80.4 | 75.1 ±2.9 | 81.2 | 75.7 ±2.7 | 78.8 | 75.9 ±2.6 | 78.8 | 70.0 ±8.2 | 82.1 | 69.9 ±8.1 | 82.2 | 70.0 ±8.1 | 72.4 | 70.2 ±6.9 |
| | CVaR-DORO | 80.9 | 76.7 ±2.0 | 80.5 | 75.6 ±2.7 | 81.1 | 75.2 ±2.8 | 79.6 | 75.8 ±2.7 | 80.7 | 73.0 ±7.2 | 84.3 | 75.4 ±6.7 | 81.9 | 69.9 ±8.1 | 76.4 | 74.2 ±5.8 |
| | Group DRO | 81.9 | 77.7 ±2.0 | 81.3 | 77.0 ±2.2 | 81.7 | 76.8 ±2.4 | 79.8 | 75.7 ±3.6 | 83.3 | 76.1 ±6.7 | 85.4 | 76.6 ±6.6 | 83.5 | 72.8 ±6.5 | 72.2 | 70.2 ±8.2 |
| Imbalanced Learning Methods | SUBY | 81.0 | 75.1 ±3.1 | 81.5 | 76.0 ±2.8 | 82.0 | 76.1 ±2.4 | 74.8 | 75.7 ±1.3 | 79.8 | 72.8 ±6.4 | 82.3 | 71.5 ±7.7 | 77.8 | 71.5 ±3.4 | 77.8 | 74.9 ±3.0 |
| | RWY | 81.5 | 75.7 ±3.0 | 81.8 | 75.9 ±2.8 | 82.1 | 76.5 ±2.7 | 77.7 | 76.2 ±1.1 | 80.7 | 75.2 ±6.4 | 82.5 | 72.2 ±7.5 | 83.7 | 74.3 ±4.5 | 78.7 | 75.6 ±3.3 |
| | SUBG | 81.9 | 77.3 ±2.2 | 81.6 | 76.8 ±2.4 | 82.0 | 76.9 ±2.5 | 80.2 | 76.0 ±3.2 | 84.3 | 77.4 ±6.5 | 86.7 | 77.1 ±6.7 | 86.3 | 76.1 ±5.8 | 80.8 | 77.8 ±5.2 |
| | RWG | 82.0 | 77.4 ±2.2 | 81.9 | 76.7 ±2.4 | 82.2 | 77.2 ±2.3 | 80.2 | 76.8 ±3.0 | 84.5 | 77.5 ±6.4 | 86.8 | 77.4 ±6.8 | 86.7 | 76.4 ±5.9 | 81.0 | 78.0 ±5.4 |
| | JTT | 78.3 | 74.1 ±2.5 | 78.1 | 75.3 ±1.7 | 79.1 | 75.8 ±1.8 | 77.9 | 74.7 ±2.6 | 80.6 | 74.9 ±5.1 | 83.1 | 75.6 ±5.7 | 84.7 | 75.4 ±5.7 | 77.6 | 74.2 ±4.6 |
| | DWR | 81.9 | 77.5 ±2.1 | 81.6 | 76.8 ±2.2 | 82.1 | 77.0 ±2.5 | 80.1 | 76.3 ±2.0 | 84.3 | 77.5 ±6.4 | 86.8 | 77.3 ±6.7 | 85.9 | 76.4 ±6.4 | 80.6 | 78.0 ±5.2 |
| Fairness Methods | DP | 81.3 | 76.3 ±2.4 | 81.0 | 75.4 ±2.6 | 81.8 | 76.7 ±2.4 | 79.3 | 75.5 ±3.3 | 84.3 | 77.4 ±6.4 | 86.6 | 77.2 ±6.7 | 86.4 | 76.3 ±5.8 | 80.6 | 77.8 ±5.2 |
| | EO | 81.8 | 77.1 ±2.3 | 81.8 | 76.5 ±2.4 | 82.1 | 77.2 ±2.4 | 80.0 | 76.7 ±2.9 | 84.4 | 77.2 ±6.7 | 86.8 | 77.1 ±6.9 | 86.4 | 76.6 ±6.3 | 80.8 | 77.7 ±4.9 |
| | EP | 81.8 | 77.2 ±2.3 | 81.9 | 76.5 ±2.5 | 82.1 | 77.5 ±2.2 | 80.3 | 76.2 ±3.3 | 84.4 | 77.3 ±6.6 | 86.8 | 77.3 ±6.6 | 86.4 | 76.5 ±6.2 | 80.9 | 77.6 ±4.9 |
| | Exp | 81.4 | 76.8 ±2.2 | 0.81 | 75.3 ±2.6 | 81.8 | 76.5 ±2.4 | 79.1 | 75.5 ±3.1 | 84.5 | 77.5 ±6.6 | 86.8 | 77.1 ±6.8 | 86.4 | 76.6 ±6.3 | 80.9 | 78.2 ±5.2 |
| | Threshold | 77.9 | 75.0 ±1.3 | 77.0 | 74.1 ±1.8 | 77.5 | 74.4 ±1.3 | 78.0 | 71.7 ±4.0 | 84.3 | 77.2 ±6.3 | 86.2 | 76.7 ±6.9 | 85.8 | 76.0 ±6.5 | 80.0 | 77.0 ±5.3 |

**Table 7. Average target domain accuracy** of all algorithms on the `ACS Mobility` and `US Accident` datasets. In `ACS Mobility`, we test all algorithms on the 50 American states and report the mean and standard deviation across all 50 target states. And in `US Accident`, we choose 13 American states with relatively high sample sizes in testing. The standard deviation here reflects the stability of performances across different target states.

| Dataset | | ACS MOBILITY | | | | | | | | US ACCIDENT | | | | | | | |
|---|---|---|---|---|---|---|---|---|---|---|---|---|---|---|---|---|---|
| Source State | | NY | | CA | | MS | | PA | | CA | | FL | | TX | | MN | |
| | | i.d. | o.o.d | i.d. | o.o.d | i.d. | o.o.d | i.d. | o.o.d | i.d. | o.o.d | i.d. | o.o.d | i.d. | o.o.d | i.d. | o.o.d |
| Base Methods | LR | 78.6 | 72.7 ±3.6 | 77.1 | 72.7 ±3.7 | 77.3 | 72.7 ±3.7 | 75.4 | 72.7 ±3.4 | 83.2 | 70.1 ±15.4 | 90.7 | 71.7 ±13.9 | 95.1 | 74.5 ±9.6 | 86.4 | 72.2 ±11.8 |
| | SVM | 79.8 | 72.7 ±3.6 | 77.7 | 72.9 ±3.1 | 79.4 | 72.9 ±3.6 | 77.1 | 72.7 ±3.6 | 83.2 | 69.4 ±12.6 | 88.7 | 71.9 ±9.6 | 93.7 | 70.7 ±9.1 | 88.6 | 69.7 ±8.3 |
| | MLP | 78.7 | 72.7 ±3.6 | 77.1 | 72.7 ±3.7 | 78.1 | 72.4 ±3.3 | 75.9 | 73.0 ±3.1 | 84.7 | 75.7 ±10.7 | 92.7 | 78.2 ±8.7 | 94.9 | 73.0 ±9.9 | 88.9 | 74.0 ±10.2 |
| Tree Ensemble Methods | Random Forest | 80.4 | 74.0 ±3.1 | 79.0 | 74.1 ±3.1 | 80.6 | 73.5 ±3.4 | 78.6 | 74.3 ±3.0 | 86.6 | 77.9 ±7.8 | 93.5 | 84.4 ±10.1 | 95.2 | 84.4 ±9.9 | 91.6 | 82.0 ±5.7 |
| | LightGBM | 80.6 | 74.6 ±2.8 | 78.7 | 74.1 ±2.8 | 80.0 | 72.9 ±2.9 | 78.6 | 74.7 ±2.5 | 87.1 | 79.1 ±7.8 | 93.3 | 84.3 ±9.8 | 95.4 | 84.3 ±10.0 | 92.2 | 80.9 ±5.4 |
| | XGBoost | 80.4 | 74.6 ±3.0 | 0.79 | 74.5 ±2.8 | 80.9 | 73.5 ±3.0 | 79.0 | 74.3 ±2.4 | 87.1 | 79.8 ±7.6 | 94.6 | 82.6 ±9.3 | 95.3 | 84.4 ±9.9 | 92.5 | 79.6 ±5.4 |
| DRO Methods | $\chi^2$-DRO | 78.5 | 72.7 ±3.6 | 77.5 | 72.9 ±3.4 | 76.8 | 72.7 ±3.5 | 76.0 | 73.1 ±3.3 | 84.4 | 77.3 ±9.3 | 92.6 | 73.4 ±12.6 | 95.0 | 74.6 ±9.1 | 89.0 | 72.3 ±10.4 |
| | CVaR-DRO | 78.6 | 72.6 ±3.5 | 77.2 | 72.8 ±3.5 | 77.7 | 72.6 ±3.7 | 75.7 | 73.3 ±3.2 | 84.7 | 74.5 ±10.3 | 92.7 | 74.4 ±11.4 | 95.0 | 74.9 ±9.4 | 89.5 | 73.4 ±10.0 |
| | $\chi^2$-DORO | 78.6 | 72.6 ±3.6 | 77.8 | 72.8 ±3.3 | 77.1 | 72.7 ±3.7 | 75.3 | 72.7 ±3.7 | 84.7 | 72.9 ±11.5 | 92.3 | 77.6 ±8.9 | 94.3 | 74.3 ±9.6 | 87.6 | 74.7 ±8.7 |
| | CVaR-DORO | 78.7 | 72.7 ±3.6 | 78.0 | 73.6 ±3.1 | 77.4 | 72.7 ±3.7 | 75.3 | 72.7 ±3.6 | 84.9 | 74.1 ±10.5 | 92.8 | 74.4 ±11.2 | 94.9 | 74.7 ±9.5 | 89.1 | 74.1 ±9.5 |
| | Group DRO | 78.6 | 72.5 ±3.4 | 77.1 | 72.7 ±3.6 | 76.0 | 72.7 ±3.7 | 76.0 | 72.8 ±3.2 | N/A | N/A | N/A | N/A | N/A | N/A | N/A | N/A |
| Imbalanced Learning Methods | SUBY | 78.2 | 72.7 ±3.5 | 76.3 | 72.5 ±3.6 | 76.6 | 70.0 ±2.8 | 74.7 | 69.5 ±2.0 | 86.5 | 85.4 ±7.3 | 93.7 | 84.6 ±10.0 | 95.2 | 84.4 ±9.9 | 91.8 | 83.8 ±6.6 |
| | RWY | 78.5 | 72.7 ±3.6 | 76.7 | 72.7 ±3.6 | 76.5 | 72.2 ±3.4 | 74.5 | 72.6 ±3.7 | 86.9 | 82.2 ±7.1 | 93.4 | 84.5 ±10.2 | 95.3 | 84.3 ±10.0 | 92.5 | 81.1 ±5.4 |
| | SUBG | 80.2 | 74.3 ±2.9 | 79.2 | 74.2 ±2.8 | 79.5 | 72.5 ±2.9 | 78.6 | 74.3 ±2.7 | 86.3 | 81.0 ±7.3 | 93.0 | 83.8 ±9.9 | 95.1 | 84.4 ±9.9 | 91.1 | 80.1 ±5.8 |
| | RWG | 80.3 | 74.6 ±3.0 | 79.1 | 74.4 ±2.8 | 80.1 | 73.3 ±2.9 | 78.7 | 74.2 ±2.6 | 87.5 | 79.7 ±7.5 | 93.4 | 83.8 ±9.4 | 95.2 | 84.4 ±9.9 | 92.4 | 79.3 ±5.3 |
| | JTT | 78.3 | 72.7 ±3.6 | 76.5 | 72.4 ±3.7 | 77.6 | 71.5 ±2.6 | 75.1 | 70.7 ±2.4 | 86.5 | 79.0 ±7.7 | 92.7 | 83.6 ±9.5 | 94.9 | 84.3 ±9.9 | 91.8 | 78.2 ±5.6 |
| | DWR | 79.7 | 74.2 ±3.1 | 78.3 | 73.9 ±2.9 | 79.1 | 72.9 ±2.9 | 77.5 | 74.0 ±2.6 | 87.2 | 79.8 ±7.7 | 93.4 | 83.7 ±9.9 | 95.3 | 84.4 ±9.9 | 92.3 | 78.5 ±5.8 |
| Fairness Methods | DP | 80.5 | 74.7 ±2.8 | 78.8 | 74.2 ±2.9 | 80.0 | 73.8 ±3.1 | 78.3 | 74.0 ±2.6 | 86.6 | 79.7 ±7.5 | 92.8 | 83.9 ±9.5 | 94.8 | 84.3 ±9.9 | 91.9 | 81.9 ±5.8 |
| | EO | 80.3 | 74.7 ±2.8 | 78.9 | 74.1 ±2.8 | 79.8 | 73.4 ±3.0 | 78.5 | 73.9 ±2.7 | 86.6 | 79.5 ±7.5 | 93.2 | 84.4 ±9.6 | 95.2 | 84.4 ±10.1 | 92.2 | 79.6 ±5.4 |
| | EP | 80.3 | 74.7 ±2.8 | 78.9 | 74.1 ±3.0 | 80.1 | 74.0 ±3.0 | 78.4 | 74.1 ±2.7 | 87.0 | 80.3 ±7.5 | 93.3 | 84.2 ±9.9 | 95.1 | 84.4 ±10.0 | 91.9 | 79.6 ±5.2 |
| | Exp | 80.5 | 74.8 ±2.9 | 78.9 | 74.1 ±2.9 | 81.0 | 73.9 ±2.9 | 78.4 | 74.3 ±2.6 | 85.9 | 78.6 ±7.3 | 90.1 | 82.3 ±9.1 | 93.1 | 81.6 ±9.6 | 86.3 | 80.4 ±6.0 |
| | Threshold | 80.5 | 74.5 ±2.9 | 78.6 | 74.3 ±2.8 | 78.2 | 71.9 ±2.9 | 78.5 | 73.9 ±2.6 | 84.3 | 77.8 ±7.3 | 88.1 | 79.9 ±7.9 | 91.2 | 79.9 ±9.3 | 82.0 | 75.7 ±4.8 |

**Table 8. Average target domain accuracy** of all algorithms on the `Taxi`, `ACS Pub.Cov` (Temporal) and `ACS Income` (Synthetic) datasets. In `Taxi` and `ACS Pub.Cov` (Temporal), we test all algorithms on the 3 target domains and report the mean and standard deviation across all 3 target domains. The standard deviation here reflects the stability of performances across different target states. In `ACS Income` (Synthetic), we simulate strong covariate shifts according to the "Age" feature, where 10% and 20% means the minor group ratio in training, respectively.

| Dataset | | Taxi | | | | ACS Pub.Cov (Temporal) | | | | ACS Income (Synthetic) | | | |
|---|---|---|---|---|---|---|---|---|---|---|---|---|---|
| Source | | BOG | | NYC | | NY | | CA | | 20% | | 10% | |
| | | i.d. | o.o.d | i.d. | o.o.d | i.d. | o.o.d | i.d. | o.o.d | i.d. | o.o.d | i.d. | o.o.d |
| Base Methods | LR | 75.6 | 71.2 ±2.0 | 83.6 | 74.2 ±1.2 | 76.3 | 68.5 ±2.3 | 80.8 | 68.2 ±4.0 | 92.2 | 81.6 | 94.0 | 79.3 |
| | SVM | 75.7 | 76.1 ±3.1 | 83.4 | 74.4 ±1.4 | 77.4 | 69.1 ±2.0 | 80.7 | 67.7 ±3.9 | 92.0 | 80.0 | 94.1 | 79.5 |
| | MLP | 77.7 | 70.0 ±1.3 | 84.5 | 73.7 ±0.6 | 78.0 | 70.6 ±2.4 | 82.5 | 70.3 ±3.8 | 92.5 | 82.1 | 94.3 | 79.5 |
| Tree Ensemble Methods | Random Forest | 76.1 | 77.5 ±4.7 | 85.7 | 71.6 ±0.9 | 79.1 | 71.3 ±2.5 | 83.4 | 70.4 ±4.1 | 92.7 | 81.7 | 94.4 | 78.5 |
| | LightGBM | 86.3 | 73.2 ±4.3 | 87.0 | 72.0 ±0.9 | 79.7 | 72.1 ±2.4 | 84.0 | 70.7 ±3.9 | 92.6 | 83.0 | 94.2 | 80.4 |
| | XGBoost | 85.0 | 71.5 ±2.7 | 86.7 | 72.5 ±1.0 | 79.3 | 71.9 ±2.6 | 83.6 | 71.1 ±3.7 | 92.7 | 82.2 | 94.4 | 79.7 |
| DRO Methods | $\chi^2$-DRO | 77.4 | 72.2 ±2.4 | 84.6 | 73.0 ±0.8 | 77.3 | 69.0 ±2.5 | 81.9 | 70.0 ±3.8 | 92.5 | 81.3 | 94.2 | 79.7 |
| | CVaR-DRO | 77.0 | 73.4 ±2.5 | 85.9 | 72.8 ±0.6 | 76.6 | 69.1 ±2.4 | 82.2 | 69.1 ±4.5 | 92.5 | 81.2 | 94.3 | 79.0 |
| | $\chi^2$-DORO | 75.8 | 73.3 ±2.9 | 84.9 | 73.9 ±0.8 | 71.7 | 61.2 ±3.5 | 79.7 | 64.8 ±3.9 | 92.3 | 81.4 | 93.8 | 77.4 |
| | CVaR-DORO | 75.6 | 74.3 ±3.0 | 85.1 | 73.8 ±0.8 | 76.0 | 67.9 ±2.6 | 79.5 | 64.9 ±4.0 | 92.5 | 81.4 | 93.9 | 76.7 |
| | Group DRO | N/A | N/A | N/A | N/A | 76.8 | 70.5 ±2.1 | 81.9 | 70.0 ±3.9 | 92.5 | 81.4 | 94.2 | 79.8 |
| Imbalanced Learning Methods | SUBY | 87.2 | 67.1 ±1.6 | 85.9 | 70.8 ±0.5 | 73.9 | 69.8 ±1.3 | 81.4 | 68.3 ±3.8 | 87.0 | 69.9 | 88.5 | 64.2 |
| | RWY | 81.2 | 73.0 ±1.4 | 86.7 | 72.4 ±0.7 | 75.1 | 70.9 ±1.6 | 79.4 | 65.8 ±4.1 | 90.3 | 78.6 | 92.1 | 76.3 |
| | SUBG | 86.3 | 72.5 ±4.4 | 85.6 | 71.4 ±0.9 | 79.4 | 71.8 ±2.3 | 83.7 | 70.7 ±3.6 | 92.6 | 81.7 | 94.4 | 79.5 |
| | RWG | 81.1 | 69.1 ±3.6 | 86.5 | 71.3 ±0.9 | 79.4 | 72.8 ±2.2 | 83.8 | 70.8 ±3.7 | 92.6 | 82.7 | 94.5 | 80.1 |
| | JTT | 84.6 | 69.5 ±0.9 | 85.4 | 71.2 ±0.8 | 73.8 | 66.7 ±2.3 | 79.5 | 67.4 ±3.5 | 91.5 | 78.9 | 93.5 | 76.5 |
| | DWR | 82.9 | 68.2 ±1.9 | 83.7 | 70.7 ±0.9 | 79.2 | 71.3 ±2.4 | 83.8 | 70.7 ±3.9 | 92.5 | 82.6 | 94.2 | 80.3 |
| Fairness Methods | DP | 84.1 | 70.2 ±4.0 | 86.3 | 72.0 ±0.8 | 79.6 | 72.7 ±2.0 | 83.7 | 71.1 ±3.9 | 92.5 | 82.7 | 94.2 | 80.2 |
| | EO | 81.1 | 71.9 ±3.0 | 87.1 | 71.8 ±0.8 | 79.5 | 72.3 ±2.5 | 83.4 | 70.1 ±4.0 | 92.6 | 81.7 | 94.3 | 79.5 |
| | EP | 79.7 | 71.6 ±1.7 | 85.7 | 70.9 ±0.6 | 79.6 | 72.5 ±2.1 | 83.6 | 71.0 ±3.9 | 92.4 | 82.6 | 94.2 | 80.0 |
| | Exp | 79.2 | 68.6 ±0.4 | 84.9 | 71.6 ±0.6 | 79.3 | 72.3 ±2.3 | 83.4 | 71.1 ±4.2 | 92.6 | 81.9 | 94.3 | 79.4 |
| | Threshold | 80.8 | 69.8 ±1.7 | 84.3 | 59.7 ±3.9 | 79.2 | 71.9 ±2.2 | 83.4 | 70.4 ±3.9 | 91.8 | 78.3 | 93.5 | 74.0 |

**Table 9. Worst target domain accuracy** of all algorithms on the `ACS Income` and `ACS Pub.Cov` datasets. In each setting, we test all algorithms on the 50 American states and report the worst accuracy among all 50 target states.

| Dataset | | ACS INCOME | | | | | | | | ACS PUB.COV | | | | | | | |
|---|---|---|---|---|---|---|---|---|---|---|---|---|---|---|---|---|---|
| Source State | | CA | | CT | | MA | | SD | | FL | | TX | | NE | | IN | |
| | | i.d. | o.o.d | i.d. | o.o.d | i.d. | o.o.d | i.d. | o.o.d | i.d. | o.o.d | i.d. | o.o.d | i.d. | o.o.d | i.d. | o.o.d |
| Base Methods | LR | 81.1 | 72.3 | 80.7 | 70.5 | 80.9 | 71.1 | 80.1 | 69.4 | 81.3 | 49.6 | 84.1 | 49.0 | 84.3 | 0.52 | 78.6 | 54.7 |
| | SVM | 79.4 | 67.1 | 78.6 | 61.6 | 80.1 | 62.2 | 79.3 | 64.6 | 80.5 | 53.0 | 84.2 | 50.3 | 84.8 | 50.4 | 77.1 | 57.4 |
| | MLP | 81.8 | 68.3 | 81.4 | 70.8 | 81.7 | 71.3 | 80.5 | 70.5 | 83.3 | 54.3 | 85.4 | 53.9 | 84.5 | 52.9 | 78.8 | 56.7 |
| Tree Ensemble Methods | Random Forest | 81.7 | 72.6 | 81.2 | 71.6 | 81.5 | 71.2 | 80.2 | 68.3 | 84.6 | 54.2 | 86.8 | 52.8 | 86.7 | 52.9 | 81.2 | 59.0 |
| | LightGBM | 81.8 | 71.1 | 82.1 | 69.5 | 82.3 | 70.1 | 80.2 | 69.6 | 84.6 | 56.9 | 86.9 | 53.9 | 86.9 | 52.3 | 81.2 | 60.4 |
| | XGBoost | 82.0 | 72.2 | 81.9 | 70.1 | 82.2 | 70.9 | 80.5 | 70.5 | 84.6 | 54.8 | 86.7 | 53.7 | 87.3 | 55.4 | 81.2 | 59.6 |
| DRO Methods | $\chi^2$-DRO | 81.8 | 69.7 | 81.2 | 69.7 | 81.6 | 67.9 | 80.2 | 70.3 | 82.9 | 52.5 | 85.4 | 50.5 | 83.4 | 52.9 | 78.4 | 62.4 |
| | CVaR-DRO | 81.4 | 72.1 | 81.2 | 70.7 | 82.0 | 70.0 | 79.7 | 68.4 | 82.6 | 52.2 | 85.2 | 51.7 | 83.3 | 52.3 | 77.8 | 53.9 |
| | $\chi^2$-DORO | 81.1 | 70.3 | 80.4 | 62.6 | 81.2 | 63.8 | 78.8 | 69.5 | 78.8 | 43.6 | 82.1 | 43.6 | 82.2 | 43.6 | 72.4 | 49.9 |
| | CVaR-DORO | 80.9 | 68.7 | 80.5 | 64.9 | 81.1 | 64.3 | 79.6 | 69.1 | 80.7 | 49.5 | 84.3 | 52.4 | 81.9 | 43.6 | 76.4 | 56.2 |
| | Group DRO | 81.9 | 72.6 | 81.3 | 71.8 | 81.7 | 69.2 | 79.8 | 0.68 | 83.3 | 53.2 | 85.4 | 53.2 | 83.5 | 51.3 | 72.2 | 43.6 |
| Imbalanced Learning Methods | SUBY | 81.0 | 65.2 | 81.5 | 68.0 | 82.0 | 66.3 | 74.8 | 72.5 | 79.8 | 51.3 | 82.3 | 45.4 | 77.8 | 64.5 | 77.8 | 68.3 |
| | RWY | 81.5 | 65.2 | 81.8 | 68.3 | 82.1 | 67.6 | 77.7 | 73.2 | 80.7 | 53.3 | 82.5 | 45.6 | 83.7 | 57.1 | 78.7 | 65.8 |
| | SUBG | 81.9 | 71.4 | 81.6 | 70.1 | 82.0 | 68.4 | 80.2 | 70.3 | 84.3 | 56.1 | 86.7 | 53.0 | 86.3 | 53.6 | 80.8 | 60.2 |
| | RWG | 82.0 | 71.3 | 81.9 | 70.6 | 82.2 | 70.3 | 80.2 | 70.6 | 84.5 | 56.8 | 86.8 | 52.1 | 86.7 | 55.1 | 81.0 | 57.2 |
| | JTT | 78.3 | 68.4 | 78.1 | 71.2 | 79.1 | 71.6 | 77.9 | 68.6 | 80.6 | 59.4 | 83.1 | 56.2 | 84.7 | 55.4 | 77.6 | 56.7 |
| | DWR | 81.9 | 71.2 | 81.6 | 71.0 | 82.1 | 69.0 | 80.1 | 72.0 | 84.3 | 58.1 | 86.8 | 54.2 | 85.9 | 52.8 | 80.6 | 59.4 |
| Fairness Methods | DP | 81.3 | 70.3 | 81.0 | 68.8 | 81.8 | 70.7 | 79.3 | 68.4 | 84.3 | 57.4 | 86.6 | 53.7 | 86.4 | 52.9 | 80.6 | 57.8 |
| | EO | 81.8 | 70.6 | 81.8 | 70.5 | 82.1 | 70.8 | 80.0 | 70.6 | 84.4 | 55.4 | 86.8 | 53.6 | 86.4 | 53.6 | 80.8 | 61.6 |
| | EP | 81.8 | 70.7 | 81.9 | 69.8 | 82.1 | 70.9 | 80.3 | 69.2 | 84.4 | 55.5 | 86.8 | 54.4 | 86.4 | 53.6 | 80.9 | 60.1 |
| | Exp | 81.4 | 70.6 | 81.0 | 69.4 | 81.8 | 70.4 | 79.1 | 68.8 | 84.5 | 55.4 | 86.8 | 52.8 | 86.4 | 52.5 | 80.9 | 61.5 |
| | Threshold | 77.9 | 71.8 | 77.0 | 69.8 | 77.5 | 70.9 | 78.0 | 62.8 | 84.3 | 57.3 | 86.2 | 52.4 | 85.8 | 52.9 | 80.0 | 59.9 |

**Table 10. Worst target domain accuracy** of all algorithms on the `ACS Mobility` and `US Accident` datasets. In `ACS Mobility`, we test all algorithms on the 50 American states and report the worst accuracy among all 50 target states. And in `US Accident`, we choose 13 American states with relatively high sample sizes in testing.

| Dataset | | ACS MOBILITY | | | | | | | | US ACCIDENT | | | | | | | |
|---|---|---|---|---|---|---|---|---|---|---|---|---|---|---|---|---|---|
| Source State | | NY | | CA | | MS | | PA | | CA | | FL | | TX | | MN | |
| | | i.d. | o.o.d | i.d. | o.o.d | i.d. | o.o.d | i.d. | o.o.d | i.d. | o.o.d | i.d. | o.o.d | i.d. | o.o.d | i.d. | o.o.d |
| Base Methods | LR | 78.6 | 64.7 | 77.1 | 64.6 | 77.3 | 64.8 | 75.4 | 66.3 | 83.2 | 44.9 | 90.7 | 48.3 | 95.1 | 57.4 | 86.4 | 52.8 |
| | SVM | 79.8 | 64.6 | 77.7 | 66.8 | 79.4 | 65.3 | 77.1 | 64.7 | 83.2 | 49.2 | 88.7 | 58.3 | 93.7 | 55.7 | 88.6 | 56.1 |
| | MLP | 78.7 | 64.7 | 77.1 | 64.8 | 78.1 | 66.3 | 75.9 | 67.0 | 84.7 | 58.4 | 92.7 | 61.1 | 94.9 | 57.2 | 88.9 | 57.1 |
| Tree Ensemble Methods | Random Forest | 80.4 | 68.0 | 79.0 | 68.2 | 80.6 | 67.5 | 78.6 | 68.7 | 86.6 | 66.6 | 93.5 | 58.2 | 95.2 | 58.5 | 91.6 | 68.0 |
| | LightGBM | 80.6 | 68.7 | 78.7 | 69.0 | 80.0 | 67.9 | 78.6 | 69.8 | 87.1 | 65.3 | 93.3 | 59.0 | 95.4 | 58.6 | 92.2 | 68.1 |
| | XGBoost | 80.4 | 68.6 | 79.0 | 69.0 | 80.9 | 68.3 | 79.0 | 69.4 | 87.1 | 65.6 | 94.6 | 59.0 | 95.3 | 58.8 | 92.5 | 68.5 |
| DRO Methods | $\chi^2$-DRO | 78.5 | 65.1 | 77.5 | 66.6 | 76.8 | 65.4 | 76.0 | 67.1 | 84.4 | 62.9 | 92.6 | 53.3 | 95.0 | 57.9 | 89.0 | 53.5 |
| | CVaR-DRO | 78.6 | 65.0 | 77.2 | 65.6 | 77.7 | 64.4 | 75.7 | 67.6 | 84.7 | 57.2 | 92.7 | 57.5 | 95.0 | 58.4 | 89.5 | 56.3 |
| | $\chi^2$-DORO | 78.6 | 64.3 | 77.8 | 66.9 | 77.1 | 64.7 | 75.3 | 64.7 | 84.7 | 53.9 | 92.3 | 61.1 | 94.3 | 57.0 | 87.6 | 59.8 |
| | CVaR-DORO | 78.7 | 64.7 | 78.0 | 67.4 | 77.4 | 64.7 | 75.3 | 64.9 | 84.9 | 57.8 | 92.8 | 58.1 | 94.9 | 57.5 | 89.1 | 57.6 |
| | Group DRO | 78.6 | 65.5 | 77.1 | 65.1 | 76.0 | 64.5 | 76.0 | 66.8 | N/A | N/A | N/A | N/A | N/A | N/A | N/A | N/A |
| Imbalanced Learning Methods | SUBY | 78.2 | 65.0 | 76.3 | 64.6 | 76.6 | 64.7 | 74.7 | 65.0 | 86.5 | 64.2 | 93.7 | 58.8 | 95.2 | 58.5 | 91.8 | 65.6 |
| | RWY | 78.5 | 64.7 | 76.7 | 64.5 | 76.5 | 65.6 | 74.5 | 64.6 | 86.9 | 64.8 | 93.4 | 58.3 | 95.3 | 58.5 | 92.5 | 67.5 |
| | SUBG | 80.2 | 68.8 | 79.2 | 68.9 | 79.5 | 67.0 | 78.6 | 69.2 | 86.3 | 66.4 | 93.0 | 58.1 | 95.1 | 58.4 | 91.1 | 67.4 |
| | RWG | 80.3 | 68.4 | 79.1 | 69.4 | 80.1 | 68.3 | 78.7 | 69.4 | 87.5 | 65.4 | 93.4 | 59.4 | 95.2 | 58.4 | 92.4 | 68.6 |
| | JTT | 78.3 | 64.7 | 76.5 | 64.7 | 77.6 | 66.6 | 75.1 | 65.6 | 86.5 | 65.9 | 92.7 | 58.6 | 94.9 | 58.6 | 91.8 | 68.2 |
| | DWR | 79.7 | 67.4 | 78.3 | 68.5 | 79.1 | 67.4 | 77.5 | 69.0 | 87.2 | 65.8 | 93.4 | 58.1 | 95.3 | 58.5 | 92.3 | 67.6 |
| Fairness Methods | DP | 80.5 | 69.2 | 78.8 | 68.4 | 80.0 | 68.2 | 78.3 | 68.7 | 86.6 | 65.2 | 92.8 | 59.0 | 94.8 | 58.4 | 91.9 | 66.6 |
| | EO | 80.3 | 69.3 | 78.9 | 68.9 | 79.8 | 0.68 | 78.5 | 68.5 | 86.6 | 65.6 | 93.2 | 58.8 | 95.2 | 58.5 | 92.2 | 68.1 |
| | EP | 80.3 | 68.7 | 78.9 | 68.5 | 80.1 | 68.1 | 78.4 | 68.5 | 87.0 | 65.6 | 93.3 | 59.0 | 95.1 | 58.6 | 91.9 | 67.9 |
| | Exp | 80.5 | 69.0 | 78.9 | 68.6 | 81.0 | 68.8 | 78.4 | 69.0 | 85.9 | 65.4 | 90.1 | 58.7 | 93.1 | 56.5 | 86.3 | 64.7 |
| | Threshold | 80.5 | 69.0 | 78.6 | 69.2 | 78.2 | 66.7 | 78.5 | 68.7 | 84.3 | 64.5 | 88.1 | 59.5 | 91.2 | 55.7 | 82.0 | 64.0 |

**Table 11. Average target domain Macro-F1 score** of all algorithms on the `ACS Income` and `ACS Pub.Cov` datasets. In each setting, we test all algorithms on the 50 American states and report the mean and standard deviation across all 50 target states. The standard deviation here reflects the stability of performances across different target states.

| Dataset | | ACS Income | | | | | | | | ACS Pub.Cov | | | | | | | |
|---|---|---|---|---|---|---|---|---|---|---|---|---|---|---|---|---|---|
| Source State | | CA | | CT | | MA | | IN | | FL | | TX | | NE | | IN | |
| | | i.d. | o.o.d | i.d. | o.o.d | i.d. | o.o.d | i.d. | o.o.d | i.d. | o.o.d | i.d. | o.o.d | i.d. | o.o.d | i.d. | o.o.d |
| Base Methods | LR | 0.804 | 0.757 ±0.032 | 0.807 | 0.747 ±0.035 | 0.809 | 0.752 ±0.034 | 0.714 | 0.692 ±0.019 | 0.652 | 0.620 ±0.050 | 0.644 | 0.580 ±0.052 | 0.653 | 0.612 ±0.046 | 0.698 | 0.659 ±0.044 |
| | SVM | 0.795 | 0.726 ±0.037 | 0.790 | 0.709 ±0.039 | 0.793 | 0.718 ±0.041 | 0.698 | 0.664 ±0.023 | 0.663 | 0.623 ±0.044 | 0.665 | 0.614 ±0.044 | 0.698 | 0.611 ±0.043 | 0.696 | 0.650 ±0.032 |
| | MLP | 0.814 | 0.751 ±0.038 | 0.814 | 0.753 ±0.036 | 0.816 | 0.755 ±0.037 | 0.734 | 0.729 ±0.018 | 0.709 | 0.664 ±0.049 | 0.713 | 0.664 ±0.049 | 0.698 | 0.644 ±0.046 | 0.716 | 0.680 ±0.038 |
| Tree Ensemble Methods | Random Forest | 0.809 | 0.759 ±0.032 | 0.812 | 0.755 ±0.034 | 0.814 | 0.758 ±0.034 | 0.687 | 0.673 ±0.024 | 0.715 | 0.659 ±0.055 | 0.727 | 0.651 ±0.058 | 0.729 | 0.648 ±0.046 | 0.746 | 0.691 ±0.039 |
| | LightGBM | 0.811 | 0.757 ±0.035 | 0.821 | 0.753 ±0.037 | 0.823 | 0.759 ±0.037 | 0.732 | 0.718 ±0.021 | 0.730 | 0.670 ±0.053 | 0.739 | 0.665 ±0.057 | 0.745 | 0.663 ±0.048 | 0.750 | 0.704 ±0.034 |
| | XGBoost | 0.814 | 0.651 ±0.049 | 0.819 | 0.751 ±0.036 | 0.822 | 0.757 ±0.036 | 0.723 | 0.708 ±0.021 | 0.723 | 0.673 ±0.051 | 0.730 | 0.659 ±0.058 | 0.755 | 0.667 ±0.042 | 0.747 | 0.701 ±0.038 |
| DRO Methods | $\chi^2$-DRO | 0.813 | 0.754 ±0.036 | 0.812 | 0.752 ±0.036 | 0.816 | 0.748 ±0.039 | 0.732 | 0.737 ±0.019 | 0.704 | 0.659 ±0.044 | 0.697 | 0.657 ±0.046 | 0.685 | 0.646 ±0.038 | 0.715 | 0.673 ±0.030 |
| | CVaR-DRO | 0.808 | 0.750 ±0.038 | 0.812 | 0.749 ±0.036 | 0.820 | 0.752 ±0.037 | 0.730 | 0.726 ±0.021 | 0.693 | 0.649 ±0.046 | 0.706 | 0.655 ±0.049 | 0.668 | 0.658 ±0.021 | 0.700 | 0.675 ±0.036 |
| | $\chi^2$-DORO | 0.801 | 0.756 ±0.032 | 0.804 | 0.738 ±0.041 | 0.812 | 0.744 ±0.040 | 0.717 | 0.711 ±0.018 | 0.526 | 0.511 ±0.013 | 0.518 | 0.481 ±0.021 | 0.526 | 0.491 ±0.016 | 0.516 | 0.487 ±0.021 |
| | CVaR-DORO | 0.802 | 0.746 ±0.037 | 0.804 | 0.740 ±0.041 | 0.811 | 0.740 ±0.041 | 0.720 | 0.706 ±0.020 | 0.660 | 0.619 ±0.044 | 0.675 | 0.629 ±0.052 | 0.559 | 0.541 ±0.025 | 0.649 | 0.623 ±0.023 |
| | Group DRO | 0.812 | 0.760 ±0.033 | 0.813 | 0.754 ±0.035 | 0.817 | 0.753 ±0.037 | 0.741 | 0.742 ±0.018 | 0.706 | 0.665 ±0.051 | 0.717 | 0.660 ±0.050 | 0.700 | 0.635 ±0.042 | 0.419 | 0.411 ±0.030 |
| Imbalanced Learning Methods | SUBY | 0.807 | 0.739 ±0.041 | 0.815 | 0.748 ±0.039 | 0.821 | 0.750 ±0.040 | 0.719 | 0.739 ±0.025 | 0.710 | 0.699 ±0.023 | 0.710 | 0.692 ±0.027 | 0.704 | 0.682 ±0.031 | 0.748 | 0.710 ±0.025 |
| | RWY | 0.812 | 0.747 ±0.041 | 0.818 | 0.747 ±0.039 | 0.821 | 0.753 ±0.039 | 0.733 | 0.732 ±0.020 | 0.725 | 0.705 ±0.025 | 0.729 | 0.699 ±0.032 | 0.736 | 0.680 ±0.031 | 0.749 | 0.709 ±0.024 |
| | SUBG | 0.813 | 0.759 ±0.034 | 0.816 | 0.753 ±0.036 | 0.820 | 0.758 ±0.037 | 0.726 | 0.715 ±0.021 | 0.719 | 0.671 ±0.048 | 0.731 | 0.662 ±0.053 | 0.739 | 0.666 ±0.041 | 0.742 | 0.686 ±0.034 |
| | RWG | 0.814 | 0.760 ±0.035 | 0.819 | 0.754 ±0.036 | 0.822 | 0.757 ±0.036 | 0.726 | 0.723 ±0.019 | 0.723 | 0.667 ±0.051 | 0.733 | 0.661 ±0.057 | 0.748 | 0.672 ±0.045 | 0.746 | 0.700 ±0.035 |
| | JTT | 0.777 | 0.728 ±0.036 | 0.781 | 0.733 ±0.030 | 0.79 | 0.739 ±0.031 | 0.710 | 0.702 ±0.017 | 0.711 | 0.690 ±0.027 | 0.726 | 0.691 ±0.038 | 0.727 | 0.669 ±0.036 | 0.718 | 0.669 ±0.031 |
| | DWR | 0.812 | 0.759 ±0.035 | 0.816 | 0.755 ±0.036 | 0.821 | 0.756 ±0.037 | 0.733 | 0.719 ±0.019 | 0.721 | 0.668 ±0.047 | 0.731 | 0.661 ±0.057 | 0.732 | 0.676 ±0.041 | 0.741 | 0.697 ±0.033 |
| Fairness Methods | DP | 0.807 | 0.746 ±0.037 | 0.810 | 0.740 ±0.038 | 0.817 | 0.752 ±0.037 | 0.716 | 0.708 ±0.017 | 0.727 | 0.677 ±0.049 | 0.729 | 0.659 ±0.054 | 0.750 | 0.668 ±0.043 | 0.742 | 0.698 ±0.032 |
| | EO | 0.813 | 0.757 ±0.035 | 0.818 | 0.752 ±0.036 | 0.820 | 0.757 ±0.037 | 0.721 | 0.709 ±0.020 | 0.724 | 0.668 ±0.054 | 0.734 | 0.661 ±0.052 | 0.737 | 0.671 ±0.044 | 0.748 | 0.704 ±0.031 |
| | EP | 0.811 | 0.759 ±0.035 | 0.819 | 0.752 ±0.037 | 0.821 | 0.756 ±0.037 | 0.723 | 0.709 ±0.018 | 0.724 | 0.668 ±0.051 | 0.731 | 0.657 ±0.058 | 0.747 | 0.671 ±0.039 | 0.748 | 0.699 ±0.034 |
| | Exp | 0.808 | 0.750 ±0.037 | 0.810 | 0.737 ±0.038 | 0.818 | 0.749 ±0.037 | 0.706 | 0.691 ±0.021 | 0.724 | 0.667 ±0.053 | 0.729 | 0.656 ±0.057 | 0.738 | 0.661 ±0.049 | 0.741 | 0.699 ±0.033 |
| | Threshold | 0.766 | 0.709 ±0.037 | 0.768 | 0.718 ±0.032 | 0.771 | 0.716 ±0.028 | 0.665 | 0.669 ±0.019 | 0.722 | 0.669 ±0.050 | 0.717 | 0.652 ±0.056 | 0.714 | 0.644 ±0.040 | 0.728 | 0.681 ±0.031 |

**Table 12. Average target domain Macro-F1 score** of all algorithms on the `ACS Mobility` and `US Accident` datasets. In `ACS Mobility`, we test all algorithms on the 50 American states and report the mean and standard deviation across all 50 target states. And in `US Accident`, we choose 13 American states with relatively high sample sizes in testing. The standard deviation here reflects the stability of performances across different target states.

| Dataset | | ACS Mobility | | | | | | | | US Accident | | | | | | | |
|---|---|---|---|---|---|---|---|---|---|---|---|---|---|---|---|---|---|
| Source State | | NY | | CA | | MS | | PA | | CA | | FL | | TX | | MN | |
| | | i.d. | o.o.d | i.d. | o.o.d | i.d. | o.o.d | i.d. | o.o.d | i.d. | o.o.d | i.d. | o.o.d | i.d. | o.o.d | i.d. | o.o.d |
| Base Methods | LR | 0.451 | 0.427 ±.015 | 0.462 | 0.446 ±.026 | 0.531 | 0.488 ±.020 | 0.527 | 0.488 ±.020 | 0.800 | 0.678 ±.147 | 0.898 | 0.703 ±.133 | 0.936 | 0.736 ±.092 | 0.855 | 0.703 ±.110 |
| | SVM | 0.604 | 0.572 ±.015 | 0.592 | 0.550 ±.014 | 0.635 | 0.532 ±.018 | 0.623 | 0.581 ±.018 | 0.793 | 0.674 ±.114 | 0.891 | 0.698 ±.093 | 0.928 | 0.698 ±.093 | 0.880 | 0.669 ±.078 |
| | MLP | 0.600 | 0.583 ±.020 | 0.622 | 0.607 ±.018 | 0.607 | 0.583 ±.012 | 0.608 | 0.578 ±.020 | 0.827 | 0.755 ±.103 | 0.923 | 0.769 ±.090 | 0.934 | 0.720 ±.094 | 0.886 | 0.713 ±.096 |
| Tree Ensemble Methods | Random Forest | 0.570 | 0.530 ±.019 | 0.587 | 0.563 ±.023 | 0.641 | 0.534 ±.022 | 0.635 | 0.561 ±.025 | 0.846 | 0.752 ±.077 | 0.932 | 0.832 ±.096 | 0.938 | 0.834 ±.095 | 0.911 | 0.800 ±.058 |
| | LightGBM | 0.637 | 0.589 ±.014 | 0.625 | 0.601 ±.019 | 0.671 | 0.590 ±.021 | 0.660 | 0.590 ±.015 | 0.851 | 0.768 ±.075 | 0.929 | 0.831 ±.094 | 0.940 | 0.833 ±.096 | 0.918 | 0.786 ±.054 |
| | XGBoost | 0.637 | 0.591 ±.018 | 0.625 | 0.605 ±.018 | 0.678 | 0.586 ±.023 | 0.664 | 0.596 ±.017 | 0.852 | 0.786 ±.072 | 0.942 | 0.812 ±.088 | 0.939 | 0.833 ±.095 | 0.920 | 0.771 ±.052 |
| DRO Methods | $\chi^2$-DRO | 0.613 | 0.613 ±.019 | 0.599 | 0.579 ±.015 | 0.540 | 0.514 ±.031 | 0.606 | 0.583 ±.017 | 0.824 | 0.754 ±.099 | 0.922 | 0.721 ±.121 | 0.933 | 0.747 ±.091 | 0.883 | 0.704 ±.095 |
| | CVaR-DRO | 0.621 | 0.612 ±.017 | 0.628 | 0.603 ±.015 | 0.574 | 0.565 ±.025 | 0.639 | 0.613 ±.021 | 0.825 | 0.742 ±.094 | 0.924 | 0.731 ±.111 | 0.936 | 0.740 ±.090 | 0.888 | 0.716 ±.092 |
| | $\chi^2$-DORO | 0.509 | 0.501 ±.011 | 0.619 | 0.612 ±.018 | 0.530 | 0.505 ±.013 | 0.540 | 0.508 ±.014 | 0.815 | 0.707 ±.104 | 0.920 | 0.760 ±.090 | 0.927 | 0.734 ±.092 | 0.874 | 0.727 ±.080 |
| | CVaR-DORO | 0.533 | 0.524 ±.011 | 0.625 | 0.578 ±.019 | 0.521 | 0.500 ±.008 | 0.510 | 0.477 ±.016 | 0.825 | 0.748 ±.090 | 0.922 | 0.732 ±.110 | 0.934 | 0.738 ±.092 | 0.887 | 0.729 ±.094 |
| | Group DRO | 0.588 | 0.567 ±.018 | 0.566 | 0.528 ±.020 | 0.564 | 0.536 ±.019 | 0.595 | 0.555 ±.021 | N/A | N/A | N/A | N/A | N/A | N/A | N/A | N/A |
| Imbalanced Learning Methods | SUBY | 0.657 | 0.628 ±.017 | 0.649 | 0.603 ±.017 | 0.674 | 0.621 ±.016 | 0.678 | 0.629 ±.019 | 0.854 | 0.839 ±.073 | 0.933 | 0.834 ±.095 | 0.938 | 0.833 ±.095 | 0.915 | 0.821 ±.066 |
| | RWY | 0.657 | 0.627 ±.017 | 0.647 | 0.610 ±.014 | 0.678 | 0.621 ±.017 | 0.677 | 0.633 ±.018 | 0.856 | 0.845 ±.076 | 0.930 | 0.834 ±.097 | 0.940 | 0.833 ±.095 | 0.920 | 0.790 ±.055 |
| | SUBG | 0.635 | 0.589 ±.016 | 0.624 | 0.601 ±.020 | 0.672 | 0.595 ±.020 | 0.658 | 0.592 ±.014 | 0.847 | 0.790 ±.073 | 0.925 | 0.826 ±.094 | 0.937 | 0.833 ±.095 | 0.906 | 0.784 ±.057 |
| | RWG | 0.637 | 0.587 ±.018 | 0.621 | 0.584 ±.016 | 0.676 | 0.582 ±.018 | 0.663 | 0.596 ±.018 | 0.855 | 0.775 ±.074 | 0.929 | 0.825 ±.089 | 0.938 | 0.833 ±.095 | 0.919 | 0.768 ±.051 |
| | JTT | 0.639 | 0.594 ±.014 | 0.616 | 0.574 ±.013 | 0.660 | 0.589 ±.019 | 0.639 | 0.584 ±.017 | 0.843 | 0.786 ±.074 | 0.922 | 0.823 ±.091 | 0.935 | 0.833 ±.095 | 0.912 | 0.766 ±.052 |
| | DWR | 0.622 | 0.581 ±.013 | 0.620 | 0.605 ±.020 | 0.656 | 0.585 ±.018 | 0.647 | 0.597 ±.018 | 0.853 | 0.777 ±.075 | 0.929 | 0.824 ±.094 | 0.939 | 0.833 ±.094 | 0.917 | 0.759 ±.054 |
| Fairness Methods | DP | 0.630 | 0.588 ±.014 | 0.621 | 0.590 ±.014 | 0.663 | 0.590 ±.020 | 0.668 | 0.593 ±.018 | 0.847 | 0.776 ±.073 | 0.922 | 0.814 ±.085 | 0.933 | 0.832 ±.095 | 0.915 | 0.799 ±.059 |
| | EO | 0.632 | 0.590 ±.018 | 0.625 | 0.602 ±.019 | 0.665 | 0.593 ±.022 | 0.662 | 0.597 ±.020 | 0.846 | 0.771 ±.077 | 0.926 | 0.731 ±.091 | 0.938 | 0.833 ±.096 | 0.917 | 0.772 ±.051 |
| | EP | 0.633 | 0.596 ±.017 | 0.620 | 0.598 ±.017 | 0.671 | 0.581 ±.015 | 0.660 | 0.593 ±.017 | 0.852 | 0.787 ±.071 | 0.928 | 0.830 ±.094 | 0.937 | 0.832 ±.097 | 0.914 | 0.773 ±.049 |
| | Exp | 0.633 | 0.589 ±.015 | 0.622 | 0.594 ±.020 | 0.669 | 0.595 ±.020 | 0.663 | 0.595 ±.016 | 0.837 | 0.769 ±.079 | 0.893 | 0.808 ±.085 | 0.909 | 0.803 ±.091 | 0.854 | 0.784 ±.060 |
| | Threshold | 0.629 | 0.587 ±.013 | 0.620 | 0.596 ±.019 | 0.661 | 0.571 ±.019 | 0.658 | 0.597 ±.022 | 0.820 | 0.755 ±.069 | 0.870 | 0.782 ±.074 | 0.885 | 0.787 ±.086 | 0.809 | 0.730 ±.045 |

**Table 13. Average target domain Macro-F1 score** of all algorithms on the `Taxi`, `ACS Pub.Cov` (Temporal) and `ACS Income` (Synthetic) datasets. In `Taxi` and `ACS Pub.Cov` (Temporal), we test all algorithms on the 3 target domains and report the mean and standard deviation across all 3 target domains. The standard deviation here reflects the stability of performances across different target states. In `ACS Income` (Synthetic), we simulate strong covariate shifts according to the "Age" feature, where 10% and 20% means the minor group ratio in training, respectively.

| Dataset | | Taxi | | | | ACS Pub.Cov (Temporal) | | | | ACS Income (Synthetic) | | | |
|---|---|---|---|---|---|---|---|---|---|---|---|---|---|
| Source | | BOG | | NYC | | NY | | CA | | 20% | | 10% | |
| | | i.d. | o.o.d | i.d. | o.o.d | i.d. | o.o.d | i.d. | o.o.d | i.d. | o.o.d | i.d. | o.o.d |
| Base Methods | LR | 0.754 | 0.695 ±.007 | 0.802 | 0.731 ±.014 | 0.641 | 0.599 ±.009 | 0.586 | 0.528 ±.024 | 0.797 | 0.799 | 0.766 | 0.780 |
| | SVM | 0.757 | 0.761 ±.036 | 0.809 | 0.735 ±.019 | 0.663 | 0.613 ±.013 | 0.612 | 0.556 ±.016 | 0.795 | 0.764 | N/A | N/A |
| | MLP | 0.757 | 0.684 ±.012 | 0.823 | 0.719 ±.008 | 0.692 | 0.652 ±.012 | 0.674 | 0.587 ±.030 | 0.817 | 0.796 | 0.792 | 0.762 |
| Tree Ensemble Methods | Random Forest | 0.760 | 0.762 ±.032 | 0.840 | 0.718 ±.018 | 0.692 | 0.648 ±.016 | 0.663 | 0.571 ±.032 | 0.810 | 0.797 | 0.782 | 0.777 |
| | LightGBM | 0.866 | 0.717 ±.029 | 0.841 | 0.701 ±.011 | 0.711 | 0.668 ±.016 | 0.683 | 0.595 ±.027 | 0.817 | 0.813 | 0.787 | 0.788 |
| | XGBoost | 0.839 | 0.695 ±.015 | 0.846 | 0.723 ±.017 | 0.715 | 0.679 ±.014 | 0.689 | 0.591 ±.034 | 0.815 | 0.801 | 0.787 | 0.788 |
| DRO Methods | $\chi^2$-DRO | 0.776 | 0.693 ±.016 | 0.830 | 0.726 ±.009 | 0.675 | 0.633 ±.016 | 0.663 | 0.580 ±.031 | 0.820 | 0.788 | 0.790 | 0.783 |
| | CVaR-DRO | 0.767 | 0.744 ±.032 | 0.835 | 0.732 ±.010 | 0.668 | 0.625 ±.014 | 0.654 | 0.586 ±.025 | 0.820 | 0.799 | 0.791 | 0.783 |
| | $\chi^2$-DORO | 0.750 | 0.752 ±.027 | 0.816 | 0.732 ±.009 | 0.568 | 0.540 ±.014 | 0.531 | 0.501 ±.010 | 0.810 | 0.796 | 0.778 | 0.766 |
| | CVaR-DORO | 0.756 | 0.724 ±.023 | 0.829 | 0.730 ±.006 | 0.652 | 0.610 ±.017 | 0.618 | 0.556 ±.021 | 0.812 | 0.781 | 0.778 | 0.763 |
| | Group DRO | N/A | N/A | N/A | N/A | 0.682 | 0.655 ±.010 | 0.667 | 0.605 ±.020 | 0.814 | 0.798 | 0.795 | 0.763 |
| Imbalanced Learning Methods | SUBY | 0.859 | 0.669 ±.014 | 0.835 | 0.708 ±.014 | 0.706 | 0.690 ±.009 | 0.678 | 0.660 ±.010 | 0.774 | 0.699 | 0.716 | 0.562 |
| | RWY | 0.811 | 0.737 ±.033 | 0.840 | 0.713 ±.017 | 0.716 | 0.703 ±.009 | 0.693 | 0.656 ±.019 | 0.795 | 0.780 | 0.768 | 0.765 |
| | SUBG | 0.858 | 0.712 ±.032 | 0.837 | 0.701 ±.013 | 0.707 | 0.692 ±.013 | 0.687 | 0.588 ±.033 | 0.816 | 0.802 | 0.788 | 0.783 |
| | RWG | 0.802 | 0.731 ±.035 | 0.840 | 0.720 ±.018 | 0.713 | 0.654 ±.015 | 0.686 | 0.601 ±.034 | 0.816 | 0.810 | 0.798 | 0.791 |
| | JTT | 0.842 | 0.686 ±.003 | 0.830 | 0.704 ±.017 | 0.677 | 0.654 ±.011 | 0.676 | 0.641 ±.016 | 0.801 | 0.780 | 0.772 | 0.750 |
| | DWR | 0.845 | 0.671 ±.016 | 0.813 | 0.673 ±.015 | 0.713 | 0.675 ±.017 | 0.685 | 0.604 ±.029 | 0.817 | 0.812 | 0.785 | 0.785 |
| Fairness Methods | DP | 0.842 | 0.696 ±.020 | 0.843 | 0.702 ±.014 | 0.712 | 0.671 ±.016 | 0.689 | 0.600 ±.031 | 0.815 | 0.813 | 0.785 | 0.795 |
| | EO | 0.813 | 0.727 ±.028 | 0.846 | 0.706 ±.015 | 0.713 | 0.670 ±.018 | 0.686 | 0.586 ±.031 | 0.814 | 0.798 | 0.782 | 0.789 |
| | EP | 0.801 | 0.728 ±.027 | 0.835 | 0.710 ±.018 | 0.712 | 0.671 ±.016 | 0.687 | 0.597 ±.031 | 0.814 | 0.815 | 0.786 | 0.786 |
| | Exp | 0.797 | 0.724 ±.034 | 0.838 | 0.702 ±.015 | 0.712 | 0.648 ±.018 | 0.685 | 0.597 ±.033 | 0.814 | 0.798 | 0.783 | 0.783 |
| | Threshold | 0.806 | 0.660 ±.003 | 0.812 | 0.536 ±.027 | 0.711 | 0.670 ±.018 | 0.688 | 0.605 ±.030 | 0.785 | 0.786 | 0.752 | 0.761 |

**Table 14. Worst target domain Macro-F1 Score** of all algorithms on the `ACS Income` and `ACS Pub.Cov` datasets. In each setting, we test all algorithms on the 50 American states and report the worst Macro-F1 Score among all 50 target states.

| Dataset | | ACS INCOME | | | | | | | | ACS PUB.COV | | | | | | | |
| --- | --- | --- | --- | --- | --- | --- | --- | --- | --- | --- | --- | --- | --- | --- | --- | --- | --- |
| Source State | | CA | | CT | | MA | | SD | | FL | | TX | | NE | | IN | |
| | | i.d. | o.o.d | i.d. | o.o.d | i.d. | o.o.d | i.d. | o.o.d | i.d. | o.o.d | i.d. | o.o.d | i.d. | o.o.d | i.d. | o.o.d |
| Base Methods | LR | 0.804 | 0.610 | 0.807 | 0.597 | 0.809 | 0.599 | 0.714 | 0.644 | 0.652 | 0.478 | 0.644 | 0.432 | 0.653 | 0.468 | 0.698 | 0.514 |
| | SVM | 0.795 | 0.539 | 0.790 | 0.505 | 0.793 | 0.521 | 0.698 | 0.599 | 0.663 | 0.493 | 0.665 | 0.475 | 0.698 | 0.457 | 0.696 | 0.559 |
| | MLP | 0.814 | 0.574 | 0.814 | 0.591 | 0.816 | 0.578 | 0.734 | 0.669 | 0.709 | 0.527 | 0.713 | 0.518 | 0.698 | 0.496 | 0.716 | 0.569 |
| Tree Ensemble Methods | Random Forest | 0.809 | 0.608 | 0.812 | 0.599 | 0.814 | 0.595 | 0.687 | 0.600 | 0.715 | 0.503 | 0.727 | 0.484 | 0.729 | 0.492 | 0.746 | 0.573 |
| | LightGBM | 0.811 | 0.595 | 0.821 | 0.584 | 0.823 | 0.586 | 0.732 | 0.644 | 0.730 | 0.545 | 0.739 | 0.501 | 0.745 | 0.474 | 0.750 | 0.601 |
| | XGBoost | 0.814 | 0.602 | 0.819 | 0.588 | 0.822 | 0.591 | 0.723 | 0.632 | 0.723 | 0.547 | 0.730 | 0.489 | 0.755 | 0.525 | 0.747 | 0.580 |
| DRO Methods | $\chi^2$-DRO | 0.813 | 0.583 | 0.812 | 0.584 | 0.816 | 0.570 | 0.732 | 0.669 | 0.704 | 0.536 | 0.697 | 0.514 | 0.685 | 0.544 | 0.715 | 0.588 |
| | CVaR-DRO | 0.808 | 0.583 | 0.812 | 0.591 | 0.820 | 0.584 | 0.730 | 0.643 | 0.693 | 0.517 | 0.706 | 0.509 | 0.668 | 0.603 | 0.700 | 0.549 |
| | $\chi^2$-DORO | 0.801 | 0.588 | 0.804 | 0.535 | 0.812 | 0.543 | 0.717 | 0.662 | 0.526 | 0.476 | 0.518 | 0.431 | 0.526 | 0.449 | 0.516 | 0.419 |
| | CVaR-DORO | 0.802 | 0.560 | 0.804 | 0.543 | 0.811 | 0.539 | 0.720 | 0.655 | 0.660 | 0.481 | 0.675 | 0.474 | 0.559 | 0.444 | 0.649 | 0.562 |
| | Group DRO | 0.812 | 0.605 | 0.813 | 0.596 | 0.817 | 0.580 | 0.741 | 0.677 | 0.706 | 0.511 | 0.717 | 0.513 | 0.700 | 0.504 | 0.419 | 0.304 |
| Imbalanced Learning Methods | SUBY | 0.807 | 0.562 | 0.815 | 0.574 | 0.820 | 0.559 | 0.719 | 0.606 | 0.710 | 0.642 | 0.710 | 0.609 | 0.704 | 0.616 | 0.748 | 0.660 |
| | RWY | 0.812 | 0.553 | 0.818 | 0.576 | 0.821 | 0.569 | 0.733 | 0.645 | 0.725 | 0.646 | 0.729 | 0.602 | 0.736 | 0.555 | 0.749 | 0.654 |
| | SUBG | 0.813 | 0.603 | 0.816 | 0.588 | 0.820 | 0.584 | 0.726 | 0.637 | 0.719 | 0.566 | 0.731 | 0.552 | 0.739 | 0.517 | 0.742 | 0.542 |
| | RWG | 0.814 | 0.597 | 0.819 | 0.593 | 0.822 | 0.588 | 0.726 | 0.659 | 0.723 | 0.542 | 0.733 | 0.485 | 0.748 | 0.515 | 0.746 | 0.591 |
| | JTT | 0.777 | 0.576 | 0.781 | 0.601 | 0.790 | 0.602 | 0.710 | 0.649 | 0.711 | 0.618 | 0.726 | 0.574 | 0.727 | 0.525 | 0.718 | 0.548 |
| | DWR | 0.812 | 0.595 | 0.816 | 0.597 | 0.821 | 0.579 | 0.733 | 0.658 | 0.721 | 0.540 | 0.731 | 0.492 | 0.732 | 0.562 | 0.741 | 0.608 |
| Fairness Methods | DP | 0.807 | 0.585 | 0.810 | 0.582 | 0.817 | 0.593 | 0.716 | 0.658 | 0.727 | 0.569 | 0.729 | 0.507 | 0.750 | 0.488 | 0.742 | 0.587 |
| | EO | 0.813 | 0.599 | 0.818 | 0.590 | 0.820 | 0.591 | 0.721 | 0.632 | 0.724 | 0.544 | 0.734 | 0.516 | 0.737 | 0.514 | 0.748 | 0.601 |
| | EP | 0.811 | 0.596 | 0.819 | 0.588 | 0.821 | 0.580 | 0.723 | 0.646 | 0.724 | 0.561 | 0.731 | 0.487 | 0.747 | 0.539 | 0.748 | 0.589 |
| | Exp | 0.808 | 0.586 | 0.810 | 0.579 | 0.818 | 0.585 | 0.706 | 0.626 | 0.724 | 0.551 | 0.729 | 0.484 | 0.738 | 0.476 | 0.741 | 0.611 |
| | Threshold | 0.766 | 0.548 | 0.768 | 0.570 | 0.771 | 0.590 | 0.665 | 0.611 | 0.722 | 0.548 | 0.717 | 0.493 | 0.714 | 0.504 | 0.728 | 0.568 |

**Table 15. Worst target domain Macro-F1 Score** of all algorithms on the `ACS Mobility` and `US Accident` datasets. In `ACS Mobility`, we test all algorithms on the 50 American states and report the worst Macro-F1 Score among all 50 target states. And in `US Accident`, we choose 13 American states with relatively high sample sizes in testing.

| Dataset | | ACS Mobility | | | | | | | | US Accident | | | | | | | |
| --- | --- | --- | --- | --- | --- | --- | --- | --- | --- | --- | --- | --- | --- | --- | --- | --- | --- |
| Source State | | NY | | CA | | MS | | PA | | CA | | FL | | TX | | MN | |
| | | i.d. | o.o.d | i.d. | o.o.d | i.d. | o.o.d | i.d. | o.o.d | i.d. | o.o.d | i.d. | o.o.d | i.d. | o.o.d | i.d. | o.o.d |
| Base Methods | LR | 0.451 | 0.397 | 0.462 | 0.399 | 0.531 | 0.450 | 0.527 | 0.449 | 0.800 | 0.437 | 0.898 | 0.489 | 0.936 | 0.573 | 0.855 | 0.528 |
| | SVM | 0.604 | 0.544 | 0.592 | 0.527 | 0.635 | 0.489 | 0.623 | 0.539 | 0.793 | 0.505 | 0.891 | 0.568 | 0.928 | 0.542 | 0.880 | 0.540 |
| | MLP | 0.600 | 0.530 | 0.622 | 0.560 | 0.607 | 0.551 | 0.608 | 0.535 | 0.827 | 0.592 | 0.923 | 0.595 | 0.934 | 0.567 | 0.886 | 0.548 |
| Tree Ensemble Methods | Random Forest | 0.570 | 0.482 | 0.587 | 0.514 | 0.641 | 0.492 | 0.635 | 0.510 | 0.846 | 0.613 | 0.932 | 0.579 | 0.938 | 0.584 | 0.911 | 0.642 |
| | LightGBM | 0.637 | 0.554 | 0.625 | 0.545 | 0.671 | 0.531 | 0.660 | 0.559 | 0.851 | 0.617 | 0.929 | 0.582 | 0.940 | 0.584 | 0.918 | 0.639 |
| | XGBoost | 0.637 | 0.557 | 0.625 | 0.566 | 0.678 | 0.521 | 0.664 | 0.560 | 0.852 | 0.620 | 0.942 | 0.582 | 0.939 | 0.585 | 0.920 | 0.636 |
| DRO Methods | $\chi^2$-DRO | 0.613 | 0.571 | 0.599 | 0.539 | 0.540 | 0.374 | 0.606 | 0.539 | 0.824 | 0.596 | 0.922 | 0.534 | 0.933 | 0.573 | 0.883 | 0.534 |
| | CVaR-DRO | 0.621 | 0.551 | 0.628 | 0.557 | 0.574 | 0.512 | 0.639 | 0.518 | 0.825 | 0.591 | 0.924 | 0.572 | 0.936 | 0.582 | 0.888 | 0.561 |
| | $\chi^2$-DORO | 0.509 | 0.464 | 0.619 | 0.568 | 0.530 | 0.474 | 0.540 | 0.481 | 0.815 | 0.540 | 0.920 | 0.592 | 0.927 | 0.569 | 0.874 | 0.590 |
| | CVaR-DORO | 0.533 | 0.500 | 0.625 | 0.531 | 0.521 | 0.479 | 0.510 | 0.443 | 0.825 | 0.607 | 0.922 | 0.569 | 0.934 | 0.575 | 0.887 | 0.571 |
| | Group DRO | 0.588 | 0.516 | 0.566 | 0.480 | 0.564 | 0.491 | 0.595 | 0.508 | N/A | N/A | N/A | N/A | N/A | N/A | N/A | N/A |
| Imbalanced Learning Methods | SUBY | 0.657 | 0.590 | 0.649 | 0.548 | 0.674 | 0.565 | 0.678 | 0.568 | 0.854 | 0.623 | 0.933 | 0.585 | 0.938 | 0.582 | 0.915 | 0.635 |
| | RWY | 0.657 | 0.574 | 0.647 | 0.573 | 0.678 | 0.557 | 0.677 | 0.576 | 0.856 | 0.618 | 0.930 | 0.581 | 0.940 | 0.582 | 0.920 | 0.637 |
| | SUBG | 0.635 | 0.546 | 0.624 | 0.542 | 0.672 | 0.525 | 0.658 | 0.552 | 0.847 | 0.627 | 0.925 | 0.578 | 0.937 | 0.582 | 0.906 | 0.635 |
| | RWG | 0.637 | 0.542 | 0.621 | 0.553 | 0.676 | 0.521 | 0.663 | 0.539 | 0.855 | 0.614 | 0.929 | 0.588 | 0.938 | 0.583 | 0.919 | 0.635 |
| | JTT | 0.639 | 0.555 | 0.616 | 0.546 | 0.660 | 0.536 | 0.639 | 0.528 | 0.843 | 0.611 | 0.922 | 0.581 | 0.935 | 0.583 | 0.912 | 0.635 |
| | DWR | 0.622 | 0.553 | 0.620 | 0.550 | 0.656 | 0.513 | 0.647 | 0.561 | 0.853 | 0.623 | 0.929 | 0.576 | 0.939 | 0.583 | 0.917 | 0.627 |
| Fairness Methods | DP | 0.630 | 0.555 | 0.621 | 0.556 | 0.663 | 0.536 | 0.668 | 0.557 | 0.847 | 0.618 | 0.922 | 0.590 | 0.933 | 0.582 | 0.915 | 0.630 |
| | EO | 0.632 | 0.552 | 0.625 | 0.547 | 0.665 | 0.549 | 0.662 | 0.543 | 0.846 | 0.614 | 0.926 | 0.584 | 0.938 | 0.583 | 0.917 | 0.634 |
| | EP | 0.633 | 0.550 | 0.620 | 0.552 | 0.671 | 0.529 | 0.660 | 0.563 | 0.852 | 0.621 | 0.920 | 0.587 | 0.937 | 0.583 | 0.914 | 0.636 |
| | Exp | 0.633 | 0.544 | 0.622 | 0.551 | 0.669 | 0.531 | 0.663 | 0.544 | 0.837 | 0.613 | 0.893 | 0.580 | 0.909 | 0.564 | 0.854 | 0.619 |
| | Threshold | 0.629 | 0.551 | 0.620 | 0.550 | 0.661 | 0.528 | 0.658 | 0.551 | 0.820 | 0.605 | 0.870 | 0.583 | 0.885 | 0.560 | 0.809 | 0.601 |

Table 15. For other settings, we do not report the worst-case results because there are not many target domains there and the mean/standard deviation results could reflect the generalization performance well.

For the results of Macro-f1 Score reported in Tables 11, 12, 13, 14 and 15, we find similar conclusion patterns to the accuracy score, while we usually witness a wider gap between *o.o.d.* and *i.i.d.* model performance.

### D.10 Agreement and Maintenance Plan

**Hosting Platform.** We will use Github as the hosting platform of our code. We provide detailed preprocessing Python scripts to guide users to replicate and process data from scratch. We also illustrate methodologies to run full experiment results in the code. We also list the license for each dataset and user guidance in the `Readme` file in that Github.

**Dependencies.** The benchmark is built upon Python 3.8+ and depends on PyTorch, aif360, fairlearn, xgboost, lightgbm. Besides, it uses numpy, scipy, and pandas for basic data manipulation.

**Maintenance Plan.** The datasets provided here will be maintained by the authors of the paper, which can be contacted by raising an issue on GitHub or by contacting the first authors directly. Our benchmark and a simple open-sourced Python package based on that may be updated at the discretion of authors in the future, which includes more refined algorithm implementations, datasets, and framework with improved efficiency.

**Author Statements.** To the best of our knowledge, the proposed benchmark is based on existing datasets and does not violate any existing licenses non contain personally identifiable or privacy-related information. And we claim all the responsibility in case of a violation of rights if such a violation were to exist.

