| 0.804 | 0.757 $_{\pm 0.032}$ | 0.807 | 0.747 $_{\pm 0.035}$ | 0.809 | 0.752 $_{\pm 0.034}$ | 0.714 | 0.692 $_{\pm 0.019}$ | 0.652 | 0.620 $_{\pm 0.050}$ | 0.644 | 0.580 $_{\pm 0.052}$ | 0.653 | 0.612 $_{\pm 0.046}$ | 0.698 | 0.659 $_{\pm 0.044}$ |
| | SVM | 0.795 | 0.726 $_{\pm 0.037}$ | 0.790 | 0.709 $_{\pm 0.039}$ | 0.793 | 0.718 $_{\pm 0.041}$ | 0.698 | 0.664 $_{\pm 0.023}$ | 0.663 | 0.623 $_{\pm 0.044}$ | 0.665 | 0.614 $_{\pm 0.044}$ | 0.698 | 0.611 $_{\pm 0.043}$ | 0.696 | 0.650 $_{\pm 0.032}$ |
| | MLP | 0.814 | 0.751 $_{\pm 0.038}$ | 0.814 | 0.753 $_{\pm 0.036}$ | 0.816 | 0.755 $_{\pm 0.037}$ | 0.734 | 0.729 $_{\pm 0.018}$ | 0.709 | 0.664 $_{\pm 0.049}$ | 0.713 | 0.664 $_{\pm 0.049}$ | 0.698 | 0.644 $_{\pm 0.046}$ | 0.716 | 0.680 $_{\pm 0.038}$ |
| Tree Ensemble Methods | Random Forest | 0.809 | 0.759 $_{\pm 0.032}$ | 0.812 | 0.755 $_{\pm 0.034}$ | 0.814 | 0.758 $_{\pm 0.034}$ | 0.687 | 0.673 $_{\pm 0.024}$ | 0.715 | 0.659 $_{\pm 0.055}$ | 0.727 | 0.651 $_{\pm 0.058}$ | 0.729 | 0.648 $_{\pm 0.046}$ | 0.746 | 0.691 $_{\pm 0.039}$ |
| | LightGBM | 0.811 | 0.757 $_{\pm 0.035}$ | 0.821 | 0.753 $_{\pm 0.037}$ | 0.823 | 0.759 $_{\pm 0.037}$ | 0.732 | 0.718 $_{\pm 0.021}$ | 0.730 | 0.670 $_{\pm 0.053}$ | 0.739 | 0.665 $_{\pm 0.057}$ | 0.745 | 0.663 $_{\pm 0.048}$ | 0.750 | 0.704 $_{\pm 0.034}$ |
| | XGBoost | 0.814 | 0.651 $_{\pm 0.049}$ | 0.819 | 0.751 $_{\pm 0.036}$ | 0.822 | 0.757 $_{\pm 0.036}$ | 0.723 | 0.708 $_{\pm 0.021}$ | 0.723 | 0.673 $_{\pm 0.051}$ | 0.730 | 0.659 $_{\pm 0.058}$ | 0.755 | 0.667 $_{\pm 0.042}$ | 0.747 | 0.701 $_{\pm 0.038}$ |
| DRO Methods | $\chi^2$-DRO | 0.813 | 0.754 $_{\pm 0.036}$ | 0.812 | 0.752 $_{\pm 0.036}$ | 0.816 | 0.748 $_{\pm 0.039}$ | 0.732 | 0.737 $_{\pm 0.019}$ | 0.704 | 0.659 $_{\pm 0.044}$ | 0.697 | 0.657 $_{\pm 0.046}$ | 0.685 | 0.646 $_{\pm 0.038}$ | 0.715 | 0.673 $_{\pm 0.030}$ |
| | CVaR-DRO | 0.808 | 0.750 $_{\pm 0.038}$ | 0.812 | 0.749 $_{\pm 0.036}$ | 0.820 | 0.752 $_{\pm 0.037}$ | 0.730 | 0.726 $_{\pm 0.021}$ | 0.693 | 0.649 $_{\pm 0.046}$ | 0.706 | 0.655 $_{\pm 0.049}$ | 0.668 | 0.658 $_{\pm 0.021}$ | 0.700 | 0.675 $_{\pm 0.036}$ |
| | $\chi^2$-DORO | 0.801 | 0.756 $_{\pm 0.032}$ | 0.804 | 0.738 $_{\pm 0.041}$ | 0.812 | 0.744 $_{\pm 0.040}$ | 0.717 | 0.711 $_{\pm 0.018}$ | 0.526 | 0.511 $_{\pm 0.013}$ | 0.518 | 0.481 $_{\pm 0.021}$ | 0.526 | 0.491 $_{\pm 0.016}$ | 0.516 | 0.487 $_{\pm 0.021}$ |
| | CVaR-DORO | 0.802 | 0.746 $_{\pm 0.037}$ | 0.804 | 0.740 $_{\pm 0.041}$ | 0.811 | 0.740 $_{\pm 0.041}$ | 0.720 | 0.706 $_{\pm 0.020}$ | 0.660 | 0.619 $_{\pm 0.044}$ | 0.675 | 0.629 $_{\pm 0.052}$ | 0.559 | 0.541 $_{\pm 0.025}$ | 0.649 | 0.623 $_{\pm 0.023}$ |
| | Group DRO | 0.812 | 0.760 $_{\pm 0.033}$ | 0.813 | 0.754 $_{\pm 0.035}$ | 0.817 | 0.753 $_{\pm 0.037}$ | 0.741 | 0.742 $_{\pm 0.018}$ | 0.706 | 0.665 $_{\pm 0.051}$ | 0.717 | 0.660 $_{\pm 0.050}$ | 0.700 | 0.635 $_{\pm 0.042}$ | 0.419 | 0.411 $_{\pm 0.030}$ |
| Imbalanced Learning Methods | SUBY | 0.807 | 0.739 $_{\pm 0.041}$ | 0.815 | 0.748 $_{\pm 0.039}$ | 0.821 | 0.750 $_{\pm 0.040}$ | 0.719 | 0.739 $_{\pm 0.025}$ | 0.710 | 0.699 $_{\pm 0.023}$ | 0.710 | 0.692 $_{\pm 0.027}$ | 0.704 | 0.682 $_{\pm 0.031}$ | 0.748 | 0.710 $_{\pm 0.025}$ |
| | RWY | 0.812 | 0.747 $_{\pm 0.041}$ | 0.818 | 0.747 $_{\pm 0.039}$ | 0.820 | 0.753 $_{\pm 0.039}$ | 0.733 | 0.732 $_{\pm 0.020}$ | 0.725 | 0.705 $_{\pm 0.029}$ | 0.729 | 0.699 $_{\pm 0.032}$ | 0.736 | 0.680 $_{\pm 0.031}$ | 0.749 | 0.709 $_{\pm 0.024}$ |
| | SUBG | 0.813 | 0.759 $_{\pm 0.034}$ | 0.816 | 0.753 $_{\pm 0.036}$ | 0.820 | 0.758 $_{\pm 0.037}$ | 0.726 | 0.715 $_{\pm 0.021}$ | 0.719 | 0.671 $_{\pm 0.048}$ | 0.731 | 0.662 $_{\pm 0.053}$ | 0.739 | 0.666 $_{\pm 0.041}$ | 0.742 | 0.686 $_{\pm 0.034}$ |
| | RWG | 0.814 | 0.760 $_{\pm 0.035}$ | 0.819 | 0.754 $_{\pm 0.036}$ | 0.822 | 0.757 $_{\pm 0.036}$ | 0.726 | 0.723 $_{\pm 0.019}$ | 0.723 | 0.667 $_{\pm 0.051}$ | 0.733 | 0.661 $_{\pm 0.057}$ | 0.748 | 0.672 $_{\pm 0.045}$ | 0.746 | 0.700 $_{\pm 0.035}$ |
| | JTT | 0.777 | 0.728 $_{\pm 0.036}$ | 0.781 | 0.733 $_{\pm 0.030}$ | 0.79 | 0.739 $_{\pm 0.031}$ | 0.710 | 0.702 $_{\pm 0.017}$ | 0.711 | 0.690 $_{\pm 0.027}$ | 0.726 | 0.691 $_{\pm 0.038}$ | 0.727 | 0.669 $_{\pm 0.036}$ | 0.718 | 0.669 $_{\pm 0.031}$ |
| | DWR | 0.812 | 0.759 $_{\pm 0.035}$ | 0.816 | 0.755 $_{\pm 0.036}$ | 0.821 | 0.756 $_{\pm 0.037}$ | 0.733 | 0.719 $_{\pm 0.019}$ | 0.721 | 0.668 $_{\pm 0.047}$ | 0.731 | 0.661 $_{\pm 0.057}$ | 0.732 | 0.676 $_{\pm 0.041}$ | 0.741 | 0.697 $_{\pm 0.033}$ |
| Fairness Methods | DP | 0.807 | 0.746 $_{\pm 0.037}$ | 0.810 | 0.740 $_{\pm 0.038}$ | 0.817 | 0.752 $_{\pm 0.037}$ | 0.716 | 0.708 $_{\pm 0.017}$ | 0.727 | 0.677 $_{\pm 0.049}$ | 0.729 | 0.659 $_{\pm 0.054}$ | 0.750 | 0.668 $_{\pm 0.043}$ | 0.742 | 0.698 $_{\pm 0.032}$ |
| | EO | 0.813 | 0.757 $_{\pm 0.035}$ | 0.818 | 0.752 $_{\pm 0.036}$ | 0.820 | 0.757 $_{\pm 0.037}$ | 0.721 | 0.709 $_{\pm 0.020}$ | 0.724 | 0.668 $_{\pm 0.054}$ | 0.734 | 0.661 $_{\pm 0.052}$ | 0.737 | 0.671 $_{\pm 0.044}$ | 0.748 | 0.704 $_{\pm 0.031}$ |
| | EP | 0.811 | 0.759 $_{\pm 0.035}$ | 0.819 | 0.752 $_{\pm 0.037}$ | 0.821 | 0.756 $_{\pm 0.037}$ | 0.723 | 0.709 $_{\pm 0.018}$ | 0.724 | 0.668 $_{\pm 0.051}$ | 0.731 | 0.657 $_{\pm 0.058}$ | 0.747 | 0.671 $_{\pm 0.039}$ | 0.748 | 0.699 $_{\pm 0.034}$ |
| | Exp | 0.808 | 0.750 $_{\pm 0.037}$ | 0.810 | 0.737 $_{\pm 0.038}$ | 0.818 | 0.749 $_{\pm 0.037}$ | 0.706 | 0.691 $_{\pm 0.021}$ | 0.724 | 0.667 $_{\pm 0.053}$ | 0.729 | 0.656 $_{\pm 0.057}$ | 0.738 | 0.661 $_{\pm 0.049}$ | 0.741 | 0.699 $_{\pm 0.033}$ |
| | Threshold | 0.766 | 0.709 $_{\pm 0.037}$ | 0.768 | 0.718 $_{\pm 0.032}$ | 0.771 | 0.716 $_{\pm 0.028}$ | 0.665 | 0.669 $_{\pm 0.019}$ | 0.722 | 0.669 $_{\pm 0.050}$ | 0.717 | 0.652 $_{\pm 0.056}$ | 0.714 | 0.644 $_{\pm 0.040}$ | 0.728 | 0.681 $_{\pm 0.031}$ |

**Table 12. Average target domain Macro-F1 score** of all algorithms on the `ACS Mobility` and `US Accident` datasets. In `ACS Mobility`, we test all algorithms on the 50 American states and report the mean and standard deviation across all 50 target states. And in `US Accident`, we choose 13 American states with relatively high sample sizes in testing. The standard deviation here reflects the stability of performances across different target states.

| Dataset | | ACS Mobility | | | | | | | | US Accident | | | | | | | |
|---|---|---|---|---|---|---|---|---|---|---|---|---|---|---|---|---|---|
| Source State | | NY | | CA | | MS | | PA | | CA | | FL | | TX | | MN | |
| | | i.d. | o.o.d | i.d. | o.o.d | i.d. | o.o.d | i.d. | o.o.d | i.d. | o.o.d | i.d. | o.o.d | i.d. | o.o.d | i.d. | o.o.d |
| Base Methods | LR | 0.451 | $0.427_{\pm.015}$ | 0.462 | $0.446_{\pm.026}$ | 0.531 | $0.488_{\pm.020}$ | 0.527 | $0.488_{\pm.020}$ | 0.800 | $0.678_{\pm.147}$ | 0.898 | $0.703_{\pm.133}$ | 0.936 | $0.736_{\pm.092}$ | 0.855 | $0.703_{\pm.110}$ |
| | SVM | 0.604 | $0.572_{\pm.015}$ | 0.592 | $0.550_{\pm.014}$ | 0.635 | $0.532_{\pm.018}$ | 0.623 | $0.581_{\pm.018}$ | 0.793 | $0.674_{\pm.114}$ | 0.891 | $0.698_{\pm.093}$ | 0.928 | $0.698_{\pm.093}$ | 0.880 | $0.669_{\pm.078}$ |
| | MLP | 0.600 | $0.583_{\pm.020}$ | 0.622 | $0.607_{\pm.018}$ | 0.607 | $0.583_{\pm.012}$ | 0.608 | $0.578_{\pm.020}$ | 0.827 | $0.755_{\pm.103}$ | 0.923 | $0.769_{\pm.090}$ | 0.934 | $0.720_{\pm.094}$ | 0.886 | $0.713_{\pm.096}$ |
| Tree Ensemble Methods | Random Forest | 0.570 | $0.530_{\pm.019}$ | 0.587 | $0.563_{\pm.023}$ | 0.641 | $0.534_{\pm.022}$ | 0.635 | $0.561_{\pm.025}$ | 0.846 | $0.752_{\pm.077}$ | 0.932 | $0.832_{\pm.096}$ | 0.938 | $0.834_{\pm.095}$ | 0.911 | $0.800_{\pm.058}$ |
| | LightGBM | 0.637 | $0.589_{\pm.014}$ | 0.625 | $0.601_{\pm.019}$ | 0.671 | $0.590_{\pm.021}$ | 0.660 | $0.590_{\pm.015}$ | 0.851 | $0.768_{\pm.075}$ | 0.929 | $0.831_{\pm.094}$ | 0.940 | $0.833_{\pm.096}$ | 0.918 | $0.786_{\pm.054}$ |
| | XGBoost | 0.637 | $0.591_{\pm.018}$ | 0.625 | $0.605_{\pm.018}$ | 0.678 | $0.586_{\pm.023}$ | 0.664 | $0.596_{\pm.017}$ | 0.852 | $0.786_{\pm.072}$ | 0.942 | $0.812_{\pm.088}$ | 0.939 | $0.833_{\pm.095}$ | 0.920 | $0.771_{\pm.052}$ |
| DRO Methods | $\chi^2$-DRO | 0.613 | $0.613_{\pm.019}$ | 0.599 | $0.579_{\pm.015}$ | 0.540 | $0.514_{\pm.031}$ | 0.606 | $0.583_{\pm.017}$ | 0.824 | $0.754_{\pm.099}$ | 0.922 | $0.721_{\pm.121}$ | 0.933 | $0.747_{\pm.091}$ | 0.883 | $0.704_{\pm.095}$ |
| | CVaR-DRO | 0.621 | $0.612_{\pm.017}$ | 0.628 | $0.603_{\pm.015}$ | 0.574 | $0.565_{\pm.025}$ | 0.639 | $0.613_{\pm.021}$ | 0.825 | $0.742_{\pm.094}$ | 0.924 | $0.731_{\pm.111}$ | 0.936 | $0.740_{\pm.090}$ | 0.888 | $0.716_{\pm.092}$ |
| | $\chi^2$-DORO | 0.509 | $0.501_{\pm.011}$ | 0.619 | $0.612_{\pm.018}$ | 0.530 | $0.505_{\pm.013}$ | 0.540 | $0.508_{\pm.014}$ | 0.815 | $0.707_{\pm.104}$ | 0.920 | $0.760_{\pm.090}$ | 0.927 | $0.734_{\pm.092}$ | 0.874 | $0.727_{\pm.080}$ |
| | CVaR-DORO | 0.533 | $0.524_{\pm.011}$ | 0.625 | $0.578_{\pm.019}$ | 0.521 | $0.500_{\pm.008}$ | 0.510 | $0.477_{\pm.016}$ | 0.825 | $0.748_{\pm.090}$ | 0.922 | $0.732_{\pm.110}$ | 0.934 | $0.738_{\pm.092}$ | 0.887 | $0.729_{\pm.094}$ |
| | Group DRO | 0.588 | $0.567_{\pm.018}$ | 0.566 | $0.528_{\pm.020}$ | 0.564 | $0.536_{\pm.019}$ | 0.595 | $0.555_{\pm.021}$ | N/A | N/A | N/A | N/A | N/A | N/A | N/A | N/A |
| Imbalanced Learning Methods | SUBY | 0.657 | $0.628_{\pm.017}$ | 0.649 | $0.603_{\pm.017}$ | 0.674 | $0.621_{\pm.016}$ | 0.678 | $0.629_{\pm.019}$ | 0.854 | $0.839_{\pm.073}$ | 0.933 | $0.834_{\pm.095}$ | 0.938 | $0.833_{\pm.095}$ | 0.915 | $0.821_{\pm.066}$ |
| | RWY | 0.657 | $0.627_{\pm.017}$ | 0.647 | $0.610_{\pm.014}$ | 0.678 | $0.621_{\pm.017}$ | 0.677 | $0.633_{\pm.018}$ | 0.856 | $0.845_{\pm.076}$ | 0.930 | $0.834_{\pm.097}$ | 0.940 | $0.833_{\pm.095}$ | 0.920 | $0.790_{\pm.055}$ |
| | SUBG | 0.635 | $0.589_{\pm.016}$ | 0.624 | $0.601_{\pm.020}$ | 0.672 | $0.595_{\pm.020}$ | 0.658 | $0.592_{\pm.020}$ | 0.847 | $0.790_{\pm.073}$ | 0.925 | $0.826_{\pm.094}$ | 0.937 | $0.833_{\pm.095}$ | 0.906 | $0.784_{\pm.057}$ |
| | RWG | 0.637 | $0.587_{\pm.018}$ | 0.621 | $0.584_{\pm.016}$ | 0.676 | $0.582_{\pm.018}$ | 0.663 | $0.596_{\pm.018}$ | 0.855 | $0.775_{\pm.074}$ | 0.929 | $0.825_{\pm.089}$ | 0.938 | $0.833_{\pm.095}$ | 0.919 | $0.768_{\pm.051}$ |
| | JTT | 0.639 | $0.594_{\pm.014}$ | 0.616 | $0.574_{\pm.013}$ | 0.660 | $0.589_{\pm.019}$ | 0.639 | $0.584_{\pm.017}$ | 0.843 | $0.786_{\pm.074}$ | 0.922 | $0.823_{\pm.091}$ | 0.935 | $0.833_{\pm.095}$ | 0.912 | $0.766_{\pm.052}$ |
| | DWR | 0.622 | $0.581_{\pm.013}$ | 0.620 | $0.605_{\pm.020}$ | 0.656 | $0.585_{\pm.018}$ | 0.647 | $0.597_{\pm.018}$ | 0.853 | $0.777_{\pm.075}$ | 0.929 | $0.824_{\pm.094}$ | 0.939 | $0.833_{\pm.094}$ | 0.917 | $0.759_{\pm.054}$ |
| Fairness Methods | DP | 0.630 | $0.588_{\pm.014}$ | 0.621 | $0.590_{\pm.014}$ | 0.663 | $0.590_{\pm.020}$ | 0.668 | $0.593_{\pm.018}$ | 0.847 | $0.776_{\pm.073}$ | 0.922 | $0.814_{\pm.085}$ | 0.933 | $0.832_{\pm.095}$ | 0.915 | $0.799_{\pm.059}$ |
| | EO | 0.632 | $0.590_{\pm.018}$ | 0.625 | $0.602_{\pm.019}$ | 0.665 | $0.593_{\pm.022}$ | 0.662 | $0.597_{\pm.020}$ | 0.846 | $0.771_{\pm.077}$ | 0.926 | $0.731_{\pm.081}$ | 0.938 | $0.833_{\pm.096}$ | 0.917 | $0.772_{\pm.051}$ |
| | EP | 0.633 | $0.596_{\pm.017}$ | 0.620 | $0.598_{\pm.017}$ | 0.671 | $0.581_{\pm.015}$ | 0.660 | $0.593_{\pm.017}$ | 0.852 | $0.787_{\pm.071}$ | 0.928 | $0.830_{\pm.094}$ | 0.937 | $0.832_{\pm.097}$ | 0.914 | $0.773_{\pm.049}$ |
| | Exp | 0.633 | $0.589_{\pm.015}$ | 0.622 | $0.594_{\pm.020}$ | 0.669 | $0.595_{\pm.020}$ | 0.663 | $0.595_{\pm.016}$ | 0.837 | $0.769_{\pm.079}$ | 0.893 | $0.808_{\pm.085}$ | 0.909 | $0.803_{\pm.091}$ | 0.854 | $0.784_{\pm.060}$ |
| | Threshold | 0.629 | $0.587_{\pm.013}$ | 0.620 | $0.596_{\pm.019}$ | 0.661 | $0.571_{\pm.019}$ | 0.658 | $0.597_{\pm.022}$ | 0.820 | $0.755_{\pm.069}$ | 0.870 | $0.782_{\pm.074}$ | 0.885 | $0.787_{\pm.086}$ | 0.809 | $0.730_{\pm.045}$ |

**Table 13. Average target domain Macro-F1 score** of all algorithms on the `Taxi`, `ACS Pub.Cov` (Temporal) and `ACS Income` (Synthetic) datasets. In `Taxi` and `ACS Pub.Cov` (Temporal), we test all algorithms on the 3 target domains and report the mean and standard deviation across all 3 target domains. The standard deviation here reflects the stability of performances across different target states. In `ACS Income` (Synthetic), we simulate strong covariate shifts according to the "Age" feature, where 10% and 20% means the minor group ratio in training, respectively.

| Dataset | | Taxi | | | | ACS Pub.Cov (Temporal) | | | | ACS Income (Synthetic) | | | |
|---|---|---|---|---|---|---|---|---|---|---|---|---|---|
| Source | | BOG | | NYC | | NY | | CA | | 20% | | 10% | |
| | | i.d. | o.o.d | i.d. | o.o.d | i.d. | o.o.d | i.d. | o.o.d | i.d. | o.o.d | i.d. | o.o.d |
| Base Methods | LR | 0.754 | $0.695_{\pm.007}$ | 0.802 | $0.731_{\pm.014}$ | 0.641 | $0.599_{\pm.009}$ | 0.586 | $0.528_{\pm.024}$ | 0.797 | 0.799 | 0.766 | 0.780 |
| | SVM | 0.757 | $0.761_{\pm.036}$ | 0.809 | $0.735_{\pm.019}$ | 0.663 | $0.613_{\pm.013}$ | 0.612 | $0.556_{\pm.016}$ | 0.795 | 0.764 | N/A | N/A |
| | MLP | 0.757 | $0.684_{\pm.012}$ | 0.823 | $0.719_{\pm.008}$ | 0.692 | $0.652_{\pm.012}$ | 0.674 | $0.587_{\pm.030}$ | 0.817 | 0.796 | 0.792 | 0.762 |
| Tree Ensemble Methods | Random Forest | 0.760 | $0.762_{\pm.032}$ | 0.840 | $0.718_{\pm.018}$ | 0.692 | $0.648_{\pm.016}$ | 0.663 | $0.571_{\pm.032}$ | 0.810 | 0.797 | 0.782 | 0.777 |
| | LightGBM | 0.866 | $0.717_{\pm.029}$ | 0.841 | $0.701_{\pm.011}$ | 0.711 | $0.668_{\pm.016}$ | 0.683 | $0.595_{\pm.027}$ | 0.817 | 0.813 | 0.787 | 0.788 |
| | XGBoost | 0.839 | $0.695_{\pm.015}$ | 0.846 | $0.723_{\pm.017}$ | 0.715 | $0.679_{\pm.014}$ | 0.689 | $0.591_{\pm.034}$ | 0.815 | 0.801 | 0.787 | 0.788 |
| DRO Methods | $\chi^2$-DRO | 0.776 | $0.693_{\pm.016}$ | 0.830 | $0.726_{\pm.009}$ | 0.675 | $0.633_{\pm.016}$ | 0.663 | $0.580_{\pm.031}$ | 0.820 | 0.788 | 0.790 | 0.783 |
| | CVaR-DRO | 0.767 | $0.744_{\pm.032}$ | 0.835 | $0.732_{\pm.010}$ | 0.668 | $0.625_{\pm.014}$ | 0.654 | $0.586_{\pm.025}$ | 0.820 | 0.799 | 0.791 | 0.783 |
| | $\chi^2$-DORO | 0.750 | $0.752_{\pm.027}$ | 0.816 | $0.732_{\pm.009}$ | 0.568 | $0.540_{\pm.014}$ | 0.531 | $0.501_{\pm.010}$ | 0.810 | 0.796 | 0.778 | 0.766 |
| | CVaR-DORO | 0.756 | $0.724_{\pm.023}$ | 0.829 | $0.730_{\pm.006}$ | 0.652 | $0.610_{\pm.017}$ | 0.618 | $0.556_{\pm.021}$ | 0.812 | 0.781 | 0.778 | 0.763 |
| | Group DRO | N/A | N/A | N/A | N/A | 0.682 | $0.655_{\pm.010}$ | 0.667 | $0.605_{\pm.020}$ | 0.814 | 0.798 | 0.795 | 0.763 |
| Imbalanced Learning Methods | SUBY | 0.859 | $0.669_{\pm.014}$ | 0.835 | $0.708_{\pm.014}$ | 0.706 | $0.690_{\pm.009}$ | 0.678 | $0.660_{\pm.010}$ | 0.774 | 0.699 | 0.716 | 0.562 |
| | RWY | 0.811 | $0.737_{\pm.033}$ | 0.840 | $0.713_{\pm.017}$ | 0.716 | $0.703_{\pm.009}$ | 0.693 | $0.656_{\pm.019}$ | 0.795 | 0.780 | 0.768 | 0.765 |
| | SUBG | 0.858 | $0.712_{\pm.032}$ | 0.837 | $0.701_{\pm.013}$ | 0.707 | $0.692_{\pm.013}$ | 0.687 | $0.588_{\pm.033}$ | 0.816 | 0.802 | 0.788 | 0.783 |
| | RWG | 0.802 | $0.731_{\pm.035}$ | 0.840 | $0.720_{\pm.018}$ | 0.713 | $0.654_{\pm.015}$ | 0.686 | $0.601_{\pm.034}$ | 0.816 | 0.810 | 0.798 | 0.791 |
| | JTT | 0.842 | $0.686_{\pm.003}$ | 0.830 | $0.704_{\pm.017}$ | 0.677 | $0.654_{\pm.011}$ | 0.676 | $0.641_{\pm.016}$ | 0.801 | 0.780 | 0.772 | 0.750 |
| | DWR | 0.845 | $0.671_{\pm.016}$ | 0.813 | $0.673_{\pm.015}$ | 0.713 | $0.675_{\pm.017}$ | 0.685 | $0.604_{\pm.029}$ | 0.817 | 0.812 | 0.785 | 0.785 |
| Fairness Methods | DP | 0.842 | $0.696_{\pm.020}$ | 0.843 | $0.702_{\pm.014}$ | 0.712 | $0.671_{\pm.016}$ | 0.689 | $0.600_{\pm.031}$ | 0.815 | 0.813 | 0.785 | 0.795 |
| | EO | 0.813 | $0.727_{\pm.028}$ | 0.846 | $0.706_{\pm.015}$ | 0.713 | $0.670_{\pm.018}$ | 0.686 | $0.586_{\pm.031}$ | 0.814 | 0.798 | 0.782 | 0.789 |
| | EP | 0.801 | $0.728_{\pm.027}$ | 0.835 | $0.710_{\pm.018}$ | 0.712 | $0.671_{\pm.016}$ | 0.687 | $0.597_{\pm.031}$ | 0.814 | 0.815 | 0.786 | 0.786 |
| | Exp | 0.797 | $0.724_{\pm.034}$ | 0.838 | $0.702_{\pm.015}$ | 0.712 | $0.648_{\pm.018}$ | 0.685 | $0.597_{\pm.033}$ | 0.814 | 0.798 | 0.783 | 0.783 |
| | Threshold | 0.806 | $0.660_{\pm.003}$ | 0.812 | $0.536_{\pm.027}$ | 0.711 | $0.670_{\pm.018}$ | 0.688 | $0.605_{\pm.030}$ | 0.785 | 0.786 | 0.752 | 0.761 |

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

Table 15. For other settings, we do not report the worst-case results because there are not many target domains there and the mean/standard deviation results could reflect the generalization performance well.

For the results of Macro-f1 Score reported in Tables 11, 12, 13, 14 and 15, we find similar conclusion patterns to the accuracy score, while we usually witness a wider gap between *o.o.d.* and *i.i.d.* model performance.

### D.10 Agreement and Maintenance Plan

**Hosting Platform.** We will use Github as the hosting platform of our code. We provide detailed preprocessing Python scripts to guide users to replicate and process data from scratch. We also illustrate methodologies to run full experiment results in the code. We also list the license for each dataset and user guidance in the `Readme` file in that Github.

**Dependencies.** The benchmark is built upon Python 3.8+ and depends on PyTorch, aif360, fairlearn, xgboost, lightgbm. Besides, it uses numpy, scipy, and pandas for basic data manipulation.

**Maintenance Plan.** The datasets provided here will be maintained by the authors of the paper, which can be contacted by raising an issue on GitHub or by contacting the first authors directly. Our benchmark and a simple open-sourced Python package based on that may be updated at the discretion of authors in the future, which includes more refined algorithm implementations, datasets, and framework with improved efficiency.

**Author Statements.** To the best of our knowledge, the proposed benchmark is based on existing datasets and does not violate any existing licenses non contain personally identifiable or privacy-related information. And we claim all the responsibility in case of a violation of rights if such a violation were to exist.