# OpenReview forum: "On the Need for a Language Describing Distribution Shifts: Illustrations on Tabular Datasets"
_NeurIPS.cc/2023/Track/Datasets_and_Benchmarks — NeurIPS 2023 Datasets and Benchmarks Poster_

### Official Review · Reviewer_AmNg · 2023-06-27
**Good paper that makes a number of contributions to work on distribution shift, suggested revisions are mostly for clarity**

**Rating:** 8
**Confidence:** 3

**Strengths:**

•	effectively distinguishes Y|X-shift and X shift and motivates qualitatively and empirically why it matters.

•	Shift is an important issue for real world ML deployment and this paper packs in a number of contributions. It provides datasets, compares algorithms on those datasets, complicates some existing results, and introduces and demonstrates non-algorithmic methods for addressing shift .

•	Provides effective figures illustrating the main points and integrates them well into the text. I particularly like Figure 4 and the accompanying case study.

•	Although presented in a preliminary manner, the tool for identifying target covariate regions where shift may be an issue (section 3.1) could be a useful addition to the body of tools for identifying sources of shift and how to address them.

•	No ethical issues

•	Comes with an appendix that provides considerable further detail

**Additional Feedback:**

Some miscellaneous issues

* Pg. 4 the following sentence does not make sense grammatically: “87.2% of the pairs with over 50% of the performance gap are attributed to Y |X shifts (for 70.2% of the pairs, over 60% of the gap is attributed to Y|X shifts).” I understand what is meant, but it should say something like “For 87.2% of pairs, more than 50% of the drop in accuracy was attributed to Y|X shift”
* Pg. 5, the second equality in equations (3.4) and (3.5) is missing an argmin in front.
* Pg. 5, Notation-wise, it isn’t great to have S and script S for the shared distribution and the region of S in X – they look very similar, and this adds an unnecessary potential source of confusion.
* Pg 6. In figure 4c, it isn’t clear to me what shift(P) and shift(Q) mean.
* Pg. 6-8 Table 1 is missing the income (CA $\rightarrow$ SD) case from Figure 2. This isn’t a problem per se, but it is a bit confusing since on pg. 8, the table is described as “corresponding with Figure 2”
* Pg. 9: Figure 7a is not colorblind friendly – it might be nice to pick colors other than red and green

Note: I would be happy to change my 6 rating to a 7 or 8 if a few of my recommendations were taken into account – particularly writing 3.1 more clearly, addressing how to actually obtain the Y|X and X decomposition if I want to use the data, and giving a clearer roadmap at the beginning

**Clarity:**

Grammatically, I have no major issues. Organizationally, I find it a bit jumbled at times and have discussed this above as a major but not-too-hard-to-fix thing. Within sub-sections, this paper is mostly straightforward to read.

The one section that I found harder to read was section 3.,  which introduces a method for identifying regions with strong Y|X shift. I find it too sparsely explained and there is no appendix section to support it. It took me a while to pick out what was going on, convince myself that equation (3.3) made sense and think of some intuition for why the method makes sense. I had to define some additional notation and write out steps as below. I’ve copied this from my notes so that you can check whether I’ve misunderstood anything:

(note: I add in the notation $\Pi_Q$ to mark the event “observation drawn from population/group Q”)

1.	Define $s(x)$ theoretically as in equation (3.1) for reasons that will become clear

2.	By Bayes Rule with $P(\Pi_Q) = P(\Pi_P)= 1/2, P(\pi_Q|x) = q(x)/(p(x) + q(x)) $, which is by construction of $s(x)$, exactly the likelihood ratio $s(x)/q(x)$  (and similarly for $s(x)/p(x)$)

3.	We do not have access to these ratios, so we need to estimate them. To do this, we will use the fact that $\frac{q(x)}{(p(x)+q(x))}$ can be re-written as $\frac{P(\Pi_Q|x)}{P(\Pi_Q|x)P(\Pi_P) + P(\Pi_P|x)P(\Pi_Q)}.$ For our finite sample, $P(\Pi_Q)$ is just  $n_q/(n_q+n_p) = \alpha$. We can estimate $\pi(x) = P(\Pi_Q|x)$ via a classifier since in our training data, we have labels for Q vs P

4.	Next, we note the equality in equations (3.4) and (3.5) which tells us that to obtain a predictor for $P(Y|X)$ under $X\sim s(x)$, we can fit a model to our observed (X,Y) pairs but with losses weighted by likelihood ratios $s(x)/p(x)$ or $s(x)/q(x)$. This means we will prioritize minimizing loss where ratio is big, which, in case of $f_P$ corresponds to areas where $q(x) \gg p(x)$ and vice versa for $f_Q$. Intuition here is (I think?) that these are areas where we have less data with which to compare $P(Y|X)$ and $Q(Y|X)$ and also areas where there is already one kind of shift...so we want to really try to get this part right and not let our model disregard error here.

5.	Given $(X_i, |f_P(X_i)-f_Q(X_i)|)$ pairs, learn  $h(x)$ function for describing this relationship in general and use h(x) to obtain a region

Overall, I still feel that I don’t have an entirely satisfactory intuition for why this choice of $s(x)$ makes sense, but I can rationalize the consequences as in step 4 above. It seems to work in the case study at least. I think the section would be clearer if written out in a more narrative, step-by-step style, perhaps with the intuition above if it is correct, and perhaps with some supporting details in the appendix (e.g., showing a few more intermediary calculations, e.g., considering what happens in the case study example if we just estimate $P(Y|X)$ on each population without weighting?).

Also: perhaps related to the set-up in section 3.1 being not fully clear, I do not understand the description on pg. 17 (paragraph starting with “similar to our setup”) of how exactly other related methods differ from this one.

(I should note I do understand this method is meant to be a first step for further research, so it is ok if it isn’t fully explored and theoretically justified).

**Correctness:**

I have not identified any major issues with correctness of methods or claims.

A slightly more fuzzy objection is the way the paper occasionally makes claims of representativeness or comprehensiveness. The abstract says they curate 7 shifts with “representative patterns” – representative of what? It turns out these are a subset of 169 source-target pairs, and it is not clear what the selection principle for the 7 are. Are they examples picked to display different possible things that can happen, possibly including extreme cases? Are they common examples? I would avoid claiming representativeness and just describe them as examples picked for some purpose – e.g., for variety in how much performance decline attributable to Y|X.

For 169 source pairs, it also isn’t entirely clear what they represent. It seems implausible that they would be ‘representative’ of all kinds of real-world tabular datasets. The authors recognize this on pg. 17 of the appendix but at times they sound like their benchmark is meant to comprehensively cover tabular data in general (“a comprehensive benchmark”). Again, I would instead emphasize variety. The key point really seems to be that the benchmark includes a variety of examples of both Y|X and X patterns and that there is a way to identify which is which.

Relatedly, some claims seem a bit over-stated. On pg. 2 and 3, the claims that “Y|X shifts are prevalent in tabular settings” and “natural distribution shifts are largely induced by Y|X-shifts” are really only demonstrated on a few datasets in a particular domain. Like the reasoning about missing features on pg. 4 (which I like), the more general claims about Y|X prevalence are conjectures (plausible ones, I think).

I would also be interested to see some characterization of all 169 pairs. Of course they cannot all be displayed in figures, but perhaps some table with summary statistics could be put in the appendix or there could be a plot showing the distribution of Y|X performance decline and the correlation between “performance decline attributable to Y|X” and “R^2 between source and target accuracy”. I would be more convinced on the point that accuracy-on-the-line holds only for X-shift if it were demonstrated on all 169 and/or on a large set of synthetic examples like that in Figure 2g.

**Documentation:**

The appendix gives extensive documentation of the authors’ analysis process. There is little discussion of the code and logistics of the benchmark in the main paper, but the appendix gives an overview and links to the GitHub (it would be good if the link to the GitHub were also somewhere in the main body of the text). The GitHub seems well laid-out and accessible.

My one big question is: if I want to use these datasets, is there anywhere where I can get access to the calculations of performance decrease attributable to Y|X as displayed in Figure 2? Or at least to accessible code used to calculate this? The benchmark seems less useful if I cannot easily access that information and thereby evaluate my algorithms on different source-target tasks while dividing them up by more Y|X shift vs more X shift as the paper does. I am not familiar with DISDE so having to go figure that out and implement it myself would be a barrier to use and reproducibility.

**Ethics:**

I have no ethical concerns for this paper. It provides tools for understanding an issue of ethical importance (shift) and the datasets are open source

**Limitations:**

I see no potential negative societal impact of this work. The work means to develop tools and benchmarks for dealing with a phenomenon which can lead to negative societal impact.

The authors have a section on limitations, but it only occurs in the appendix. If there is room, it would be better to integrate it into the conclusion.

**Opportunities For Improvement:**

Organizationally, I find the paper jumbled at times, with various topics addressed in rapid fashion and returned to multiple times. When I first read it, I did not have a good sense from the introduction of where we were headed. I expected to quickly hear about the benchmark datasets (but this was not until section 4), thought the relative regret metric and demographic benchmark datasets would play a bigger role than they did, and wasn’t sure what the key contributions would be. This paper does a number of things, so it is important to begin with a clear roadmap.

Specifically, I would suggest adding to the introduction:

1.	More discussion of relation to prior work and what existing benchmarks lack (see “relation to prior work” section for a specific suggestion)

2.	The “Choice of Domain” discussion on pg. 17 (the appendix) to motivate the focus on time/space shifts. I don’t fully understand how the authors view the demographic-shift datasets in relation to the time and space datasets. Do they complement each other? Should methods intended for demographic shift also be evaluated on the time and space shift datasets or vice versa? Or is the point more that time/space shifts are also important, and we shouldn’t assume that what works for demographic shift datasets works there?

I would replace the three bold headings in the introduction with a more integrated roadmap and motivation discussion. The bold heading format works well elsewhere in the paper but in the intro, I think it is more important to describe the questions the paper will answer and in what order than to give initial observations the do not yet have much context.

My mental roadmap of the paper looks something like

1.	Discuss different types of shifts and how can measure whether performance decline comes from Y|X vs X; emphasize importance of Y|X shift in non-image/text data context

2.	To emphasize importance of this distinction and study Y|X shifts...

a.	Demonstrate empirically that accuracy on the line finding doesn’t hold across these types of shifts

b.	Demonstrate empirically that algorithm performance varies with shift pattern

c.	Explore non-algorithmic interventions for addressing Y|X shifts via a case study and propose a method for figuring out covariate regions where Y|X shift is particularly a problem

3.	Provide benchmark with naturally occurring shifts so that other researchers can also evaluate methods with attention to type of shift

This roadmap only became clear to me after reading it more than once. I think it should be clear by page 2.

**Relation To Prior Work:**

Discussion of prior work in the main text is not extensive, but there are numerous references to other work. For example, the paper builds directly on the DISDE framework for decomposing performance degradation into Y|X and X shifts. This method isn’t explained in the paper, but readers are pointed to the paper introducing DISDE.

The appendix provides a bit more detail on a few points. Although there is not room for an extensive review in the main paper, I think at the very least this sentence from the appendix  should be moved to the introduction:

“To evaluate robustness of different algorithms, most of existing distribution shift benchmarks focus on the image and language datasets [52, 34, 73, 57, 67]. In contrast, few benchmark papers focus on real-world tabular datasets, which usually exhibit different patterns compared to those image datasets in terms of the prevalent methods”

This would better motivate the nice discussion in the main paper of why we might expect different kinds of shifts for tabular vs image/text data shifts.
I’m also wondering if there is any connection between this and the failure to replicate the accuracy on the line phenomenon from [34]. It would be nice if this paper engaged more explicitly with the accuracy-on-the-line paper, why its result matters, and why this paper finds a different result. Based on the current paper, I might expect [34]’s result to reflect it using more image/text data examples or having data with primarily X-shifts. Any indication that’s the case?

**Summary And Contributions:**

The paper addresses the need for benchmark data to evaluate algorithm performance under distribution shift in the context of tabular data. The authors emphasize the distinction between Y|X shifts and X shifts and use a recent method to decompose performance decrease due to shift into these two components. Using various source-target pairs from 5 economic or traffic related datasets, they compare the performance of 86,000+ model configurations across different types of shift. They find that algorithm rankings change with shift patterns and that the relationship between source and target performance is not per se strongly linear here, in contrast with previous research that has indicated this pattern. Additionally, the paper looks at non-algorithmic methods of addressing shift, including collecting data from the target population and adding more relevant features. To aid efficient data collection from the target population, they propose a method of identifying regions of the X space where there is particular shift and implement it on an example. These later contributions are fairly preliminary and meant to spur further research.

---

> ### Author Response · Authors · 2023-08-17
> **Rebuttal**
>
> We deeply appreciate your recognition of the motivation, our proposed benchmark, and the empirical findings presented in our work. Your detailed review is invaluable. We are grateful for the time and effort you have invested. We have attentively considered your suggestions and have addressed each concern as outlined below:
>
> ### **Improvement: More clear contexts to add**
> We follow your suggestion to include more discussion on what existing benchmarks lack.
>
> **Choice of Domains**: We mainly focus on time/space shifts since ML models often need to be deployed in other spatiotemporal domains while trained with abundant data from the source domain in practice. And we find that these shifts are also important instead of solely demographic shifts and exhibit more diverse behavior in terms of the shift patterns (shown in Figure 1). We add some discussion in the main body.
>
> ### **Improvement: Organization of Introduction**
>
> We follow your suggestions and restructure our introduction with a clearer roadmap. Here's a summary of the modification:
>
> * **Paragraph 1**: We initiate with an overview of distribution shifts in the image and tabular data, giving readers a broader perspective on real-world shifts.
> * **Paragraph 2**: This section delves into the varied solutions for different types of shifts ($X$-shifts and $Y|X$-shifts), underscoring the importance of deciphering shift patterns.
> * **Paragraph 3**: We add a concise discussion highlighting the shortcomings of existing benchmarks, thereby setting the stage for a deeper exploration of shift patterns in tabular data.
> * **Paragraph 4**: We briefly mention why we only consider transferring from the smallest to the largest group.
> * **Paragraph 5**: Here, we elucidate our domain choices, explaining our focus on spatiotemporal shifts.
> * **Outline**: We replace the initial three headings with a more descriptive outline that encapsulates the core ideas of Sections 2, 3, and 4, providing a comprehensive guide to the paper's flow.
>
> ### **Limitations of This Work**
>
> We appreciate your insightful feedback.
> In response to your concerns, we add a comprehensive discussion of the limitations of this study in Appendix B. Specifically, the outlined limitations encompass:
>
> * Limitations pertaining to the benchmark.
> * Limitations associated with Algorithm 1.
> * Limitations related to data collection.
> * Discussions on the applicability across various data modalities.
>
> And we will add these to the conclusion in the camera-ready version.
>
> ### **Correctness: Source-Target Pairs and Their Representativeness**
> **Selection Principle of 7 Settings**: For every setting, the source domain is chosen based on the magnitude of performance degradation. Specifically, we prioritize cases where there are substantial performance drops between the source and target domains, indicating pronounced distribution shifts. Target domains within each setting are selected if the performance degradation exceeds 5 percentage points, as detailed in Table 3. We introduce the 'Dom. Ratio' to denote the predominant ratio of $Y|X$-shifts or $X$-shifts in source-target pairs with results in Table 3 of Appendix D. This metric is relevant for pairs exhibiting a performance drop greater than 5 percentage points across the 22 settings. This helps show the prevalence of $Y|X$-shifts in our settings.
>
> **Overclaim**: We replace the claim on Y|X prevalence with a milder one and remove the "comprehensive benchmark" phrase in Section 2.
>
> **Characterization of all 169 pairs**: In the main body, where we discuss 7 selected settings, we pinpoint one source-target pair that shows relatively large performance degradation from each dataset and type of natural shifts. In total, there are 169 pairs in the 7 settings. A summary of the results of the $Y|X$-shifts ratio is deferred to Appendix C.2, demonstrating the prevalence of $Y|X$-shifts in our benchmark.

---

> > ### Author Response · Authors · 2023-08-17
> > **Clarity: Algorithm 1 (Section 3.1)**
> >
> > **Clarifications and Intuitions**: We elaborate clearer insights into the problem setting in Section 3.1 including the inputs and outputs of Algorithm 1. We have also delved into the rationale behind our choice of the shared distribution, $S_X$. While space constraints prevent us from including all derivations in the main text, we plan to incorporate them in the camera-ready version. We add Appendix C.3 to offer a comprehensive understanding. This appendix elucidates:
> > * The considerations behind selecting $S_X$;
> > * A step-by-step breakdown of Algorithm 1.
> >
> > **Question about estimating $P(Y|X)$ on individual populations without reweighting**: Since the marginal distribution $P_X$ could differ from $Q_X$, the empirical estimates, $\hat{P}(Y|X)$ and $\hat{Q}(Y|X)$, would likely be influenced by covariate shifts (referred to as $X$-shifts). Our strategy involves aligning the marginal distributions $P_X$ and $Q_X$ to $S_X$. This alignment effectively mitigates the effects of $X$-shifts. Subsequently, we determine the $Y|X$ discrepancy by assessing the difference in predictions, specifically $|f_P(X)-f_Q(X)|$.
> >
> > ### **Relation to Prior Work**
> >
> > **Introduction to DISDE**: We have incorporated a detailed introduction to the DIstribution Shift DEcomposition (DISDE) method in Appendix C.1, covering the notations needed in our paper.
> >
> > **Accuracy-on-the-line**: We provide a detailed discussion here:
> >
> > 1. **Importance of Accuracy-on-the-Line**
> >
> >     Researchers [1] identified the accuracy-on-the-line phenomenon across multiple image datasets, highlighting a robust correlation between in and out-of-distribution accuracy. If this phenomenon remains consistent, improving in-distribution performance could potentially be key to addressing out-of-distribution generalization challenges.
> >
> > 2. **Accuracy-on-the-Line and Its Relation to $X$-shifts**
> >
> >     We believe the accuracy-on-the-line phenomenon is closely related to $X$-shifts. This observation is primarily observed in image datasets, where $Y|X$-shifts are relatively subdued due to the construction of $Y$ from human-assigned labels for a given input $X$. This remains true, except for the cases of significant labeling noise. When only $X$-shifts exist between distributions $P$ and $Q$, the primary objective becomes learning $E_P [Y|X]=E_Q [Y|X]$, given that $P(Y|X)=Q(Y|X)$. Hence, an enhanced fitting to $E_P [Y|X]$ — indicative of superior in-distribution performance — would likely result in a better fitting of $E_Q[Y|X]$, which is reflected in out-of-distribution performance. Consequently, a potent positive correlation emerges between in- and out-of-distribution performances, exhibiting the accuracy-on-the-line pattern.
> >
> > 3. **Breakdown of Accuracy-on-the-Line under $Y|X$-shifts**
> >
> >     With pronounced $Y|X$-shifts, an improved fitting to $E_P [Y|X]$ does not necessarily yield better results for $E_Q [Y|X]$ because $P(Y|X) \neq Q(Y|X)$. Moreover, without test data, predicting the relationship between these becomes challenging, causing the in- and out-of-distribution accuracy correlation to appear arbitrary. This underscores the complexity and the necessity for nuanced solutions to address $Y|X$-shifts.
> >
> >     Tabular data, compared to image or language datasets, is more susceptible to intense $Y|X$-shifts due to the presence of omitted variables and latent confounders. An illustration of this is how the incidence rate of certain diseases among a population can be influenced by unrecorded covariates. These could include factors such as diet, physical activity, smoking habits, or socioeconomic conditions. Conversely, image or language data often possess comprehensive information for predicting $Y$, leading to a consistent prediction framework across training and testing phases, barring significant labeling noise.
> >
> > [1] Miller, J.P. et al. Accuracy on the Line: on the Strong Correlation Between Out-of-Distribution and In-Distribution Generalization. ICML 2021.
> >
> > ### **Documentation: Decomposition of $Y|X$ vs. $X$ Shifts in Practice**
> > **Reproducibility for Practical Application**: For hands-on implementation, we have integrated the DISDE tool within our Python package, `whyshift`. This package is publicly accessible via our GitHub repository at [namkoong-lab/whyshift](https://github.com/namkoong-lab/whyshift). A core function, `degradation_decomp`, aids users in attributing performance degradation to $Y|X$-shifts and $X$-shifts. Installation is straightforward with `pip install whyshift`. To assist users, we offer a comprehensive `README` file as well as an illustrative Jupyter Notebook tutorial. Additionally, the official codebase for DISDE is available at [namkoong-lab/disde](https://github.com/namkoong-lab/disde).

---

> > > ### Author Response · Authors · 2023-08-17
> > > **Additional Feedback**
> > >
> > > Thanks for the careful reading in terms of the grammar soundness of sentences on pg 4, missing notations on pg 5, and color suggestions. We have updated them. Other issues are addressed as follows:
> > >
> > > - **Notation of S and Script S**: We change the notation of specific regions in $\mathcal S$ to $\mathcal R$ to avoid any confusion.
> > > - **Update of Figure 4c**: Shift(P) and Shift(Q) denote the different components of $X$-shifts (Term I and III in Appendix C.1). We combine the legends of them in Figure 4c to $X$-shifts to avoid more notations.
> > > - **More subfigures in Figure 2 than benchmarks in Table 2**: Sorry for the confusion. We have removed the income (CA2SD) case in Figure 2 to be consistent with our benchmarks and to avoid any confusion.

---

> > ### Comment · Reviewer_AmNg · 2023-08-29
> >
> > **(Responding to response 1)**
> >
> > Thank you for taking into account my suggestions. The introduction is much improved!
> >
> > I appreciate the addition of a limitations section in the appendix - is some addition to the conclusion (Section 5) still pending? You mention it but I do not currently see it.
> >
> > I think I had misunderstood before exactly the taxonomy of datasets, settings, and source-target pairs. Appendix C.2. is a good clarifying addition here. It might still be good to explain this slightly more around lines 63-65 in the main paper,  clarifying that "setting" refers to a set of outcomes+features (22 in benchmark, 7 selected for illustration), each setting has a source (e.g., California), and some number of possible targets (e.g., another state) with one target per setting selected in the paper to illustrate a range of Y|X shifts
> >
> > (I will add an additional comment on the section 3.1 revisions in a day or two once I've had a chance to review them)

---

> > > ### Author Response · Authors · 2023-08-29
> > >
> > > Thank you for your response and for acknowledging the improvements in our revised manuscript.
> > >
> > > We really appreciate your further suggestions on the clarification parts of the paper. We have updated a revised version including a short summary of limitations on the conclusion part. We have also explained a bit more about the selection of different settings following your suggestions around Line 63 following your suggestions.
> > >
> > > We eagerly await your further comments on Section 3.1 and look forward to hearing from you soon.

---

> > > > ### Comment · Reviewer_AmNg · 2023-08-30
> > > >
> > > > **Re: lines 63-69 in the revised paper** – thank you for looking to clarify this. However, I think it is still a bit confusing. You reference Table 2, which is in the supplementary section, and has 7 states across the top but it isn’t obvious to a new reader that these are the settings or what the settings are.
> > > >
> > > > Rearranging/tweaking your sentences a bit, I might suggest something like the following after the sentence, “To model... shifts.”:
> > > >
> > > > “The full benchmark covers 22 settings (see Supplement Table 3), where each setting consists of some number of features and some outcome. Each setting includes one source (e.g., data on California) and a number of possible targets (e.g., data for the other states). For illustration in this paper, we focus on 7 settings, covering 169 possible source-target pairs. We also carefully select 7 specific source-target pairs (one target per setting) to represent a wide range of Y|X shifts (right bars in Figure 1)."
> > > >
> > > > That is, I would describe the benchmark first and then specify that for the paper, you will focus on the 7. Sorry to be nit-picking about this but I want to make sure readers aren’t confused as I was by the different numbers (169, 7, 22) coming by and how they relate to each other, and I want to make sure they appreciate the size of the benchmark beyond the 7 examples.
> > > >
> > > >
> > > > **Section 3.1 comments**
> > > >
> > > > Thank you for adding some more detail on DISDE and on the choice of s(x) in section C.3. I think that it will now be easier for the interested reader to follow what is going on and have some understanding of where S comes from and what source 16 describes further.
> > > >
> > > > notes:
> > > > * In line 175, you refer to “the shared distribution” but haven’t introduced the idea of a shared distribution yet – one extra sentence introducing the idea of a shared distribution would help the logical progression. (e.g., “to aid comparison on the common support, [16] introduce a shared distribution which ____”)
> > > > * Argmin notation: you use a notation I am unfamiliar with in (3.4) and (3.5) when you write statements of the form argmin(f(x)=g(x)). I’m not sure how standard this is but I found it confusing, though I also see that argmin(f(x))=argmin(g(x)) is also not quite what you want since the point here is f(x)=g(x). Not sure what the best solution is but just noting that I now see why I thought there was a typo earlier.
> > > > * I think pg. 23 line (I) has a typo - the left side as written now would just be 0 and I’m thinking the second $E_Q$ should be an $E_P$. Also, isn't it term (I) and (III) that are related to X-shifts and term (II) that reflects Y|X shift (flipped from what it says now)?
> > > >
> > > >
> > > >
> > > > **Section 5 - Limitations:** Thank you for adding the summary in the main paper. The sentence in line 357-358 is a bit confusing – I think you mean to say that the technique of identifying a risky region and collecting further data may sometimes be infeasible or unethical due to privacy concerns, but the current sentence literally says something else. Overall, since you do have a bit more room on the page, if you wanted to discuss a bit more and/or weave the conclusion and limitation together, that might be nice, but aside from that one sentence, the section is fine as is.

---

> > > > > ### Author Response · Authors · 2023-08-30
> > > > >
> > > > > Thank you for your additional feedback aimed at enhancing the clarity of our paper and helping readers digest our paper more easily.
> > > > >
> > > > > We have incorporated all of your recommendations into the revised manuscript.
> > > > > - Regarding lines 63-69: Following your guidance, we have restructured the sentences.
> > > > > - Typos: We appreciate your keen eye for detail and have rectified the errors. We have removed the typos on Page 23. Concerning the $\arg\min$ notation in (3.4) and (3.5), we have revised it to $\arg\min \bigg\\{ f(x) (=g(x))\bigg\\} $. This adjustment clarifies that $f(x) = g(x)$.
> > > > > - Conclusion: We have rephrased the sentence in line with your suggestion.
> > > > >
> > > > > We are confident that these revisions have improved the quality of the paper and hope that you find these revisions aligned with your feedback and that the updated manuscript is a more solid contribution to the field. We look forward to your reassessment of the paper.

---

> > > > > > ### Comment · Reviewer_AmNg · 2023-08-30
> > > > > >
> > > > > > Thank you - I will increase my rating.

---

> ### Author Response · Authors · 2023-08-24
> **Further Reply to Reviewer AmNg**
>
> We sincerely appreciate your approval and suggestions to improve this paper and we would really like to know if you have any further concerns or ambiguity regarding this paper. We hope to make the most of this opportunity to shed light on this paper. Therefore, we want to **make all our efforts to make sure that all your concerns can be well-addressed**.
>
>
> According to your suggestions, we summarize the main points of our revisions as follows; we highlight key revised contexts in blue.
>
> * **Introduction**: We restructured our introduction to provide a clear roadmap. By explicitly highlighting the shortcomings of existing benchmarks, we provide a sharper motivation for the work.
>
> * **Algorithm 1**: We have rewritten Section 3.1 to provide a clearer overview. In addition to a better description of Algorithm 1, we provide more details for our choice of the shared distribution, $S_X$, including in the appendix.
>
> * **Code release**: We have updated our GitHub repo at [https://github.com/namkoong-lab/whyshift](https://github.com/namkoong-lab/whyshift), to  **encompass all necessary codes to utilize our benchmark and reproduce our results**, including the implementation of **Distribution Shift Decomposition (DISDE)** method. We have provided a detailed `README` file and a Jupyter notebook as a tutorial for our `WHYSHIFT` package.
>
> **We welcome any further advice to improve this work and we will do our best to address your concerns.**

---

### Official Review · Reviewer_Tsg7 · 2023-07-20
**There is a Distribution Shift in Tabular Data Across 5 Datasets...Now What?**

**Rating:** 5
**Confidence:** 4
**Clarity:** Yes

**Strengths:**

The analysis appears interesting and like there was a lot of effort into the work.

**Additional Feedback:**

NA

**Correctness:**

I was unable to verify much of their claims, but it is not hard to believe that tabular data across the US population would experience distribution shifts. The authors do not really suggest how the effects of this could be minimized to assist other researchers with performing better analysis nor do they seem to temper the effects of distribution shifts with any type of weighting mechanisms, time series analysis, dimensionality reduction or the like that would probably change the shape of their analysis if they had.

**Documentation:**

No

**Limitations:**

The authors themselves point out that gathering supplementary data to the data used in the paper may improve some of the effects of the distribution shifts. Then, they point out that adding supplementary data would be costly and proceed to perform a benchmark that tests distribution shifts across 22 different algorithms- a quite costly endeavor. I had a hard time being convinced that such work would greatly contribute to advancement in the field.

**Opportunities For Improvement:**

Unfortunately, when going through their github repo, I was unable to see the plots, algorithms tested or analysis discussed in the paper. It would be beneficial for the authors to include this work for the purposes of verification.

**Relation To Prior Work:**

Not really- this seemed to not really position itself amongst the work it cited as either a contributing argument nor a opposing one. They simply cited work alongside theres without showing the reader how that work may have informed the formulation of their benchmark.

**Summary And Contributions:**

This paper evaluates 5 tabular datasets to examine distribution shifts across geographical location, year, sex and race depending on the dataset. When testing 22 different algorithms, they are able to prove a difference of performance of these models across spatial shifts.

---

> ### Author Response · Authors · 2023-08-17
> **Rebuttal**
>
> We sincerely appreciate your approval of the analysis and the efforts of our work.
> Thank you for the advice to improve this work, and we address all your concerns as follows:
>
> ### **Improvement: GitHub Repository**
>
> We appreciate your feedback and have updated our GitHub repository, which can be found at https://github.com/namkoong-lab/whyshift. (Please note that the link is also updated in the revised version of our paper.)
>
> We have developed a Python package titled `WHYSHIFT`, which **encompasses all necessary codes to utilize our benchmark and reproduce our results**. Key functionalities include:
>
> * **Fetching Benchmark Data**: Simply use the `get_data` function to obtain benchmark data across different settings.
> * **Testing Various Algorithms**: Invoke the `fetch_model` function. Our package boasts implementations of all the algorithms featured in our paper, such as `Logistic Regression`, `MLP`, `SVM`, `Random Forest`, `XGBoost`, `LightGBM`, `GBM`, $\chi^2$/CVaR-`DRO/DORO`, `Group DRO`, `Simple-Reweighting`, `JTT`, `Fairness-In/Postprocess`, and the `DWR` methods.
> * **Identifying Risk Regions**: This can be achieved using the `risk_region` function.
> * **Performance Degradation Decomposition**: Users can call the `degradation_decomp` function for this.
>
> In addition to the above, we have prepared a comprehensive `README` file and a Jupyter notebook as a tutorial for our `WHYSHIFT` package. It is noteworthy that Figures 2, 3, 5, 6, and 7 in our paper draw from the test performances of various algorithms in different configurations. All of these can be conveniently replicated using our package to test methods in diverse settings.
>
>
>
> ### **Limitation: Benefits of Adding Data**
> On the point of **adding supplementary data would be costly**, we demonstrate the effectiveness of adding supplementary data.
> **Our data-collecting method could improve efficiency and reduce costs.**
> The primary objective of this study is to delve into the intricate patterns of distribution shifts in real-world tabular data. Furthermore, we aim to inspire researchers to cultivate a more nuanced language for addressing distribution shifts. In alignment with this, we have established a benchmark characterized by diverse settings that underscore specific shift patterns.
> Our investigations reveal that $Y|X$-shifts are pervasive, underscoring the need for advanced tools to comprehend the nuances of these shifts more deeply. This insight led us to develop Algorithm 1, which identifies covariate risk regions with significant $Y|X$ discrepancies.
>
> Building upon our risk region algorithm, we introduce a simple method to gather target data more efficiently. Recognizing the practical challenges and costs associated with data collection, our approach optimizes this process. Instead of collecting data from the entire target distribution, we focus exclusively on the identified risk region, thereby maximizing efficiency and reducing costs.
> Moreover, the efficacy of our data-centric intervention methods accentuates the promising prospects in this domain of research.

---

> > ### Author Response · Authors · 2023-08-17
> > **Correctness: Suggestions on how to Minimize Distribution Shift Effects**
> >
> > 1. **Our Main Contribution**: Our primary emphasis is the meticulous examination of distribution shift patterns in the context of real-world tabular data. We introduced an algorithm to pinpoint risk regions and provided a benchmark characterized by distinct shift patterns. As a paper on datasets and benchmarks track, designing algorithms to address the distribution shift problem is not the main focus.
> >
> > 2. **Our Proposed Mitigation Approaches**: We recommend strategies to alleviate the impacts caused by distribution shifts. Given our discovery that $Y|X$-shifts are prevalent in real-world tabular data, Algorithm 1 was designed to identify risk regions with pronounced $Y|X$ discrepancies. Subsequent to this, we advocate two straightforward, non-algorithmic interventions: selectively gathering specific data and incorporating pertinent features. These interventions markedly outperformed several conventional baseline methods, highlighting the potential of non-algorithmic approaches. And we hope that this could inspire further research on this line.
> >
> > 3. **Methods Eliminating the Effects of Distribution Shifts**: We include almost existing feasible approaches tailored to distribution shifts and our existing benchmark encompasses a diverse array of existing methods, spanning DRO methods, tree ensembles, imbalance strategies, fairness approaches, and base methods such as LR, SVM, and MLP. In terms of the weighting mechanism, DRO and imbalance methods introduce unique weighting mechanisms, which are tailored for distribution shifts. We note that time series analysis might not be relevant in our context as our dataset doesn't include time series data and the dimension reduction approach is better suitable for image or text data with spurious features to mitigate but not quite relevant in the tabular data since the major problem in the tabular data is the missing feature.
> >
> >
> > ### **Relation to Prior Work**
> >
> > We appreciate your feedback. In response, we have reorganized the relevant literature part in Section 1, mainly highlighting the difference between our distribution shift benchmarks and existing benchmarks. We have also expanded upon the content of the related work in Appendix A, which we will move to the main body in the camera-ready version. In it, we provide a comprehensive overview of various related studies and how our work relates to each stream of them, including:
> >
> > * Existing Benchmarks for Distribution Shifts.
> > * Decomposition of Performance Degradation.
> > * Relationships between Non-algorithmic Interventions and Specific Algorithmic Interventions, including:
> >     * Active Learning.
> >     * Imitation Learning.
> >     * Causal Learning.
> >     * Feature Sensitivity / Importance Analysis.
> >     * Region Analysis: Existing Techniques for Identifying Shifts in Features and Regions.

---

### Official Review · Reviewer_wGZu · 2023-07-21
**The paper makes contributions by recognizing the importance of tailored interventions for different distribution shifts, grounding methodological research, and acknowledging the implicit dependency of empirical findings.**

**Rating:** 6
**Confidence:** 2
**Clarity:** This paper is not very well written.

**Strengths:**

1. The authors propose a test platform to aid researchers in developing more concise language for distribution shift analysis.

2. The paper discusses various types of distribution changes and their impacts on algorithms and data-driven interventions. This research is highly relevant to the broader field as it delves into the challenges of handling distribution shifts in practical applications.

3. This paper emphasizes the importance of identifying the region of covariates in a table setup where Y|X shifts are most pronounced and proposes a simple yet efficient method to identify data regions with strong Y |X shifts


**Additional Feedback:**

No

**Correctness:**

It is a benchmark, the evaluation methods and experiment design appropriate and performed correctly.

**Documentation:**

Based on what the authors have written in Section 4.1, there is enough detail to support reproducibility.

**Ethics:**

I believe that the submitted paper does not present any ethical concerns that require further discussion or review.

**Limitations:**

No. The authors not adequately addressed the limitations and potential negative societal impact of their work. Based on this statement, one limitation of this work is that it focuses on exploring distribution shifts in real-world tabular datasets but does not explicitly address potential limitations or challenges associated with generalizing the findings to other types of data or domains. Further investigation and discussion on the transferability and applicability of the proposed algorithmic and data-based interventions across different data modalities would strengthen the overall scope and impact of the research.

**Opportunities For Improvement:**

1.	The writing style of this paper is subpar, and the organization lacks clarity. There is room for improvement in terms of coherence and structure.

2.	This paper is lacking explicit discussion on the limitations of the presented work. It would be beneficial for the authors to address and acknowledge any constraints, constraints, or potential shortcomings of their approach or findings.


**Relation To Prior Work:**

Yes, the paper clearly discusses how this work differs from previous contributions.

**Summary And Contributions:**

The paper makes contributions in the following areas: recognizing the need for tailored interventions for different distribution shifts, grounding methodological research in specific shifts, acknowledging the implicit dependency of empirical findings on the type of shift, developing an empirical testbed for performance benchmarking, identifying affected covariate regions through algorithmic design, and calling for future research on distribution differences.

---

> ### Author Response · Authors · 2023-08-17
> **Rebuttal**
>
> Thank you for your valuable feedback.
>
> ### **Improvement: Coherence and Structure**
>
> We really appreciate the detailed suggestions. We have carefully followed your comments to improve our manuscript, including restructuring the introduction to provide a clearer roadmap.
>
> ### **Improvement: Discussion on Limitations**
>
> We are grateful for the feedback, and in response, we have incorporated **a comprehensive discussion on the limitations** within Appendix B, which we plan to move to the main body in the camera-ready version. Beyond the aforementioned "Transferability and Applicability" section, our limitations are articulated as follows:
>
> * **On the Benchmark**: Our current scope is confined to source-target transfer pairs predominantly from the economic and transportation sectors. We recognize the potential and importance of exploring patterns in other domains, such as medical datasets (e.g., MIMIC-III), and earmark this as a compelling avenue for future research. Additionally, our benchmark only contemplates sources and targets within a singular domain, typically specific to a state or city. A more encompassing approach would involve considering both source and target distributions across multiple domains and varied proportions.
> And our results in characterizing the distribution shift patterns highlight the importance of utilizing other refined tools. These tools can help us understand the difference between real-world distribution shifts and enable further investigation and analysis.
>
>
> * **On Algorithm 1**: This algorithm is tailored to discern risky regions with pronounced $Y|X$-shifts, especially when there is a stark model performance divergence between training and testing datasets.
> One limitation arises in its dependence on target data to determine these risky regions. Therefore, Algorithm 1 may fail when the target distribution is unobservable.
> Furthermore, when applying non-algorithmic interventions raised from Algorithm 1, it is important for researchers to carefully **combine background knowledge** with results obtained from the algorithm. As elucidated in Section 3.2, our scrutiny of risky regions spotlighted the potential significance of the "ENG" feature in mitigating $Y|X$-shifts. This is a revelation anchored in our prior understanding of language differences between the states in focus. Moreover, any cavalier application of Algorithm 1, such as deploying the identified risky regions recklessly, could inadvertently harm already-vulnerable demographics.
>
>
> * **On Data Collection**: In Section 3.2, we provide two simple non-algorithmic interventions to offset the $Y|X$-shifts, one of which is to efficiently collect target data from the risky region. This method, when synergized with several benchmarked techniques, exhibits superior outcomes. Nevertheless, we acknowledge its restricted applicability only when the target distribution permits sampling and where data collection does not raise any privacy concerns or result in predatory inclusion.
>
> ### **Limitation: Transferability and Applicability**
>
> **Our algorithm and interventions are designed with potential generalization to various data types in mind.**
>
> In the current study, we focus on addressing distribution shifts within the context of tabular data.
> While we have not delved deeply into the versatility of that method across diverse data types, we posit that Algorithm 1 has the potential to be adapted for more complex data types, such as image data, especially when paired with appropriate deep learning models.
> To this end, both the domain classifier (to estimate $\mathbb P(X\text{ from }Q_X|X)$) and the region learner, $h(X)$, would need to transition into deep neural network structures. One problem may be to explain the output important feature and regions which may not be as clear as in the tabular data. We leave further investigation to future work.

---

> > ### Comment · Reviewer_wGZu · 2023-08-30
> >
> > Thanks a lot for your response. Most of my concerns have bee addressed.

---

### Official Review · Reviewer_np4K · 2023-07-21
**Thorough and well-documented analysis of distribution shifts in tabular data which could benefit from further discussing societal implications**

**Rating:** 7
**Confidence:** 3

**Strengths:**

- The authors motivate studying distribution shifts for tabular data, citing "missing variables and hidden confounders" (line 25) (e.g., smoking status, socioeconomic status). The authors also claim that it is unclear which types of distribution shifts existing benchmarks evaluate, which can cause them to have poor external validity across diverse real-world shifts.

- The authors rightly emphasize the need to characterize and understand the source of distribution shifts, towards devising interventions beyond technical solutions; their case study with ACS Income, which includes concrete social groundings of empirical observations, is a nice example of this.

- The authors benchmark an impressive number of distribution shift methods and model configurations (including fairness methods and tree ensembles).

- The authors' findings that: (1) Y | X shifts are most responsible for performance degradation on the tabular datasets (with the exception of temporal shifts), and (2) a small validation set from the target distribution can help with transfer, are interesting.

- Figure 2 clearly shows that across datasets and hyperparameters, even the best XGBoost model experiences considerable degradation in its target accuracy vs. its source accuracy, and there is great variance in the magnitude of degradation.

**Additional Feedback:**

- Change Figure 2 to have x and y-axes with the same range and scale.

**Clarity:**

- The paper is extremely well-written and clear.

**Correctness:**

- The formulation of relative regret is reasonable. However, why do the authors only consider the largest and smallest demographic subgroups in Figure 1? Are the authors assuming that transferring from the smallest group to the largest is the most difficult shift for a model to handle? This may not always be true.

- Line 52: While Figure 1 suggests that Y | X is transferrable, it is unclear how Figure 1 demonstrates that "X-shift is relatively mild."

- Algorithm 1 appears technically correct.

- The baselines in Section 4 are reasonable and relatively comprehensive. The natural shifts that the authors consider are realistic. The authors provide error bars for all experimental results in Figure 6.

**Documentation:**

- The authors provide extensive documentation of their experiments and results.

- The authors provide an explicit plan to maintain their data and code.

- To the best of the authors' knowledge, the authors comply with the terms of use of the datasets in their experiments.

**Ethics:**

- The authors use the COMPAS dataset, which has validity concerns (e.g., "measurement biases and error") [2] and enables recidivism prediction technology, which is unethical.

[2] https://arxiv.org/abs/2106.05498

**Limitations:**

- The authors use the Adult dataset, but some papers argue for retiring this dataset and instead using ACS Income (which the authors also use) [1]. The ACS Income dataset is larger and has more recent data than Adult.

- The authors discuss the limitations of their study in the Appendix (not in Section 5, as indicated by their checklist). The authors should also discuss potential negative implications of their work. (e.g., How may Algorithm 1 be misused, and what would be the implications of this? How can incorrect predictions by Algorithm 1 harm already-vulnerable groups?)

[1] https://arxiv.org/pdf/2108.04884.pdf

**Opportunities For Improvement:**

- The authors should comment on the practical and ethical implications of collecting data from the target distribution. Algorithm 1 can only be used if the target distribution is known and sampleable; should a model ever be deployed if the target distribution is unknown? Furthermore, in many applications, Y | X shifts are largest for marginalized communities; targeting these communities for further data collection raises privacy concerns and could result in predatory inclusion.

- Could the authors clarify if "Average Accuracy" (line 283) refers to a micro or macro-average?

**Relation To Prior Work:**

- The authors should expand on the connections between their recommendation to effectively collect more data from the target (by focusing on groups with the worst predicted Y | X transfer) and: (1) existing active learning algorithms (line 213), and (2) importance sampling in imitation learning.

- The authors should also expand on the connections between their recommendation to add more relevant features and: (1) causal relation learning, (2) feature sensitivity analyis, and (3) feature importance algorithms.

- The authors discuss existing distribution shift benchmarks and prior work on identifying shifts (in the Appendix), but the authors additionally need to discuss related work on understanding/taxonomizing different distribution shifts (including outside the tabular domain); this could include surveys of observed real-world distribution shifts or other analyses of shifts in common datasets.

**Summary And Contributions:**

The authors evaluate the performance of different models on tabular data under specific distribution shifts. The authors curate 7 realistic shifts (e.g., geographic location, time, age) from 5 existing real-world datasets and test 22 shift methods and over 86,000 model configurations on them. The authors also propose a technique for identifying covariate regions where Y | X shifts are largest; the authors then use their technique to create interventions that improve model transfer for these regions.

---

> ### Author Response · Authors · 2023-08-17
> **Rebuttal**
>
> Thank you for the valuable comments. We have begun drafting an in-depth discussion on the limitations of the work in Appendix B, and plan to move it to the main body in the camera-ready version.
>
>
> We address specific points below.
>
> ### **Practical and ethical implications of collecting data from the target distribution**
>
> * **Limitations of Algorithm 1**:
> Algorithm 1 identifies covariate regions with large $Y|X$-shifts. Since it relies on a small target dataset, it cannot be applied when data collection is impossible in the target distribution. When employing non-algorithmic interventions based on Algorithm 1,  researchers must leverage domain knowledge. For instance, our analysis in Section 3.2  is based on our prior understanding of the differing official languages in the two states.
>
> * **Data collection**:
> As demonstrated in Section 3.2, our technique, when paired with several standard methods in our benchmark, yields significantly improved outcomes.
> Nonetheless, it is crucial to recognize that in real-world scenarios, this method is only feasible when the target distribution is accessible for sampling.  We will also highlight the risks of collecting data on marginalized groups (privacy concerns, predatory inclusion).
>
> ### **Clarifications**
> * **Average Accuracy**:
> Average accuracy refers to a micro-average, where we simply calculate the accuracy for the binary classification tasks.
>
> * **Retiring Adult**:
> We agree that the **Adult** dataset should be retired, and focus on ACS Income (and other tasks) in our benchmark. We only use **Adult**  in Section 1 as a "prior benchmark" to highlight the limitations of prior work.
>
> ### **Limitations**
>
> As our initial steps indicate, we will do a better job at discussing the limitations of the present work. This includes limitations of our benchmark,  Algorithm 1, and related operational interventions.
>
> ### **Correctness: Choice of Domains in Relative Regret with $X$-shifts**
>
> We now mention why we choose the two subgroups in Section 1 to avoid confusion. We use the largest and smallest demographic subgroups to illustrate since the generalization from the model trained on the largest subgroup to the smallest subgroup is a common way in practice for subgroup generalizations. If we denote the difficulty of distribution shifts by the relative regret, we agree with the reviewer that transferring from the largest group to the smallest group may not be the most difficult shift for a model to handle.  In terms of the difficulty in the generalization performance gap between subgroups, we find that conclusions remain similar when we loop over all the pairs of subgroups. The largest relative regret for Adult, BRFSS, COMPAS dataset, when paired with all possible two subgroups, is 29.4%, 7.7%, and 17.9% respectively, which is similar to the trend shown in Figure 1.
>
> **X-shift is relative mild**: Sorry for the confusion. We have removed this sentence.
>
>
> ### **Relation to Prior Work: More Relevant Literature**
>
> We appreciate your feedback. In response, we will better contextualize our work with the previous literature.
> We have taken initial steps to provide a more comprehensive literature review and will include the following discussion in the main body of the paper in subsequent revisions.
>
> * Existing Benchmarks for Distribution Shifts.
> * Decomposition of Performance Degradation.
> * Relationships between Non-algorithmic Interventions and Specific Algorithmic Interventions, including:
>     * Active Learning.
>     * Imitation Learning.
>     * Causal Learning.
>     * Feature Sensitivity / Importance Analysis.
>     * Region Analysis: Existing Techniques for Identifying Shifts in Features and Regions.

---

> > ### Comment · Reviewer_np4K · 2023-08-18
> > **Significant improvements to paper**
> >
> > I appreciate the authors' careful and thorough addressal of my concerns, as well as answering my questions; I have updated my score accordingly.

---

> > > ### Author Response · Authors · 2023-08-24
> > > **Thank you for your support**
> > >
> > > Thank you for your support! We greatly appreciate your constructive feedback, including the clarification questions and valuable suggestions on how to make the paper easier to digest.

---

### Author Response · Authors · 2023-08-17
**Rebuttal Summary**

We thank all reviewers for the thoughtful feedback. We revised our paper according to their well-taken comments; we highlight key revised contexts in blue.

* **Code release** We have updated our GitHub repo at [https://github.com/namkoong-lab/whyshift](https://github.com/namkoong-lab/whyshift), to  **encompass all necessary codes to utilize our benchmark and reproduce our results**. And we have provided a detailed `README` file and a Jupyter notebook as a tutorial for our `WHYSHIFT` package.

* **Related Work**: We expand on our discussion of the related work in Appendix A---as many have suggested, we will move this section into the main body of the paper with the extra page afforded to us in the camera-ready version. In subsequent revisions, we will clearly position our work in the context of the related literature,
including existing benchmarks on distribution shifts, methods for identifying shifts in features and regions, decomposition of performance degradation, active learning, imitation learning, causal learning, and feature sensitivity/importance analysis. (**Reviewer np4K, Tsg7, AmNg**)
* **Limitations**: For scientific progress, it is important to acknowledge and discuss the limitations of our work. We have taken initial steps in Appendix B, and plan to provide an explicit discussion at the end of the paper in the camera-ready version.  (**Reviewer np4K, wGZu, Tsg7**)
* **Introduction**: We restructured our introduction to provide a clear roadmap. By explicitly highlighting the shortcomings of existing benchmarks, we provide a sharper motivation for the work. (**Reviewer wGZu, AmNg**)
* **Algorithm 1**: We have rewritten Section 3.1 to provide a clearer overview. In addition to a better description of Algorithm 1, we provide more details for our choice of the shared distribution, $S_X$, including in the appendix. (**Reviewer AmNg**)

---

### Decision · Program_Chairs · 2023-09-22

**Decision:**

Accept (Poster)

**Comment:**

This paper introduces a dataset to benchmark model performance under distribution shifts for tabular data. The reviewers agreed that the ideas in the paper are very relevant and well-motivated, with a rigorous and comprehensive evaluation that covers multiple types of shifts. The writing and presentation were appreciated as well. The reviews contain several suggestions, including a clearer and more comprehensive discussion of limitations and negative implications, along with tips on restructuring the writing, as well as other suggestions. After an active discussion with the authors, it is clear that the paper is ready for publication.